# Behavioural pharmacology predicts disrupted signalling pathways and candidate therapeutics from zebrafish mutants of Alzheimer's disease risk genes

**François Kroll[1,2], Joshua Donnelly[1], Güliz Gürel Özcan[1], Eirinn Mackay[1], Jason Rihel[1]\***

[1]Department of Cell and Developmental Biology, University College London, London, United Kingdom; [2]Institut de la Vision, Sorbonne Université, Paris, France

**\*For correspondence:**
j.rihel@ucl.ac.uk

**Competing interest:** The authors declare that no competing interests exist.

## eLife Assessment

This **important** manuscript sets out to identify sleep/arousal phenotypes in larval zebrafish carrying mutations in Alzheimer's disease (AD)-associated genes. The authors provide detailed phenotypic data for F0 knockouts of each of 7 AD-associated genes and then compare the resulting behavioral fingerprints to those obtained from a large-scale chemical screen to generate new hypotheses about underlying molecular mechanisms. The data presented are **solid**, although extensive interpretation of pharmacological screen data does not necessarily reflect the limited mechanistic data. Nonetheless, the authors address most reviewer concerns in their revised version, providing invaluable new analyses. Phenotypic characterization presented is comprehensive, and the authors develop a well-designed behavioral analysis pipeline that will provide considerable value for zebrafish neuroscientists.

**Abstract** By exposing genes associated with disease, genomic studies provide hundreds of starting points that should lead to druggable processes. However, our ability to systematically translate these genomic findings into biological pathways remains limited. Here, we combine rapid loss-of-function mutagenesis of Alzheimer's risk genes and behavioural pharmacology in zebrafish to predict disrupted processes and candidate therapeutics. FramebyFrame, our expanded package for the analysis of larval behaviours, revealed that decreased night-time sleep was common to F0 knockouts of all four late-onset Alzheimer's risk genes tested. We developed an online tool, ZOLTAR, which compares any behavioural fingerprint to a library of fingerprints from larvae treated with 3677 compounds. ZOLTAR successfully predicted that *sorl1* mutants have disrupted serotonin signalling and identified betamethasone as a drug which normalises the excessive day-time sleep of *presenilin-2* knockout larvae with minimal side effects. Predictive behavioural pharmacology offers a general framework to rapidly link disease-associated genes to druggable pathways.

## Introduction

To prevent or slow down disease, therapies must target the biological processes that cause the disease. Genomic approaches like family studies or genome-wide association studies (GWAS) can help in this quest for causal processes by exposing genes that are mutated before disease onset. For finding causal processes, studying the genome is advantageous because the chronology from genomic variant to disease is unambiguous, providing a stronger argument for causality than when

studying the transcriptome or epigenome. In theory, all we need to do after a genomic study is follow the thread from each gene to the biological process in which it is involved. We know from the genomic study that this process increased or reduced risk when it was altered by mutations in the gene, so modulating more forcefully the process with a drug may unlock a larger therapeutic benefit. In practice, the path from genomic variant to druggable process is far from straightforward.

Research on Alzheimer's disease (AD) exemplifies well both the challenge and benefit of translating genomic studies into druggable biological processes. Family studies of early-onset AD in the 1990s (*Schellenberg et al., 1992*) identified causal mutations in amyloid precursor protein (*APP*; *Levy et al., 1990*; *Van Broeckhoven et al., 1990*), presenilin 1 (*PSEN1*; *Campion et al., 1995*), and presenilin 2 (*PSEN2*; *Rogaev et al., 1995*). Subsequent work demonstrated that amyloid beta (Aβ), a small peptide which forms aggregates in the brains of AD patients (*Glenner and Wong, 1984*), was generated by cleavage of APP by γ-secretase (*Haass and Selkoe, 1993*; *Shoji et al., 1992*), of which the catalytic subunit is PSEN1 or PSEN2 (*De Strooper et al., 1998*; *Wolfe et al., 1999*). Consequently, the field naturally converged on the amyloid hypothesis of AD, which posits that the disease is caused by toxic aggregates of Aβ (*Hardy and Selkoe, 2002*). Today, antibodies against Aβ such as lecanemab show promise in slowing down disease progression (*van Dyck et al., 2023*). This story shows that, although a great challenge, genomic studies (family studies of early-onset AD) can be successfully translated into a causal process (Aβ aggregation) that is now targeted by disease-modifying drugs (lecanemab).

However, the beneficial effects from targeting Aβ aggregation currently remain modest despite substantial reductions in brain amyloid burden (*van Dyck et al., 2023*). To completely stop disease progression, anti-amyloid therapy will likely need to be combined with drugs modulating other processes that contribute to the disease (*Hardy and Mummery, 2023*). GWAS have identified tens of genomic loci where sequence variation is associated with late-onset AD, offering an opportunity to discover new causal processes of AD that potentially go beyond the amyloid hypothesis. For example, analysis of cell types enriched for open chromatin at AD-associated loci pointed to a possible critical role of monocytes, macrophages, and microglia in AD progression (*Lu et al., 2017*; *Tansey et al., 2018*). Although GWAS are designed to generate new hypotheses, AD-associated loci have rarely been exploited to find new causal processes in an unbiased, systematic manner. Given the challenges inherent to linking genomic variants to causal biological events, new genomic associations are often first more narrowly interpreted in the context of the amyloid hypothesis. While this interpretation may be correct, it tends to create a self-fulfilling prophecy which leaves little room for the discovery of new causal processes (*Bellenguez et al., 2022*; *Kroll, 2022a*).

As AD is primarily a disease of old age, research often focuses on patients or animal models after disease onset. However, this approach hinders the discovery of new *causal* processes from genomic studies because it largely annuls the advantage of unambiguous chronology—since genomic variants were unambiguously present before disease onset, they must have modulated a process which contributed causally to disease initiation. In patients or animals after disease onset, many biological processes are disrupted, but only a small proportion may be genuinely causal for future disease progression and therefore make suitable targets for disease-modifying therapies. For example, one may find that dopamine is lacking in the basal ganglia of patients and animal models of Parkinson's disease. Treatment with levodopa, a dopamine precursor, temporarily relieves motor symptoms, but does not slow down disease progression, so dopamine deficiency is not in fact a causal process (*Verschuur et al., 2019*). Therefore, perhaps counterintuitively, studying the consequences of AD-associated mutations *early* in life seems more likely to identify processes that are genuinely causal to disease, as any disrupted process is less likely to be a secondary disease consequence.

In practice, how can we quickly follow the thread from a disease-associated gene to a (druggable) biological process in which it is involved? In this work, we describe a behavioural pharmacology approach using zebrafish larvae. Our strategy compares the behavioural profile of knockout zebrafish to a behavioural dataset of wild-type animals exposed to thousands of small molecules to predict causal processes and potential compounds that rescue the phenotype (*Rihel et al., 2010a*). In previous work (*Kroll et al., 2021*), we introduced the use of zebrafish F0 knockouts to study complex traits such as behaviour. In this study, we demonstrate how the F0 knockout method renders this behavioural pharmacology approach fast and scalable to the parallel study of tens of disease-associated genes, rather than one at a time (*Ashlin et al., 2018*; *Hoffman et al., 2016*). As genomic studies of AD can likely be further exploited to find causal processes, we used Alzheimer's risk genes as a case study for

our strategy. The strategy is not specific to any one disease or set of genes. In theory, any measurable change in behaviour could be used to predict the underlying causal pathways and small molecules that normalise this change.

## Results

## Most Alzheimer's risk genes are present in zebrafish and expressed early in development

GWAS point to small portions of the genome where variation in sequence is associated with variation in disease risk but do not readily specify the genes whose altered function are responsible for this association. Therefore, as a starting point, we used a meta-analysis of GWAS on AD that found 37 significant loci and annotated each with the most likely causal gene using mainly statistical colocalisation (*Schwartzentruber et al., 2021*; *Giambartolomei et al., 2014*). To add confidence to these calls, we cross-referenced these causal gene predictions with a transcription-wide association study and risk gene transcripts that undergo differential splicing in AD brains (*Raj et al., 2018*). Finally, we included the three genes that can cause early-onset AD when mutated: *PSEN1* (*Campion et al., 1995*), *PSEN2* (*Rogaev et al., 1995*), and *APP* (*Levy et al., 1990*; *Van Broeckhoven et al., 1990*), yielding a list of 40 genes associated with AD risk (*Supplementary file 1*).

Of these 40 Alzheimer's risk genes, 30 (75%) had at least one annotated orthologue in the zebrafish genome (source: Ensembl). Of those, 17 had one orthologue (e.g. the only zebrafish orthologue of human *SORL1* was *sorl1*); 11 had two orthologues (e.g. the zebrafish orthologues of human *APP* were *appa* and *appb*); and 2 had more than two orthologues (e.g. the zebrafish orthologues of human *MS4A6E* included *ms4a17a*, *ms4a17c.2*, *tmem176l*, and more; *Figure 1a*). A human gene often has two or more zebrafish orthologues because of a teleost-specific whole-genome duplication event around 340 million years ago (*Meyer and Málaga-Trillo, 1999*). There were no annotated orthologues for 10 Alzheimer's risk genes, including *TREM2*.

Next, we used a published single-cell RNA sequencing (scRNA-seq) dataset of the developing zebrafish brain (*Raj et al., 2020*) to ask whether the orthologues of the Alzheimer's risk genes were expressed in zebrafish embryos and larvae. Most of the genes (33/42) were detectable as early as 12 hr post-fertilisation (hpf) and remained expressed throughout development (*Figure 1b*). At 5 days post-fertilisation (dpf), 38 of the 42 orthologues (90%) were expressed.

From these observations, we selected seven high-confidence Alzheimer's risk genes for further study in zebrafish; the orthologues of the three early-onset Alzheimer's genes: *psen1*, *psen2*, *appa/appb*; and four genes associated with late-onset AD: *apoea/apoeb*, *cd2ap*, *clu*, *sorl1*. We chose *APOE* as it is the most well-known genetic risk factor for late-onset AD (*Yamazaki et al., 2019*). *CD2AP*, *CLU*, and *SORL1* were chosen because non-coding variants within or near those genes are repeatedly found by GWAS (*Kunkle et al., 2019*; *Lambert et al., 2013*). At the *CD2AP* locus, *CD2AP* was highly likely to be the causal gene by colocalisation (*Schwartzentruber et al., 2021*). At the *CLU* locus, both *PTK2B* and *CLU* were likely causal (*Schwartzentruber et al., 2021*), but differential splicing of *CLU* correlated with risk of AD (*Raj et al., 2018*). The top variant at the *SORL1* locus was within an intron of *SORL1* (*Schwartzentruber et al., 2021*), but rare protein-coding variants in *SORL1* were likely causal for some early-onset AD patients through haploinsufficiency or deleterious effects on protein function (*Nicolas et al., 2016*; *Pottier et al., 2012*; *Thonberg et al., 2017*).

For these selected Alzheimer's risk genes, we more carefully examined their expression patterns in larval zebrafish. From the scRNA-seq dataset (*Raj et al., 2020*), most of these genes were broadly expressed in the 5-dpf larva in different neuronal populations and other cell types (*Figure 1—figure supplements 1 and 2*). *apoeb* was highly expressed specifically in epidermis progenitors, Müller glia in the retina, and microglia (*Figure 1c*), as observed previously (*Herbomel et al., 2001*; *Kudoh et al., 2001*; *Raymond et al., 2006*; *Thiel et al., 2022*). Across clusters, the highest expression of *cd2ap* was in neurons of the thalamus (*Figure 1—figure supplement 2b*, cluster 3); *clu* was enriched in radial glia (*Figure 1—figure supplement 2c*, cluster 30); and *sorl1* expression was maximal in a cluster of hypothalamic neurons with enriched *tph1a* expression, a marker for serotonergic neurons (*Figure 1—figure supplement 1d*, cluster 35). To confirm and extend these observations, we used in situ hybridization chain reaction (HCR) to label mRNA in 6-dpf larvae. As we observed in the scRNA-seq data, most genes tested (*appa*, *psen1*, *psen2*, *cd2ap*, *sorl1*) were broadly expressed throughout the 6-dpf

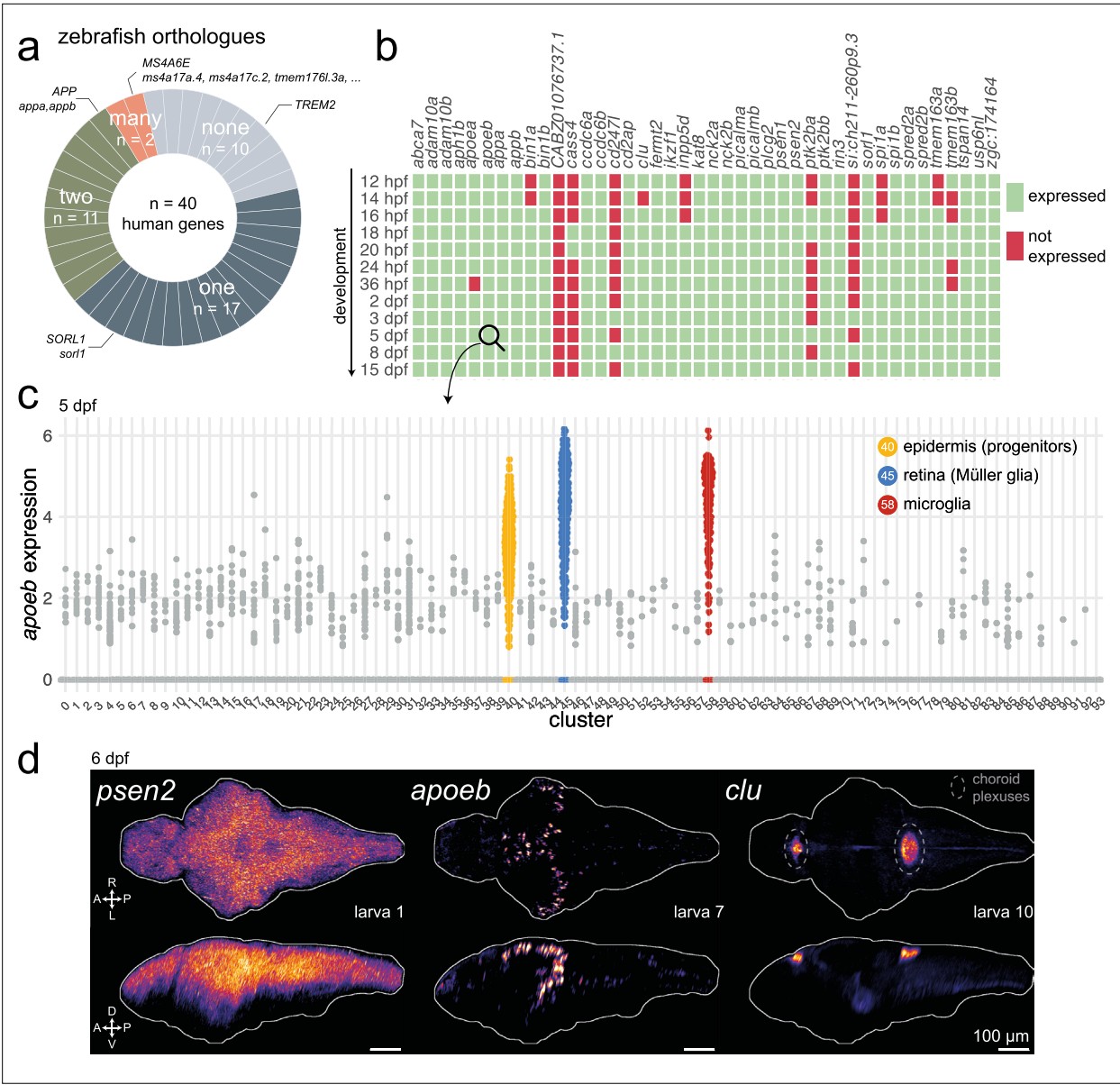

**Figure 1.** Most Alzheimer's risk genes are found in zebrafish and expressed early in development. (**a**) Of 40 Alzheimer's risk genes, 17 had one orthologue in zebrafish; 11 had two orthologues; 2 had more than two orthologues; and 10 did not have any annotated orthologue. More details about orthologues of Alzheimer's risk genes are provided in *Supplementary file 1* (source: Ensembl). (**b**) Expression of Alzheimer's risk genes during early development in zebrafish. Genes were marked as 'expressed' (green) if at least three cells had detectable transcripts in the single-cell RNA-seq dataset from *Raj et al., 2020*. *CABZ01076737.1* is the orthologue of *TSPOAP1*; *cd247l* is the orthologue of *FCER1G*; *si:ch211-260p9.3* is an orthologue of *PLCG2*; *zgc:174164* is an orthologue of *ADAM10*. Other genes have the same name as their human orthologue. The orthologues of *MS4A6E* were not included. hpf, hours post-fertilisation; dpf, days post-fertilisation.(**c**) Expression of *apoeb* in cells of the nervous system at 5 dpf. Each dot represents one cell. Cells are grouped by cluster identity, which are provided in *Supplementary file 1*. Single-cell RNA-seq data and clustering from *Raj et al., 2020*. (**d**) In situ hybridization chain reactions labelling *psen2*, *apoeb*, or *clu* mRNA in the brains of 6-dpf larvae. The images are maximum Z-projections of dorsal (top) and sagittal (bottom) views of three larvae. A, anterior; P, posterior; R, rightwards; L, leftwards; D, dorsal; V, ventral. Larva # labels individual animals across this figure and *Figure 1—figure supplements 3 and 4*. See also *Figure 1—videos 1–9*.

The online version of this article includes the following video and figure supplement(s) for figure 1:

**Figure supplement 1.** Expression of zebrafish orthologues of early-onset Alzheimer's risk genes from single-cell RNA-seq data.

**Figure supplement 2.** Expression of zebrafish orthologues of four late-onset Alzheimer's risk genes from single-cell RNA-seq data.

**Figure supplement 3.** Expression of zebrafish orthologues of early-onset Alzheimer's risk genes in the zebrafish brain.

**Figure supplement 4.** Expression of zebrafish orthologues of four late-onset Alzheimer's risk genes in the zebrafish brain.

*Figure 1 continued on next page*

*Figure 1 continued*

**Figure 1—video 1.** Expression of *appa* in the larval zebrafish brain.
https://elifesciences.org/articles/96839/figures#fig1video1

**Figure 1—video 2.** Expression of *appb* in the larval zebrafish brain.
https://elifesciences.org/articles/96839/figures#fig1video2

**Figure 1—video 3.** Expression of *psen1* in the larval zebrafish brain.
https://elifesciences.org/articles/96839/figures#fig1video3

**Figure 1—video 4.** Expression of *psen2* in the larval zebrafish brain.
https://elifesciences.org/articles/96839/figures#fig1video4

**Figure 1—video 5.** Expression of *apoea* in the larval zebrafish brain.
https://elifesciences.org/articles/96839/figures#fig1video5

**Figure 1—video 6.** Expression of *apoeb* in the larval zebrafish brain.
https://elifesciences.org/articles/96839/figures#fig1video6

**Figure 1—video 7.** Expression of *cd2ap* in the larval zebrafish brain.
https://elifesciences.org/articles/96839/figures#fig1video7

**Figure 1—video 8.** Expression of *clu* in the larval zebrafish brain.
https://elifesciences.org/articles/96839/figures#fig1video8

**Figure 1—video 9.** Expression of *sorl1* in the larval zebrafish brain.
https://elifesciences.org/articles/96839/figures#fig1video9

brain (*Figure 1d*, *Figure 1—figure supplement 3* and *Figure 1—figure supplement 4*). HCR showed strong and widespread expression of *appb*, which contradicts the minimal expression from the *Raj et al., 2018* scRNA-seq dataset but agrees with the scRNA-seq data from DanioCell (*Farrell et al., 2018*). *apoea* expression from HCR was unconvincing, which corroborates the negligible expression found in the scRNA-seq dataset. *apoeb* expression was restricted to cells in the forebrain and optic tectum (*Figure 1d*, *Figure 1—figure supplement 4b*). Based on the scRNA-seq dataset (*Figure 1c*) and literature (*Herbomel et al., 2001*; *Wu et al., 2020*), these cells are likely *ccl34b.1*⁺ amoeboid microglia derived from primitive myeloid precursors from the rostral blood island. *apoeb* was also detected in cells bordering the hindbrain ventricle, which are likely radial glia/astrocytes (*Lowery and Sive, 2005*; *Mu et al., 2019*). Contrary to the widespread expression found in the scRNA-seq dataset, *clu* expression was largely restricted to the diencephalic and myelencephalic choroid plexuses (*Figure 1d*, *Figure 1—figure supplement 4d*), confirming a previous report (*Jiao et al., 2011*).

In summary, around 75% of Alzheimer's risk genes had at least one clear orthologue in zebrafish and most of these were expressed in the brain of 5–6-dpf zebrafish larvae, so they could play a role in early brain development or function.

## The FramebyFrame R package for analysis of sleep/wake behaviour from high-throughput video-tracking data

The next stage in our behavioural pharmacology strategy is to measure sleep/wake behaviour of F0 knockout larvae for the genes under study. To uncover even subtle behavioural phenotypes caused by the loss of Alzheimer's risk genes, we developed a high-throughput sleep/wake tracking assay for zebrafish larvae capable of analysing behaviour at the sub-second resolution over multiple days and nights. To achieve this, we combined previous sleep/wake analysis methods (*Rihel et al., 2010a*; *Lee et al., 2022*; *Rihel et al., 2010b*) with some aspects of the frame-by-frame analysis developed by *Ghosh and Rihel, 2020* into a single software tool, the FramebyFrame R package (github.com/francoiskroll/FramebyFrame, copy archived at *Kroll, 2025a*). We also designed a 3D-printed mesh-bottom plate that supports long-term (up to 8.5 days tested) tracking of larvae with minimal intervention by regulating the water level with a small pump and delivering paramecia for feeding through the mesh from the water bath below (*Figure 2a and b*).

The FramebyFrame package extracts and analyses 17 behavioural parameters from frame-by-frame Δ pixel data, which represent the number of pixels that changed intensity at each frame-to-frame transition. These parameters capture both behaviours that unfold over multiple minutes or hours, such as sleep (*Figure 2c*), and actions at smaller time scales (<1 s), such as individual

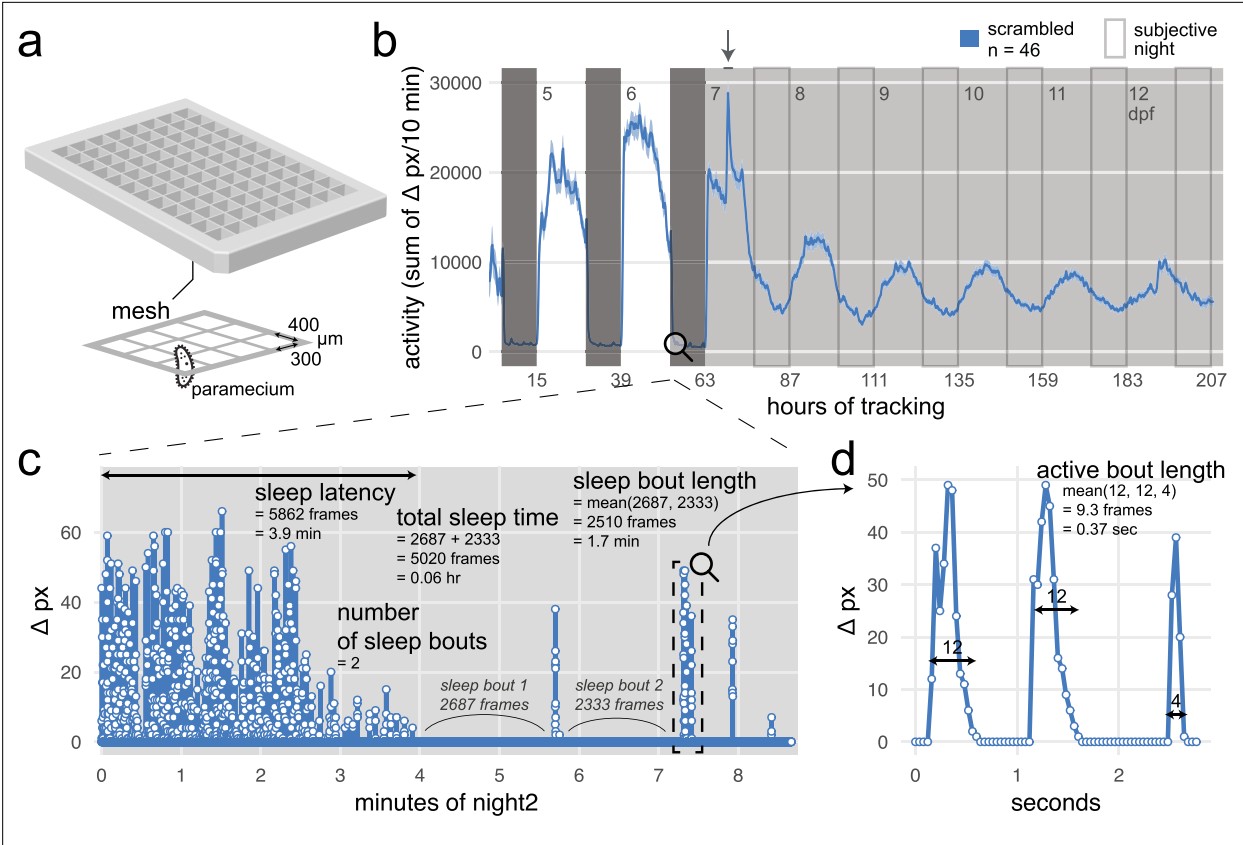

**Figure 2.** Analysis of zebrafish sleep/wake behaviour at the frame-by-frame resolution with the FramebyFrame R package. (**a**) 3D model of the 96-square well mesh-bottom plate. Available at github.com/francoiskroll/FramebyFrame (copy archived at **Kroll, 2025a**). (**b**) Example of an 8-day (208 hr total) video-tracking experiment. scrambled-injected control larvae were in a mesh-bottom plate placed in a water bath containing paramecia. Larvae were tracked for 63 hr on a 14 hr:10 hr light:dark cycle (white and dark grey backgrounds, respectively), then switched to constant dim light (30 lux) for 145 hr (subjective nights are framed). The larvae were tracked from 4 to 13 dpf and all appeared healthy at the end of the experiment. The trace is the mean ± SEM across larvae of the activity (sum of Δ pixels/10 min). The arrow indicates when the water in the bath was replaced and fresh paramecia were supplemented, causing a spike in activity. (**c**) The FramebyFrame R package calculates 17 parameters from the Δ pixel timeseries of each larva. The parameters are grouped in three categories: activity parameters, active bout parameters, and sleep parameters. Here, sleep parameters calculated by the FramebyFrame R package are annotated on the first ~9 min of the Δ pixel timeseries for one wild-type larva starting at lights off. These parameters describe the sleep behaviour of each larva during each day or night. Sleep parameters are also calculated for days, except for sleep latency. This plot is connected to (b) for illustrative purposes only; the data was not from the same experiment. (**d**) Example of an active bout parameter—active bout length—annotated on the sample data from (c). For all the activity and active bout parameters calculated by the FramebyFrame R package, see *Figure 2—figure supplement 1*.

The online version of this article includes the following figure supplement(s) for figure 2:

**Figure supplement 1.** Behavioural parameters calculated by the FramebyFrame R package on the Δ pixel timeseries.

**Figure supplement 2.** Sleep detection by the FramebyFrame R package.

swimming bouts (*Figure 2d*). The parameters are grouped into three categories: activity parameters (*Figure 2—figure supplement 1a*), which describe the overall activity of each larva during a complete day or night; active bout parameters (*Figure 2—figure supplement 1b*), which describe the structure of individual swimming bouts; and sleep parameters (*Figure 2c*) such as sleep latency (*Figure 2—figure supplement 2d*). In zebrafish larvae, sleep is defined as any period of inactivity (Δ pixel = 0) lasting longer than 1 min, a definition based on increases in arousal threshold and homeostatic rebound following deprivation (*Rihel et al., 2010b*; *Prober et al., 2006*). This frame-by-frame analysis has several advantages over previous methods that analysed activity data at the 1-min resolution (*Rihel et al., 2010a*; *Lee et al., 2022*; *Rihel et al., 2010b*). First, individual swimming bouts could not be resolved by these methods as single bouts last ~0.2 s on average. Second, the 1-min methods missed around one third of all sleep bouts because of how the Δ pixel

data is binned in 1-min epochs (*Figure 2—figure supplement 2a*). At the frame-by-frame resolution, the start and end frame of each sleep bout can be precisely determined, which also improves the measurement accuracy of sleep bout duration and sleep latency, the delay from lights-off to first sleep bout. For example, during a sample 10 hr night (*Figure 2—figure supplement 2b*), the frame-by-frame analysis detected 35 ± 14 more sleep bouts, longer average sleep bout lengths (+0.2 ± 0.2 min), and shorter sleep latencies (1.6 ± 2.3 min earlier), resulting in a 42% increase in total sleep time (+2.3 ± 0.4 hr). During the day, the FramebyFrame analysis also detected more sleep and more sleep bouts (*Figure 2—figure supplement 2c*). Most behaviour plots included in this study were created using the FramebyFrame package in just a few commands from the raw frame-by-frame data.

## *psen2* knockouts sleep more during the day

Next, we used CRISPR-Cas9 to generate F0 knockouts for the zebrafish orthologues of the three genes that can cause early-onset AD when mutated in humans—*psen1*, *psen2*, *appa/appb*—and tested these mutants for behavioural and other phenotypes. We focused first on *psen1* and *psen2*.

Zebrafish Psen1 and Psen2 are highly similar to their human counterparts (~70% identical amino acid sequence, *Figure 3a and b*), with the same critical amino acid at the four annotated active sites (two per presenilin). Mismatches between the human and zebrafish proteins were largely clustered into disordered domains, suggesting that the zebrafish presenilins have the same catalytic activity as in humans. The Cas9/guide RNA ribonucleoproteins (RNPs) used to generate F0 knockouts mutated virtually every copy of the genome at each of the three targeted sites. At each targeted locus of *psen1*, 99.0 ± 2.7% of reads had a small insertion and/or deletion (indel) and 78.6 ± 29.7% of all reads had a frameshift mutation (*Figure 3c*). At each targeted locus of *psen2*, 99.9 ± 0.1% reads had indels and 82.0 ± 33.6% of all reads had a frameshift mutation (*Figure 3d*). As each gene is mutated at three loci, the biallelic knockout probability, that is, the probability of having at least one frameshift mutation on both alleles, cumulatively reached >98% ([francoiskroll.shinyapps.io/frameshiftmodel/](francoiskroll.shinyapps.io/frameshiftmodel/)), indicating that most *psen1* and *psen2* F0 knockout larvae were complete loss-of-function mutants.

In humans, PSEN1 or PSEN2 acts as the catalytic subunit of the γ-secretase protein complex, which is responsible for the cleavage of more than 90 substrates, such as Notch. It also performs the last cleavage of APP to release Aβ, which aggregates in the brains of patients with AD (*Haass and Selkoe, 1993*; *De Strooper et al., 1998*; *Haapasalo and Kovacs, 2011*; *Kang et al., 1987*). Do presenilins also cleave zebrafish Appa/Appb into Aβ? Using a sensitive ELISA-based assay, Aβ40 and Aβ42 were detectable in control (uninjected and scrambled-injected) larvae but not in two clutches of *psen1* F0 knockouts (*Figure 3e*), confirming that zebrafish Psen1 is required for the cleavage of Appa/Appb into Aβ. In contrast, Aβ40 and Aβ42 were detectable in *psen2* F0 knockouts, suggesting that most Aβ is generated by Psen1 in zebrafish and that Psen2 alone cannot compensate. This result extends previous findings that Aβ production in zebrafish is blocked by the γ-secretase inhibitor DAPT (*Özcan et al., 2022*) and matches closely observations in mice, in which loss of PSEN2 had no measurable effect on Aβ40/42 production (*Frånberg et al., 2011*; *Herreman et al., 1999*).

In mice, knockout of *Psen1*, but not of *Psen2*, causes severe skeletal malformations and brain haemorrhages, likely because of impaired Notch signalling (*Herreman et al., 1999*). Homozygous animals die within minutes after birth (*Shen et al., 1997*). Morphologically, our *psen1* F0 knockout larvae developed normally (tested up to 16 dpf) and were indistinguishable from their wild-type siblings, as observed in a *psen1* stable knockout line (*Sundvik et al., 2013*). *psen2* F0 knockout larvae had a mild pigmentation defect (*Figure 3f*, *Figure 3—figure supplement 3a*), which has been previously reported in a *psen2* stable knockout line (*Jiang et al., 2018*). This difference suggests a specific function of Psen2 in melanophore development or function which cannot be fulfilled by Psen1. We also generated *psen1/psen2* double F0 knockouts but most were lethal early in development (~5 dpf) with severe defects in eye development (*Figure 3—figure supplement 1a*), in which the retinal pigment epithelium appeared patchy and some larvae developed oedema around the eye. The tail was severely curved outwards, in exactly the same way as larvae treated with a high dose of the γ-secretase inhibitor DAPT (*Yang et al., 2008*). Incidentally, *Psen1* knockout mice also display defects in the axial skeleton (*Shen et al., 1997*). Overall, these results suggest that zebrafish presenilins are more readily capable of compensating each other during development than in mice, as only double *psen1/psen2* knockout larvae showed a severe morphological phenotype.

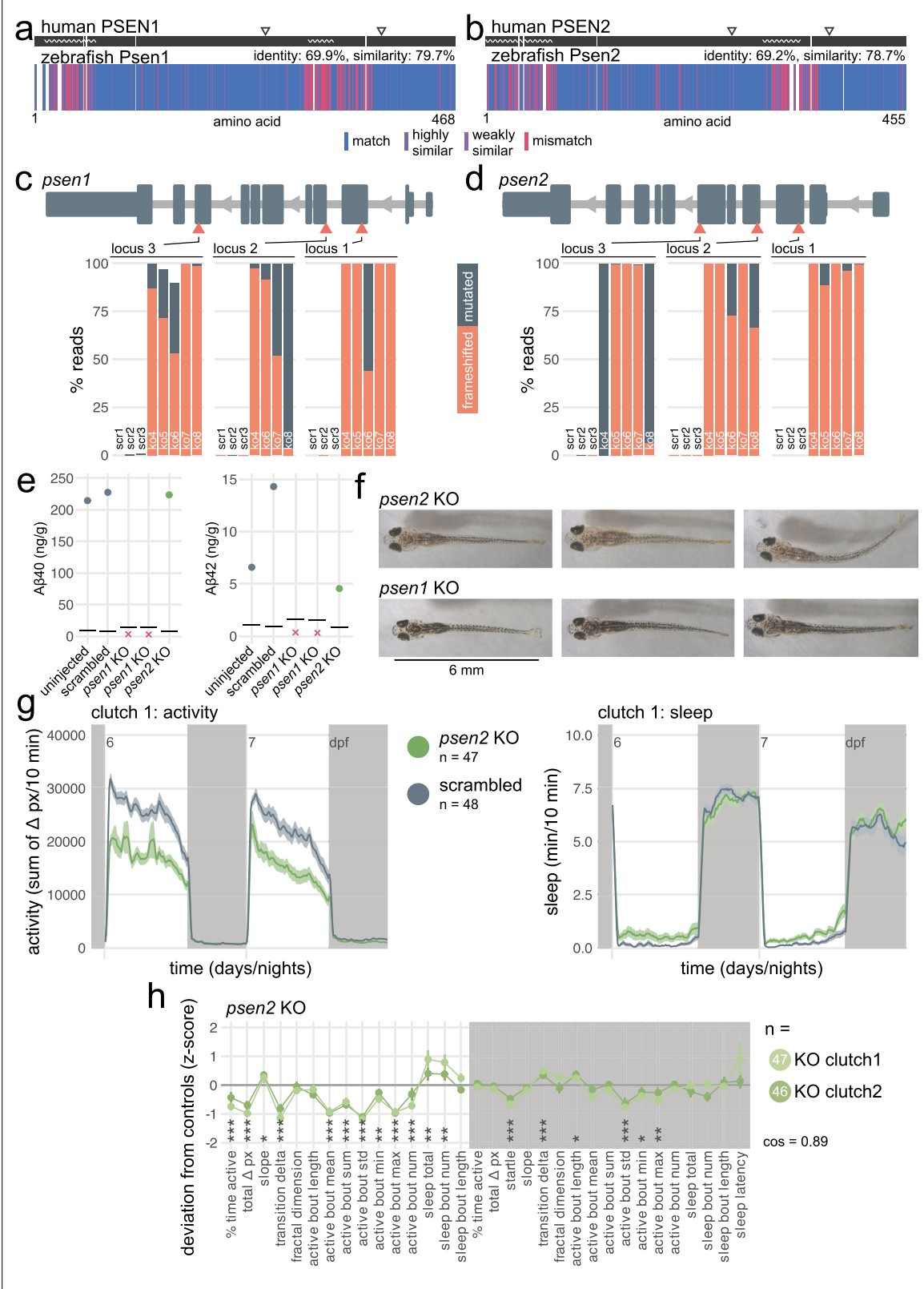

**Figure 3.** *psen2* F0 knockouts initiate more sleep bouts during the day. (**a**) Human PSEN1 amino acid sequence (top) aligned to zebrafish Psen1 amino acid sequence (bottom). In the zebrafish protein, each amino acid (vertical bar) is coloured based on its similarity with the human protein. In the human protein, wavy lines represent disordered domains and arrowheads point to the two active sites at residues 257 and 385 (source: UniProt). White gaps are added when additional residues are present in the other sequence. (**b**) Human PSEN2 amino acid sequence (top) aligned to zebrafish Psen2

*Figure 3 continued*

amino acid sequence (bottom), as in (a). Active sites are at residues 263 and 366 (source: UniProt and AlphaFold). (**c**) (above) Schematic of *psen1* in the 5'–3' genome direction. Exons are in dark grey; tall exons are protein-coding, small are 5'- or 3'-UTR. Light grey lines are introns, and grey arrows represent the direction of transcription. Orange arrowheads mark the target loci. Exons and introns are on different scales. (below) Percentage of reads mutated (height of each bar, with orange representing percentage with a frameshift mutation) at each targeted locus of *psen1*. *scr*, scrambled-injected control larva; *ko*, *psen1* F0 knockout larva. The numbers refer to individual animals. For example, ko4 refers to an individual *psen1* F0 knockout larva for which mutations at each targeted locus are plotted. Across F0 knockout samples: 99.0 ± 2.7% mutated reads, 78.6 ± 29.7% of all reads had a frameshift mutation. (**d**) (above) Schematic of *psen2* in the 5'–3' genome direction, as in (c). (below) Percentage of reads mutated and frameshifted at each targeted locus of *psen2*, as in (c). Across F0 knockout samples: 99.9 ± 0.1% mutated reads, 82.0 ± 33.6% of all reads had a frameshift mutation. (**e**) Concentration of Aβ40 and Aβ42 in pools of n=16–22 uninjected, scrambled-injected, *psen1* F0 knockout, and *psen2* F0 knockout 16-dpf larvae. Each datapoint is the mean of four technical replicates. Concentration unit is ng of Aβ40/42 per g of total protein extracted. Horizontal black line represents the limit of detection. Red crosses indicate samples for which all technical replicates were below the limit of detection. All Aβ38 measurements were below the limit of detection and are not plotted. (**f**) Pictures of *psen2* (top row) and *psen1* (bottom row) F0 knockout larvae at 16 dpf. Note the fainter pigmentation of *psen2* F0 knockout larvae. (**g**) (left) Activity (sum of Δ pixels/10 min) of *psen2* F0 knockout larvae and scrambled-injected siblings during 48 hr on a 14 hr:10 hr light:dark cycle (white background for days, dark grey background for nights). (right) Sleep (minutes per 10 min epoch) during the same experiment. Traces are mean ± SEM across larvae. See also *Figure 3—figure supplement 2a* for results from replicate clutch 2. (**h**) Behavioural fingerprints of two clutches of *psen2* F0 knockout larvae. Each dot represents the mean deviation from the same-clutch scrambled-injected mean for that parameter (z-score, mean ± SEM). Asterisks represent the p-values by likelihood-ratio test on linear mixed effects models calculated on the raw parameter values. *cos*, cosine similarity between the two clutch fingerprints.

The online version of this article includes the following figure supplement(s) for figure 3:

**Figure supplement 1.** *psen1* F0 knockouts are slightly hyperactive at night.

**Figure supplement 2.** sleep/wake behaviour of *psen2* F0 knockouts.

**Figure supplement 3.** The behavioural phenotype of *psen2* F0 knockouts is not an artefact caused by their fainter pigmentation.

**Figure supplement 4.** *appa/appb* F0 knockouts have subdued swimming bouts throughout the day/night cycle.

We next video-tracked two clutches of *psen1* and *psen2* F0 knockout larvae over multiple day-night cycles and applied our frame-by-frame analysis to detect behavioural phenotypes over long and short timescales. The loss of Psen1 only had mild effects on behaviour. At night, *psen1* F0 knockout larvae slept slightly less (−8% vs. scrambled, *Figure 3—figure supplement 1b*) and instead spent more time active than control injected siblings (+22%), mainly because they performed more swimming bouts (+20%). The *psen1* knockouts also showed a slightly reduced startle response at lights-off (−4%), in line with some data obtained from stable *psen1* knockout larvae (*Sundvik et al., 2013*). Behaviour during the day was not affected (*Figure 3—figure supplement 1c*). In contrast, *psen2* F0 knockouts of both clutches were substantially less active than controls during the day (total activity: −26% vs. scrambled, *Figure 3g,h*, *Figure 3—figure supplement 2a,b*), performing both fewer (−17%) and more subdued swimming bouts (active bout mean: −12%). *psen2* F0 knockouts also slept more during the day than controls (*Figure 3g,h*, *Figure 3—figure supplement 2a,b*), both spending more time asleep (+178%) and initiating more frequent sleep bouts (+150%). Loss of Psen2 did not strongly affect night-time behaviour. In summary, *psen2* F0 knockout larvae were substantially less active and sleeping more than controls during the day.

Since *psen2* knockout larvae were less pigmented (*Figure 3—figure supplement 3a*), we tested whether the reduction in activity could be an artefact due to fainter detection by the camera. We extracted the maximum Δ pixel value each larva reached during the startle response at lights off (*Figure 3—figure supplement 3b*). These vigorous swimming bouts can displace the whole larva in a tenth of a second, providing a measure of how many dark pixels are detectable for each larva. *psen2* F0 knockout larvae indeed displaced fewer pixels during the startle response than scrambled-injected controls (*Figure 3—figure supplement 3c*, clutch 1: *psen2* knockouts displaced 78 ± 10 pixels vs. 82 ± 8 pixels for scrambled-injected controls, clutch 2: 71 ± 8 pixels vs 80 ± 8 pixels). We then downscaled the Δ pixel values of the scrambled-injected larvae proportionally to the ratio of the startle response means (*Figure 3—figure supplement 3c,d*), akin to artificially making the scrambled-injected larvae appear less dark. Even after this adjustment, *psen2* F0 knockouts were measurably less active and sleeping more during the day, although the parameter effect sizes were somewhat reduced (*Figure 3—figure supplement 3e*). In summary, the *psen2* behavioural phenotype reflects a genuine difference in behaviour.

Finally, we examined the impact of loss of both Appa and Appb. While overexpression of APP causes AD (*Sleegers et al., 2006*), knockout could point to roles of APP during brain development. As described previously (*Özcan et al., 2022*; *Musa et al., 2001*), zebrafish Appa and Appb are highly similar to human APP (64–68% identical amino acid sequence, *Figure 3—figure supplement 4a and b*). We generated *appa/appb* double F0 knockout larvae by mutating each gene at two loci, rather than three, to limit unviability (*Kroll et al., 2021*). The CRISPR-Cas9 RNPs were highly mutagenic (across all four loci: 95.8 ± 7.0% mutated reads, 76.5 ± 21.7 of all reads had a frameshift mutation, *Figure 3—figure supplement 4c,d*). The *appa/appb* double F0 knockout larvae looked morphologically normal. Behaviourally, *appa/appb* double knockouts were less active during the day than control siblings (–14% vs. scrambled, *Figure 3—figure supplement 4e*) and showed shorter swimming bouts across the day/night cycle (active bout duration during the day: –4%; at night: –3%). Sleep was not consistently affected by the loss of *appa/appb* (*Figure 3—figure supplement 4f*).

In summary, knockout larvae for the three genes associated with early-onset AD had distinct morphological and behavioural phenotypes, with the strongest behavioural changes observed for *psen2* knockouts. Since both presenilins are broadly expressed in the larval brain (*Figure 1d*, *Figure 1—figure supplement 3c,d*), these results indicate that zebrafish Psen1- and Psen2-γ-secretases likely cleave different substrates, such as Appa/Appb which is primarily cleaved into Aβ by Psen1-γ-secretase. Moreover, the behavioural phenotypes of *appa/appb* and *psen1* knockout larvae had little overlap while they presumably both resulted in the loss of Aβ. The *appa/appb* day phenotype could be primarily caused by loss of some Appa/Appb cleavage product not relying on γ-secretase. Alternatively, the net effect of loss of all other Psen1-γ-secretase products could have masked the effect of loss of Aβ.

## Knockouts in all four tested late-onset Alzheimer's risk genes sleep less at night

To test whether knockout of genes associated with late-onset AD also impacted behaviour in larvae, we then generated F0 knockouts for the zebrafish orthologues of *APOE* (*apoea/apoeb*), *CD2AP* (*cd2ap*), *CLU* (*clu*), and *SORL1* (*sorl1*).

APOE shares 22–25% amino acid identity with its two zebrafish orthologues, Apoea and Apoeb (*Figure 4—figure supplement 1a,b*). To make *apoea/apoeb* double F0 knockouts, we targeted each gene at two loci in separate exons with highly mutagenic CRISPR-Cas9 RNPs (*Figure 4—figure supplement 1c,d*). Consistently across three clutches, double *apoea/apoeb* F0 knockout larvae performed slightly more subdued swimming bouts during the day than scrambled-injected siblings (active bout mean: −4%; active bout maximum: −5% vs. scrambled; *Figure 4—figure supplement 1e*). This slight hypoactivity is unlikely to reflect a motor defect because at night, *apoea/apoeb* knockouts performed more swimming bouts (+31%) and had less sleep (−7%, *Figure 4—figure supplement 1e*) than controls.

Knockout of other late-onset AD risk genes also impacted behaviour, especially at night. We mutated zebrafish *cd2ap* (44% identical amino acid sequence vs. human CD2AP; *Figure 4—figure supplement 2a*), at three loci on distinct exons (*Figure 4—figure supplement 2b*). While the two clutches of *cd2ap* F0 knockouts gave generally inconsistent results, knockout larvae of both clutches were more active (time active: +38%; total activity: +44% vs. scrambled) and slept less at night than control siblings (−13%), mainly because they performed more swimming bouts (+34%, *Figure 4—figure supplement 2c*). Similarly, knockout of *clu* (39% identical amino acid sequence vs. human CLU, *Figure 4—figure supplement 3a*) did not strongly affect behaviour during the day (*Figure 4—figure supplement 3c*), despite a particularly high rate of frameshift mutations at all three targeted loci (*Figure 4—figure supplement 3b*). At night, *clu* F0 knockouts were slightly more active (time active: +12% vs. scrambled) and slept less than control siblings (−7%, *Figure 4—figure supplement 3c*). Finally, we generated F0 knockouts for *sorl1* (63% identical amino acid sequence vs. human SORL1, *Figure 4—figure supplement 4a*) by mutating three exons (*Figure 4—figure supplement 4b*). *sorl1* F0 knockout larvae were less active during the day but slept less at night (*Figure 4—figure supplement 5; Figure 4—figure supplements 4c*). Specifically, during the day (*Figure 4—figure supplement 4d*), *sorl1* F0 knockouts spent less time active than scrambled-injected larvae (−15% vs. scrambled), performing fewer swimming bouts (−15%) of approximately the same duration and intensity as controls. Sleep during the day was unaffected. At night (*Figure 4—figure supplement*

*4d*), *sorl1* F0 knockouts spent more time active than control siblings (+20%), largely because their swimming bouts tended to last longer (+6%). They also slept less (−5%), initiating fewer sleep bouts (−9%) of roughly the same duration.

In summary, we video-tracked the sleep/wake behaviours of F0 knockout larvae in four genes associated with late-onset AD: *apoea/apoeb*, *cd2ap*, *clu*, and *sorl1*. Remarkably, loss of all four genes produced a fairly consistent phenotype at night (*Figure 4*), with all knockout larvae spending 5–13% less time asleep and instead spending 12–38% more time active because they were moving 13–34% more often. In contrast, each mutant had distinct day-time behavioural alterations. Comparing with early-onset genes, *psen1* knockouts had similar night-time phenotypes, but knockout of *psen2* or *appa/appb* had no effect on night-time sleep. Therefore, at least some late-onset (and one early-onset) Alzheimer's risk genes have common effects on sleep from an early age, despite being expressed in different tissues and having distinct biochemical properties.

## From behavioural fingerprint to causal process: serotonin signalling disruption by the loss of Sorl1

From genomic studies of AD, we know that mutations in genes such as *SORL1* modify risk by disrupting some biological processes (*Schwartzentruber et al., 2021*; *Nicolas et al., 2016*; *Pottier et al., 2012*; *Thonberg et al., 2017*). Presumably, the same processes are disrupted in zebrafish *sorl1* knockouts, and some caused the behavioural alterations we observed. Can we now follow the thread backwards and predict some of the biological processes in which Sorl1 is involved based on the behavioural profile of *sorl1* knockouts?

To predict disrupted biological processes from the *sorl1* knockout behavioural profile, we developed a behavioural pharmacology approach based on a database of 5756 small molecule behavioural fingerprints (3677 unique compounds) obtained in wild-type larvae (*Rihel et al., 2010a*). First, we used information from the Therapeutic Target Database (*Zhou et al., 2022*) to annotate each compound with its indications (e.g. triprolidine is used to treat hay fever), targets (e.g. triprolidine targets the histamine H1 receptor), and the pathways it affects through its targets (e.g. triprolidine affects the 'inflammatory mediator regulation of TRP channels'). Second, we converted the frame-by-frame behavioural fingerprint of *sorl1* knockouts to the 1-min format used by the database. We then compared the mean *sorl1* knockout fingerprint with each small molecule behavioural fingerprint, creating a ranked list from the small molecule fingerprint most similar to the *sorl1* fingerprint (SU6656: cosine similarity = 0.83) to the small molecule fingerprint most opposite to the *sorl1* fingerprint (nitro-caramiphen HCl: cos = −0.78). Third, we tested using a custom permutation test whether specific indication, target, or pathway annotations were significantly enriched at the top and/or the bottom of the ranked list. More present among the small molecules most correlating and/or anti-correlating with the *sorl1* fingerprint were drugs used to treat depression (*Figure 5—figure supplement 1a*, simulated p-value = 0.049), targeting the serotonin transporter SLC6A4 (*Figure 5a and b*, simulated p-value = 0.015), and affecting the 'serotonergic synapse' pathway (*Figure 5—figure supplement 1b*, simulated p-value = 0.027). Thus, *sorl1* knockout larvae behaved similarly to larvae treated with small molecules targeting serotonin signalling, suggesting that the loss of Sorl1 altered serotonin signalling.

If serotonin signalling is altered in *sorl1* knockouts, they should react differently to serotonergic drugs than wild-type animals. To test this hypothesis, we treated *sorl1* F0 knockouts and controls with citalopram, a selective serotonin reuptake inhibitor (SSRI) used to treat depression (*Andersen et al., 2009*). Citalopram binds the serotonin transporter SLC6A4, preventing the reuptake of serotonin from the synaptic cleft. At a low dose of citalopram (1 μM, added directly to the fish water), *sorl1* F0 knockouts and scrambled-injected controls reacted similarly, sleeping about 1.1 hr more during both day and night (*Figure 5c*). In contrast, at a higher dose of citalopram (10 μM), *sorl1* F0 knockouts had a stronger reaction than their control siblings. For example, sleep during the day increased 2.5× in *sorl1* knockouts, while it only increased 2.2× in controls. The heightened sensitivity of *sorl1* knockouts to citalopram was also apparent when taking all parameters into account by measuring each larva's Euclidean distance from the average $H_2O$-treated sibling of same genotype (*Figure 5d*). Indeed, 10 μM citalopram displaced the *sorl1* knockout larvae further from their behavioural baseline than it pushed controls from theirs. In a second experiment, we treated knockouts and controls with fluvoxamine maleate (10 μM), another commonly prescribed SSRI (*Omori et al., 2010*). *sorl1* F0 knockouts had a weaker reaction to fluvoxamine than their control siblings, the opposite effect than

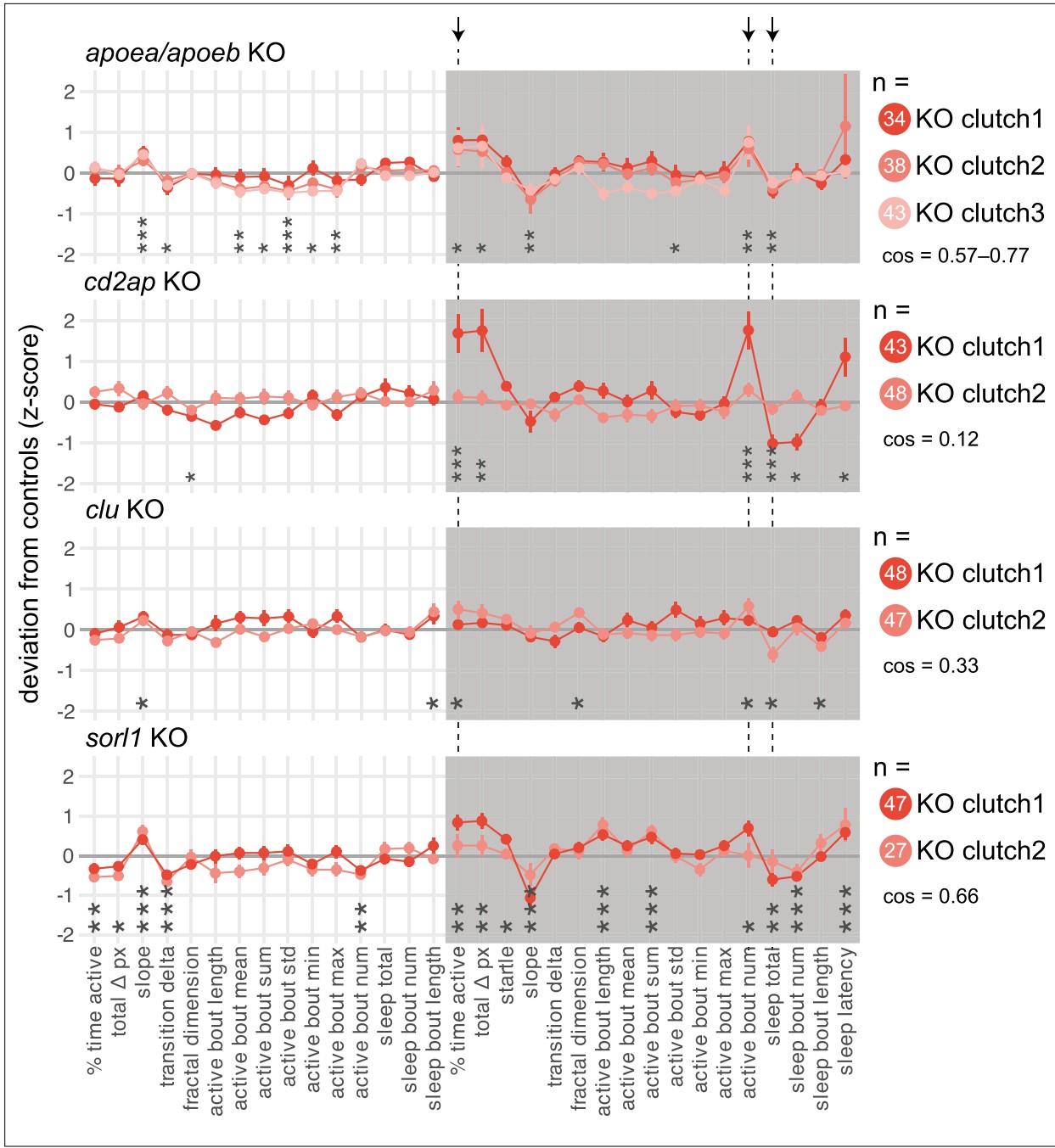

**Figure 4.** F0 knockout larvae in genes associated with late-onset Alzheimer's disease sleep less at night. For each gene: behavioural fingerprints of N = 2–3 clutches of F0 knockout larvae. Each dot represents the mean deviation from the same-clutch scrambled-injected mean for that parameter (z-score, mean ± SEM). Asterisks represent the p-values by likelihood-ratio test on linear mixed effects models calculated on the raw parameter values. *cos*, cosine similarities between fingerprints. Arrows and dashed lines mark the three parameters which are significant for all four late-onset Alzheimer's risk genes tested.

The online version of this article includes the following figure supplement(s) for figure 4:

**Figure supplement 1.** *apoea/apoeb* double F0 knockouts have subdued swimming bouts during the day and sleep less at night.

**Figure supplement 2.** *cd2ap* F0 knockouts are hyperactive at night.

**Figure supplement 3.** *clu* F0 knockouts sleep slightly less at night.

**Figure supplement 4.** *sorl1* F0 knockouts are hypoactive during the day but hyperactive at night.

**Figure supplement 5.** Replicate clutch of *sorl1* F0 knockouts.

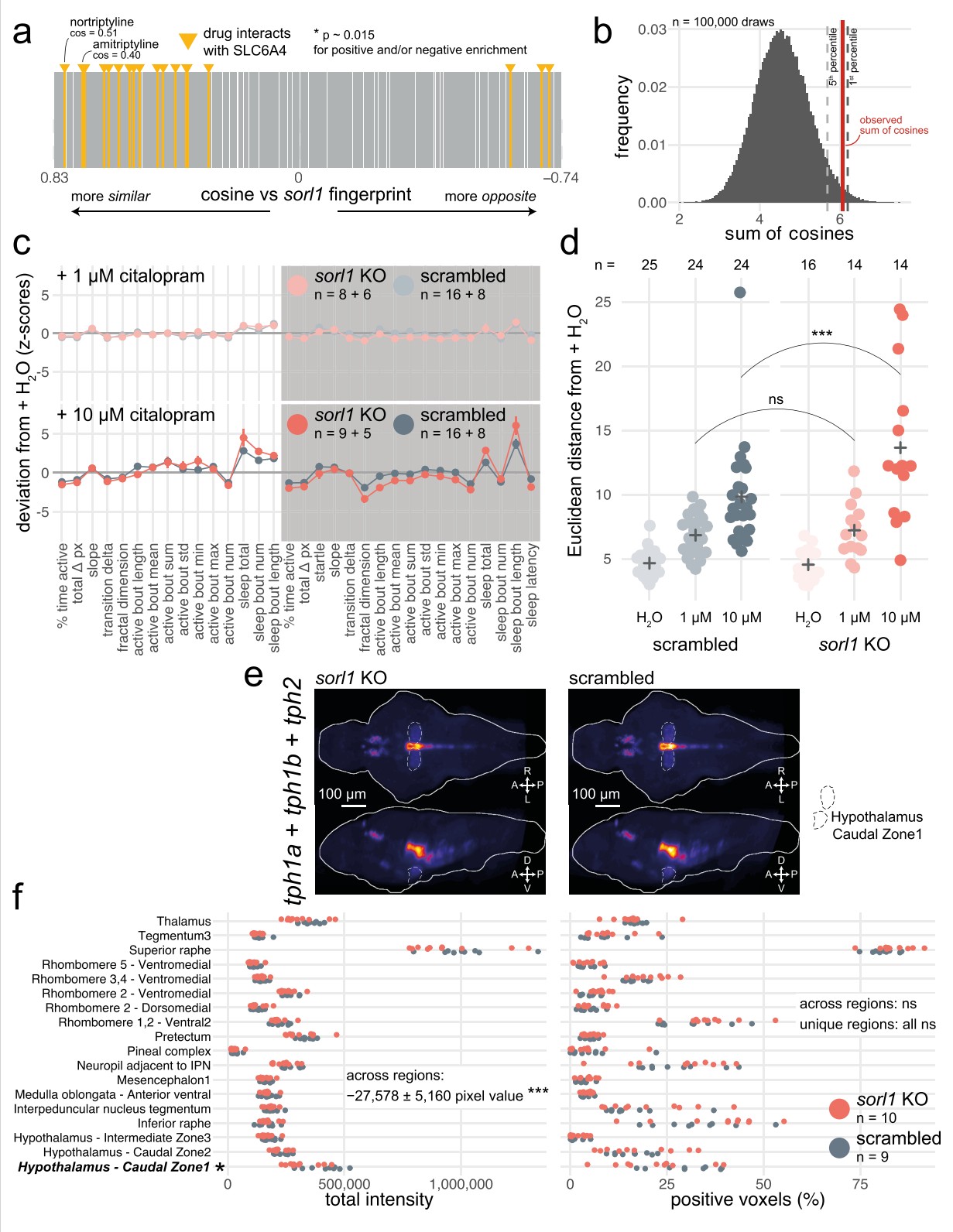

**Figure 5.** Predictive behavioural pharmacology identifies impaired serotonin signalling in *sorl1* knockouts. (**a**) Compounds interacting with the serotonin transporter SLC6A4 tend to generate behavioural phenotypes similar to the *sorl1* F0 knockout fingerprint. 2674 compound–target protein pairs (vertical bars; 1552 unique compounds) are ranked from the fingerprint with the most positive cosine to the fingerprint with the most negative cosine in comparison with the mean *sorl1* F0 knockout fingerprint. Fingerprints of compounds that interact with SLC6A4 are coloured in yellow

*Figure 5 continued on next page*

*Figure 5 continued*

(source: Therapeutic Target Database). Simulated p-value = 0.015 for enrichment of drugs interacting with SLC6A4 at the top (positive cosine) and/or bottom (negative cosine) of the ranked list by a custom permutation test. (**b**) Result of the permutation test for top and/or bottom enrichment of drugs interacting with SLC6A4 in the ranked list. The absolute cosines of the fingerprints of drugs interacting with SLC6A4 (n=18, one fingerprint per compound) were summed, giving sum of cosines = 6.1. To simulate a null distribution, 18 fingerprints were randomly drawn 100,000 times, generating a distribution of 100,000 random sum of cosines. Here, only 1470 random draws gave a larger sum of cosines, so the simulated p-value was p = 1470/100,000 = 0.015 *. (**c**) (top) Behavioural fingerprints of *sorl1* F0 knockouts and scrambled-injected siblings treated with 1 μM citalopram. (bottom) Behavioural fingerprints of *sorl1* F0 knockouts and scrambled-injected siblings treated with 10 μM citalopram. In both plots, each dot represents the mean deviation from the mean of the same-group (F0 knockout or scrambled-injected) untreated ($H_2O$) siblings (z-score, mean ± SEM); therefore, the baseline (z-scores = 0) does not represent the same larvae for *sorl1* F0 knockouts and scrambled-injected controls. Z-scores from two clutches were averaged. (**d**) Euclidean distance from same-group controls' mean across the 32 parameters. ns p=0.71, *** p<0.001 by Welch's t-test. (**e**) HCRs labelling transcripts encoding serotonin transporters (*slc6a4a* and *slc6a4b*) in 6-dpf *sorl1* F0 knockouts and scrambled-injected controls. The images are maximum Z-projections of dorsal (top) and sagittal (bottom) views of the median stack of all larvae in each group. A, anterior; P, posterior; R, rightwards; L, leftwards; D, dorsal; V, ventral. (**f**) Quantification of HCRs from (c). (left) Total grey pixel intensity per anatomical region in *sorl1* F0 knockouts and scrambled-injected controls. Across regions: ns p=0.98; unique regions: ns p>0.25. (right) Number of voxels with positive signal per anatomical region in *sorl1* F0 knockouts and scrambled-injected controls. ** p=0.001, unique regions: ns p>0.07. Statistics across regions by likelihood-ratio test on linear mixed effects models; statistics on unique regions by Welch's t-test without p-value adjustment. The same larvae are plotted in *Figure 5—figure supplement 1e f*.

The online version of this article includes the following figure supplement(s) for figure 5:

**Figure supplement 1.** Predictions of disrupted processes in *sorl1* knockouts based on indications and KEGG pathways.

for citalopram. For example, fluvoxamine doubled (2.1×) sleep bout length at night in *sorl1* knockouts but tripled it (3.4×) in controls (*Figure 5—figure supplement 1c*). Measuring each larva's Euclidean distance from its $H_2O$-treated siblings confirmed the reduced sensitivity of *sorl1* knockouts to fluvoxamine (*Figure 5—figure supplement 1d*). While it is surprising that the *sorl1* knockout larvae reacted oppositely to the two SSRIs, they reacted differently than control larvae in both cases, demonstrating that our behavioural pharmacology approach correctly predicted from behaviour alone that serotonin signalling was in some way altered in *sorl1* knockouts.

There are at least two ways *sorl1* knockouts could react differently to SSRIs. First, compared to wild types, *sorl1* knockouts could undergo a smaller or larger spike of serotonin in the synaptic cleft when reuptake is blocked. This may be because *sorl1* knockouts synthesise serotonin at a different rate, either because they have a different number of serotonergic neurons or a different expression of the enzymes required to make serotonin; or because they do not produce the same amount of serotonin transporter, which would change their sensitivity to SSRIs as a given dose would inhibit a smaller or larger proportion of the transporter than in wild-type animals. Second, *sorl1* knockouts may have a different sensitivity to serotonin itself because post-synaptic neurons have different densities of serotonin receptors.

To distinguish between these two hypotheses, we used HCR to label serotonergic neurons by tagging mRNA coding for tryptophan hydroxylases (in zebrafish: *tph1a*, *tph1b*, *tph2*; *Figure 5e*), an enzyme required for the synthesis of serotonin, and serotonin transporters by tagging *slc6a4a* and *slc6a4b* mRNA (*Lillesaar, 2011*; *Figure 5—figure supplement 1e*). We registered the brain stacks to a common atlas (Zebrafish Brain Browser) and segmented them into 168 anatomical regions. In each region with expression, we counted the number of positive voxels and measured the total signal intensity in each channel (channel 1: *tph1a* + *tph1b* + *tph2*; channel 2: *slc6a4a* + *slc6a4b*). *sorl1* F0 knockouts had slightly more (+2%) *slc6a4a/b* positive voxels than controls across regions, but this difference was not definite for any one region and not replicated when looking at total signal intensity (*Figure 5—figure supplement 1f*). *sorl1* F0 knockouts had on average 18% lower *tph1a/1b/2* signal intensities across regions, particularly in the hypothalamus (hypothalamus – caudal zone 1: –24%, *Figure 5e and f*). Incidentally, of 94 clusters from the 5-dpf scRNA-seq dataset, *sorl1* expression was highest in hypothalamic neurons enriched for *tph1a* expression (*Figure 1—figure supplement 2*). These small differences can help explain the reduced sensitivity of *sorl1* knockouts to fluvoxamine, but do not explain the heightened sensitivity to citalopram. Additionally, at least one study did not find significant changes in serotonin levels in the striatum of *SORL1* knockout mice (*Glerup et al., 2013*), and we did not find genes related to serotonin signalling to be differentially expressed in a bulk RNA-seq dataset of *sorl1* knockout zebrafish brains (*Barthelson et al., 2020*). Support for our

first hypothesis is therefore not overwhelming. Alternatively, different levels of serotonin receptors on the post-synaptic neurons could also contribute.

## From behavioural fingerprint to candidate therapeutic: betamethasone rescues the *psen2* behavioural phenotype

In addition to pointing to the disrupted pathways, behavioural pharmacology can also predict small molecules that may normalise mutant behavioural phenotypes by pointing to compounds that generate the opposite behavioural fingerprint in wild-type larvae.

Using this approach, we sought to identify compounds capable of normalising the behavioural alterations of *psen2* knockouts, such as the increased day-time sleep. We ranked the 5756 small molecule behavioural fingerprints in comparison with the mean *psen2* F0 knockout fingerprint and focused on compounds that generated the most opposite (anti-correlating) fingerprint when applied on wild-type larvae. This identified three compounds with a high negative cosine compared to *psen2* knockouts (*Figure 6a*, *Figure 6—figure supplement 1a*): tinidazole (minimum cos = −0.89), fenoprofen (minimum cos = −0.79), and betamethasone (minimum cos = −0.79). These compounds were also selected because they were each replicated at least once in the database, lending confidence to the prediction. We then applied the three compounds on *psen2* F0 knockouts and measured their effects on sleep/wake parameters. For each treatment, we grouped the behavioural parameters into four categories: 'rescue' if the parameter was significantly altered in DMSO-treated *psen2* knockout larvae but got normalised by the drug; 'missed rescue' if the *psen2* knockout parameter remained altered after drug treatment; 'side effect' if a parameter was unaffected in *psen2* knockouts but got altered by the drug; and 'no effect' if the parameter was unaffected in *psen2* knockouts and remained unchanged by the drug.

All three compounds normalised at least some aspects of the *psen2* knockout behavioural phenotype. Tinidazole did not reduce the abnormally high day-time sleep characteristic of *psen2* knockouts (*Figure 6c* left). It rescued the swimming bout alterations (active bout parameters) during the day but not at night. Overall, the compound barely caused any side effects but only rescued a small aspect of the *psen2* knockout phenotype (9/18 altered parameters rescued). Fenoprofen performed worse (*Figure 6c* middle, 8/18 altered parameters rescued), rescuing some of the swimming bout alterations during the day but aggravating them at night. Furthermore, fenoprofen worsened the day-time hypoactivity of *psen2* knockout larvae, causing sleep to increase further and activity to decrease. Strikingly, betamethasone completely resolved the excess day-time sleep without causing hyperactivity (*Figure 6b*, *Figure 6—figure supplement 1b*). Betamethasone rescued most, but not all, of the swimming bout alterations during both the day and night (*Figure 6c* right, 14/18 altered parameters rescued). Betamethasone did cause a few side effects, likely by making *psen2* knockout larvae overly aroused at the start of the night, extending sleep latency, steepening the slope in activity, and decreasing the startle response. Therefore, of the three drugs we selected using behavioural pharmacology, one almost completely normalised the *psen2* knockout phenotype, albeit with a few side effects.

## Discussion

The F0 knockout and behavioural pharmacology approach successfully predicted the different sensitivity of *sorl1* mutants to serotonin drugs and identified a drug capable of normalising the *psen2* knockout behavioural phenotype. To allow researchers to generate pharmacological predictions from their own sleep/wake datasets, we built an online app (francoiskroll.shinyapps.io/zoltar/) that can plot behavioural fingerprints, rank all 5756 small molecule fingerprints in the database in comparison to the query fingerprint, and perform permutation tests for enrichment of indications, drug targets, and KEGG pathways (*Figure 7*). While we used genes associated with AD as a case study, predictive behavioural pharmacology provides a generalisable framework that can be applied to any set of disease risk genes.

### Disrupted sleep and serotonin signalling—causal processes of AD?

Of the seven genes tested, *psen1* and all four late-onset Alzheimer's risk genes decreased sleep duration at night when mutated in zebrafish larvae. Could disrupted sleep itself be a causal process in AD?

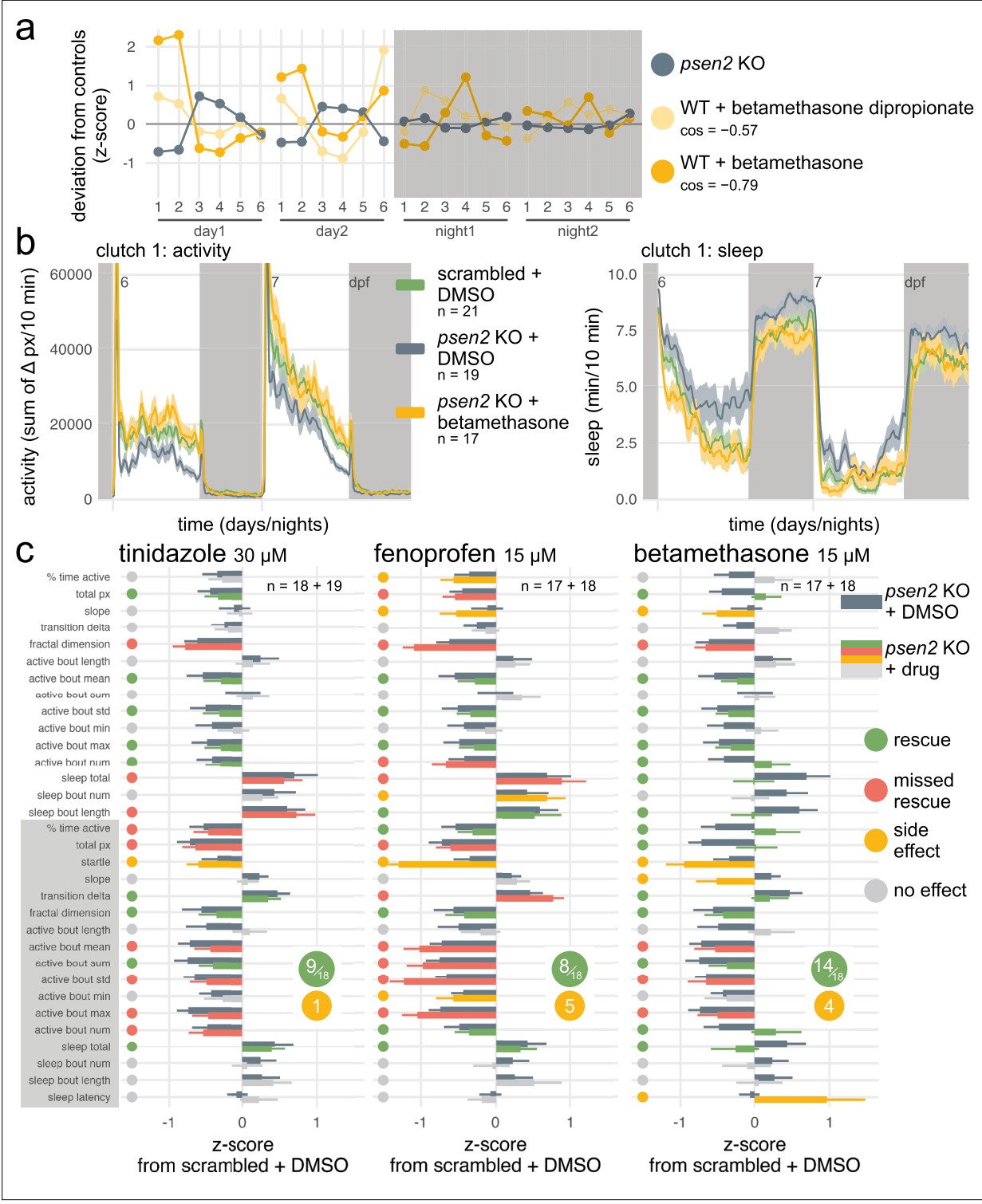

**Figure 6.** Betamethasone normalises the *psen2* knockout behavioural phenotype. (**a**) The *psen2* F0 knockout fingerprint was used as query to identify small molecules that generate the opposite behavioural phenotype when applied on wild-type larvae, returning betamethasone as candidate therapeutic. Plotted are the query *psen2* fingerprint (mean of two clutches, dark grey) and the two betamethasone fingerprints from the drug database with the largest negative cosine similarities (yellow). Parameters: 1, average activity (sec active/min); 2, average waking activity (sec active/min, excluding inactive minutes); 3, total sleep (hr); 4, number of sleep bouts; 5, sleep bout length (min); 6, sleep latency (min until first sleep bout). *cos*, cosine similarity between each betamethasone fingerprint and the *psen2* F0 knockout fingerprint. (**b**) (left) Activity (sum of Δ pixels/10 min) of scrambled-injected larvae treated with DMSO and *psen2* F0 knockout larvae treated with DMSO or 15 μM betamethasone during 48 hr on a 14 hr:10 hr light:dark cycle (white

*Figure 6 continued on next page*

*Figure 6 continued*

background for days, dark grey background for nights). (right) Sleep (minutes per 10 min epoch) during the same experiment. Traces are mean ± SEM across larvae. See also *Figure 6—figure supplement 1* for results from replicate clutch 2. (**c**) Survey of behavioural parameters for each drug treatment. Bars represent the mean deviation from scrambled-injected siblings treated with DMSO (z-score, mean ± SEM). Dark grey bars represent the *psen2* knockouts treated with DMSO, i.e. the phenotype to be treated (same population of *psen2* knockouts treated with DMSO for all the drug treatments, n=19+18). Other bars are colour-coded by the effect of each drug on *psen2* knockouts: 'rescue' (green) if the drug normalised the parameter; 'missed rescue' (red) if the drug failed to normalise the parameter; 'side effect' (yellow) if the drug significantly altered a parameter which was unaffected in *psen2* knockouts; and 'no effect' (grey). Calls were decided based on significance by likelihood-ratio test on linear mixed effects models calculated on the raw parameter values from both clutches (n is sample size of drug-treated *psen2* knockouts of clutch1 + clutch2).

The online version of this article includes the following figure supplement(s) for figure 6:

**Figure supplement 1.** Selection of candidate therapeutics to normalise the *psen2* behavioural phenotype by predictive behavioural pharmacology.

AD patients often take longer to fall asleep, have higher sleep fragmentation, and spend sharply less time in NREM sleep (*D'Atri et al., 2021*; *Prinz et al., 1982*). Sleep disruption can be present before the onset of cognitive deficits. For example, very high sleep fragmentation in elderly people is associated with subsequent diagnosis of AD (*Lim et al., 2013*). Experiments on humans and animal models also point to disrupted sleep being a causal process in some cases. In healthy adults, one night of sleep deprivation was sufficient to cause a 25–30% increase in Aβ signal on PET scans (*Shokri-Kojori et al., 2018*), a ~30% increase in CSF concentrations of Aβ (*Lucey et al., 2018*), and a ~50% increase in CSF concentrations of tau (*Holth et al., 2019*). In wild-type rats and in mice overexpressing mutated human *APP* and *PSEN1*, restricting sleep for 21 days increased Aβ deposits in the cortex (*Kang et al., 2009*; *Zhao et al., 2019*). Conversely, pharmaceutically or chemogenetically consolidating sleep in AD mouse models for 1–2 months delayed Aβ plaque formation (*Kang et al., 2009*; *Jagirdar et al., 2021*). Disrupted sleep is therefore likely to be a causal process in AD. Can it become a therapeutic target? An ongoing clinical trial (NCT04629547) is underway to test whether suvorexant, an orexin receptor antagonist that increases sleep, can reduce Aβ accumulation by consolidating sleep in older adults without dementia (*Herring et al., 2020*). Our observation that disruption of genes associated with AD diagnosis after 65 years reduces sleep in 7-day zebrafish larvae suggest that disrupted sleep may be a common mechanism through which these genes exert an effect on risk. Impaired sleep early in life may be especially deleterious as it is likely essential for brain development: infants spend most of their time asleep during the first year of life (*Ednick et al., 2009*). Incidentally, infants who carry the risk allele of *APOE*, $\varepsilon 4$, show differences in grey matter volume and myelin content in multiple brain regions (*Dean et al., 2014*). Differences are also present in children 9–17 years old who carry the disease-causing *PSEN1* E280A mutation (*Quiroz et al., 2015*). In adults without dementia, higher genetic risk of AD correlate with some sleep phenotypes, sometimes as early as their early twenties (*Chen et al., 2022*; *Leng et al., 2021*; *Muto et al., 2021*). Future work should directly assess whether sleep is impaired in infants at higher genetic risk of AD and whether it is related to these brain structural differences.

Our behavioural pharmacology approach also predicted that loss of Sorl1 impaired serotonin signalling. *sorl1* knockouts did respond differently to SSRIs, but the exact mechanism is unclear. SORL1 acts as an adaptor protein between retromer and several cargo proteins, such as APP (*Jensen et al., 2023*), GDNF receptor α1 (*Glerup et al., 2013*), and glutamate receptor 1 (GLUA1; *Mishra et al., 2022*). Retromer is a large protein complex which 'rescues' proteins from endosomes, targeting them to recycling instead of lysosomal digestion (*Burd and Cullen, 2014*). By disrupting retromer function, mutations in SORL1 disrupt endosomal recycling, which appears as swelling of early endosomes (*Mishra et al., 2023*). As a result, APP remains in endosomes for longer, where it is more likely to be cleaved by β-secretase, the first step towards production of Aβ (*Small and Petsko, 2015*). Similarly, retromer dysfunction also lowers the level of GLUA1 receptors on the neuron membrane, likely because endocytosed receptors are not recycled back to the membrane, which can result in both increased or decreased firing rates in vitro (*Mishra et al., 2022*). We speculate that some serotonin (5-hydroxytryptamine, 5-HT) receptors are also recycled via retromer and SORL1. Consequently, *sorl1* knockouts would react differently to a large spike in serotonin (SSRI treatment) as they have different levels of 5-HT receptors on the post-synaptic membrane. In support for this idea, both 5-HT type 4 receptor (*Joubert et al., 2004*) and SORL1 (*Huang et al., 2016*) interact with sorting nexin 27, a subunit of retromer. Incidentally, the 5-HT receptor type 4 was also an enriched target for *sorl1*

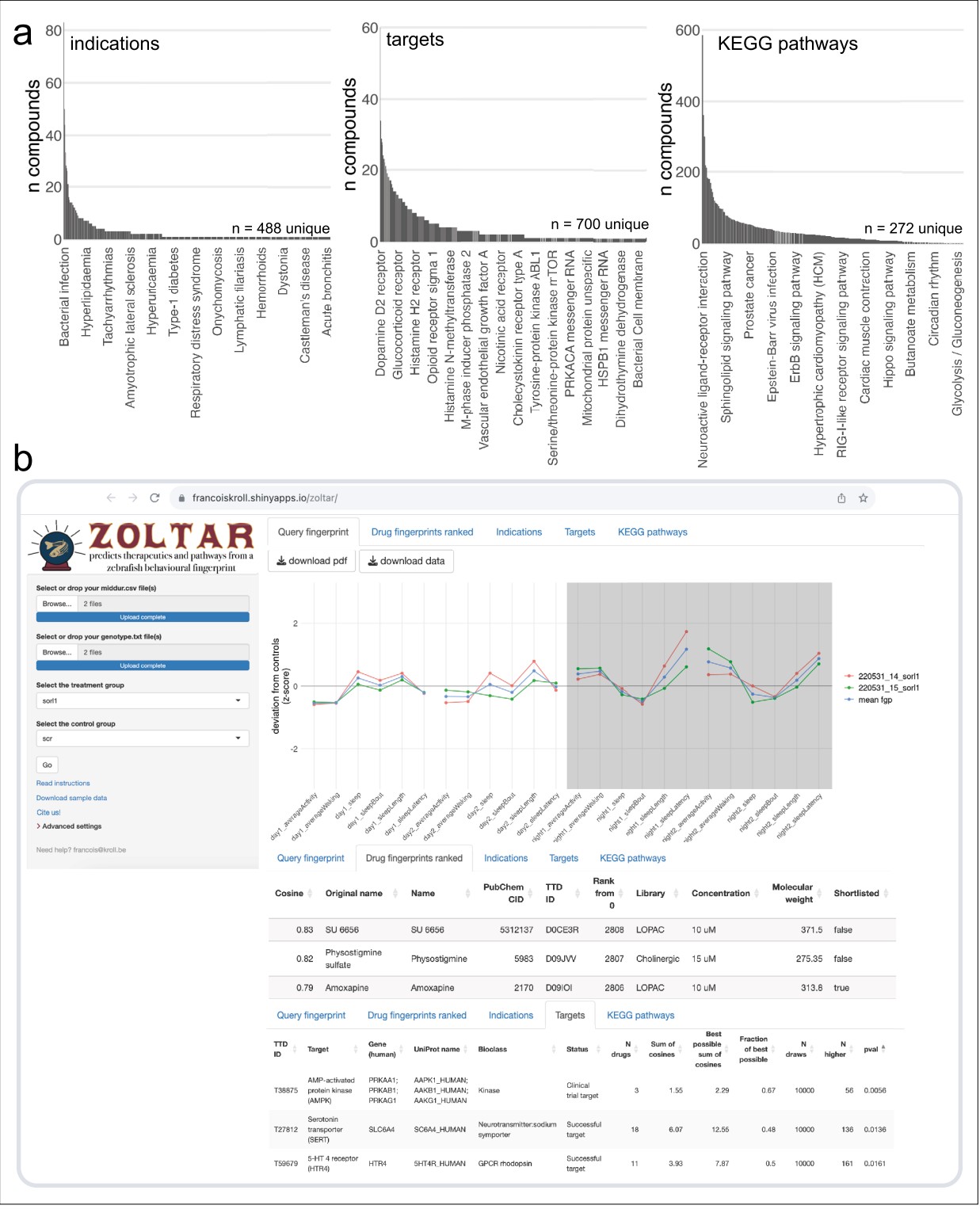

**Figure 7.** The ZOLTAR online app for prediction of therapeutics and disrupted processes from behavioural fingerprints. (**a**) Datasets currently available to the ZOLTAR online app for predictions. The height of each bar represents the number of unique compounds with this annotation. Some annotations are labelled as examples. Of 3677 unique compounds labelled with a PubChem ID, 1123 were annotated with one or more indications; 1552 were annotated with one or more target proteins; and 1140 were annotated with one or more KEGG pathways through their targets. Source of annotations: Therapeutic Target Database. (**b**) Screenshot showing some of the features of the ZOLTAR online app. User drags and drops data file(s) (*middur* = sec active/min for each larva) and file(s) labelling each well with a condition (called 'genotype file', but can be any group assignments). The app reads the groups from the genotype file(s) and the user selects the treatment and control groups in a dropdown menu. The app calculates and plots the

*Figure 7 continued on next page*

Figure 7 continued

fingerprint(s) (tab *Query fingerprint*), then ranks the 5756 small molecule fingerprints (tab *Drug fingerprints ranked*) from maximum positive cosine to maximum negative cosine. Clicking on a row in the table of ranked compounds plots all the fingerprints of this compound (all fingerprints with the same PubChem ID) in comparison with the query fingerprint, as in *Figure 6a*. The app tests, for each annotation, enrichment towards the top and/or bottom of the ranked list with a custom permutation test. Current annotations are clinical indications (tab *Indications*), target proteins (tab *Targets*), and the KEGG pathways each target protein is associated with (tab *KEGG pathways*). Clicking on a row in a table of annotation results generates the 'barcode plot' for this annotation, as in *Figure 5a*. The tables can be downloaded as .csv files and the plots as .pdf files.

(simulated p-value = 0.016). Other observations are consistent with serotonin signalling being a causal process of AD. For example, treatment of an AD mouse model with citalopram caused a rapid decrease in Aβ concentrations in the ISF and eventually reduced amyloid plaque load. Additionally, humans treated with SSRIs within the past 5 years had significantly lower cortical amyloid plaque load on PET scans, suggesting a causal relationship (*Cirrito et al., 2011*). While much work remains to be done, SSRIs can generally be used safely for many years, so modulating serotonin signalling may be an attractive approach to reduce Aβ levels years before the onset of cognitive symptoms (*Cirrito et al., 2011*).

## Limitations and future directions for predictive behavioural pharmacology

Currently, the hypotheses that our behavioural pharmacology approach can test are limited by the small molecule screen dataset (*Rihel et al., 2010a*), which was biased towards specific themes and targets. For example, of 1552 compounds with an annotated target, 60 (3.9%) target the dopamine D2 receptor but only 1 (0.06%) targets mitochondrial uncoupling proteins (UCP1, 2, 3). Consequently, the approach presumably has high sensitivity for disruptions in dopamine signalling, while some hypotheses, such as impairment in mitochondrial coupling, are never formally tested. Broadly, this means that a negative (non-significant) result may not be meaningful if the biological process in question is underrepresented. Additionally, about 30% of the annotated compounds in the database have more than one target. While this is not necessarily an issue when the goal is to find any small molecule which rescues the phenotype, this lack of specificity likely blurs the signal from modulation of a specific target to effect on behaviour. For example, all five compounds which target the histamine H4 receptor also target the histamine H3 receptor, so if the H3 receptor is detected as significantly enriched, the H4 receptor is certain to be significant too, regardless of whether the H4 receptor is indeed specifically disrupted. To develop the biological process prediction, future work should focus on selecting specific agonists and antagonists for a set list of targets, such as the compound VUF-6002 to selectively target the histamine H4 receptor without also activating H3 (*Terzioglu et al., 2004*). Alternatively, ZOLTAR could weigh the evidence for a target based on the compound's affinity constant, such that a weak inhibitor (high Ki) could only contribute proportionally weak evidence for its target in the analysis.

A way to improve the specificity of the approach is to enhance the resolution of the behaviour tracking so that targets which give similar but unique effects on behaviour can be distinguished. We introduced the FramebyFrame R package to enable analysis at sub-second resolution, but the original small molecule screen dataset was collected at the 1-min resolution (*Rihel et al., 2010a*), so comparing a new behavioural profile to the small molecule database first involves losing resolution to make the fingerprints comparable. During a new small molecule screen, one could use SLEAP (*Pereira et al., 2022*) or DeepLabCut (*Mathis et al., 2018*) to extract additional behavioural parameters from video recordings such as speed or position in the well. A faster and higher definition camera could be introduced to allow fine tracking of tail and eye movements. Exposing the larvae to various stimuli is also a way to increase the number of parameters while avoiding redundancy (*Myers-Turnbull et al., 2020*). With more behavioural parameters, the prediction of therapeutics using behavioural pharmacology would also be more precise and the detection of possible behavioural side effects more thorough. Analytically, an improvement could be to represent larval behaviour as a sequence of movements, rather than parameter averages. For example, the Δ pixel time course from our experiments can be encoded as a sequence of bout types termed modules, themselves organised in motifs which are 2–20-module long (*Ghosh and Rihel, 2020*). Alternatively, behaviour could be represented as sequences of known types of swimming bouts (*Marques et al., 2018*) or automatically discovered

'syllables' (*Wiltschko et al., 2020*). In mouse, this sequence representation was better than a finger-print at differentiating similar compounds or even doses (*Wiltschko et al., 2020*), so should also improve the predictive power of the behavioural pharmacology approach.

This leads to a thought experiment: assuming absolute resolution in behaviour tracking, does every knockout give a unique effect on behaviour? In other words, are there ~26,000 distinct knockout fingerprints or do multiple genes converge on a smaller set of possible fingerprints? How much resolution is required to distinguish every possible fingerprint?

## Conclusion

In summary, we present a behavioural pharmacology approach which uses the sleep/wake behaviour of zebrafish larvae as a tool to translate genomic findings into druggable biological processes and candidate therapeutics. We used Alzheimer's risk genes, especially *SORL1* and *PSEN2*, as case studies and correctly predicted mutant-drug interactions from behaviour alone. Our strategy is both scalable thanks to the F0 knockout method and the FramebyFrame analysis package and generalisable beyond AD through the ZOLTAR app. Other neurological conditions for which many risk genes are known, such as autism, epilepsy, or schizophrenia, are prime candidates for the application of the strategy.

## Methods

**Key resources table**

| Reagent type (species) or resource | Designation | Source or reference | Identifiers | Additional information |
|---|---|---|---|---|
| Strain, strain background (*Danio rerio*) | AB ×TL | University College London, London, UK | ZDB-GENO-031202–1 | |
| Strain, strain background (*D. rerio*) | TL | Institut de la Vision, Paris, France | ZDB-GENO-990623–2 | |
| Gene (*D. rerio*) | *appa* | Ensembl | ENSDARG00000104279 | |
| Gene (*D. rerio*) | *appb* | Ensembl | ENSDARG00000055543 | |
| Gene (*D. rerio*) | *apoea* | Ensembl | ENSDARG00000102004 | |
| Gene (*D. rerio*) | *apoeb* | Ensembl | ENSDARG00000040295 | |
| Gene (*D. rerio*) | *psen1* | Ensembl | ENSDARG00000004870 | |
| Gene (*D. rerio*) | *psen2* | Ensembl | ENSDARG00000015540 | |
| Gene (*D. rerio*) | *cd2ap* | Ensembl | ENSDARG00000015224 | |
| Gene (*D. rerio*) | *clu* | Ensembl | ENSDARG00000010434 | |
| Gene (*D. rerio*) | *sorl1* | Ensembl | ENSDARG00000013892 | |
| Genetic reagent (*D. rerio*) | *Tg(elavl3:H2b-GCaMP6s)*[if5Tg] | PMID:25068735 | ZDB-ALT-141023–2 | |
| Genetic reagent (*D. rerio*) | *mitfa*[w2] (*nacre*) | PMID:10433906 | ZDB-ALT-990423–22 | |
| Sequence-based reagent | Alt-R CRISPR-Cas9 crRNAs | IDT | | see *Supplementary file 1* |
| Sequence-based reagent | Alt-R CRISPR-Cas9 tracrRNA | IDT | Cat. #1072533 | |
| Sequence-based reagent | PCR primers | Thermo Fisher | | see *Supplementary file 1* |
| Sequence-based reagent | HCR probes | Thermo Fisher | | see *Supplementary file 1* |
| Sequence-based reagent | HCR amplifiers | Molecular Instruments | | |
| Peptide, recombinant protein | Alt-R S.p. Cas9 Nuclease V3 | IDT | Cat. #1081059 | |

*Continued on next page*

*Continued*

| Reagent type (species) or resource | Designation | Source or reference | Identifiers | Additional information |
|---|---|---|---|---|
| Chemical compound, drug | citalopram hydrobromide | Fisher Scientific | Cat. #15732987 | |
| Chemical compound, drug | fluvoxamine maleate | MedChemExpress | Cat. #HY-B0103A | |
| Chemical compound, drug | tinidazole | Fisher Scientific | Cat. #16594384 | |
| Chemical compound, drug | fenoprofen calcium salt hydrate | Sigma-Aldrich | Cat. #F1517 | |
| Chemical compound, drug | betamethasone | Cayman Chemical | Cat. #20363 | |
| Commercial assay, kit | dsDNA Broad Range Assay | Qubit | Cat. #Q33266 | |
| Commercial assay, kit | Pierce Detergent Compatible Bradford Assay | Thermo Fisher | Cat. #23246 | |
| Commercial assay, kit | V-PLEX Plus Aβ Peptide Panel 1 (4G8) | Meso Scale Diagnostics | Cat. # K15199G | |
| Software, algorithm | ampliCan | PMID:30850374 | | bioconductor.org/packages/release/bioc/html/amplican.html |
| Software, algorithm | ZebraLab | ViewPoint Behavior Technology | | |
| Software, algorithm | FramebyFrame R package | this paper | | https://github.com/francoiskroll/FramebyFrame |
| Software, algorithm | Fiji | ImageJ | | https://imagej.net/software/fiji/downloads |
| Software, algorithm | ZOLTAR | this paper | | https://francoiskroll.shinyapps.io/zoltar/ |
| Software, algorithm | MATLAB R2022b | MathWorks | | |
| Software, algorithm | Python 3 | Python | | |
| Software, algorithm | R v4.2.2 | CRAN | | |
| Other | 96-square well mesh-bottom plate, 3D model | this paper | | https://github.com/francoiskroll/FramebyFrame |

## Animals

Adult zebrafish were reared by University College London's Fish Facility on a 14 hr:10 hr light:dark cycle. To obtain eggs, pairs of one female and one male were isolated in breeding boxes overnight, separated by a divider. Around 9 AM (lights on) the next day, the dividers were removed and eggs were collected 7–10 min later. The embryos were then raised in 10 cm Petri dishes filled with fish water (0.3 g/L Instant Ocean) in a 28.5 °C incubator on a 14 hr:10 hr light:dark cycle. Debris and dead or dysmorphic embryos were removed every other day with a Pasteur pipette under a bright-field microscope and the fish water replaced. At the end of the experiments, larvae were euthanised with an overdose of 2-phenoxyethanol (ACROS Organics). Experimental procedures were in accordance with the Animals (Scientific Procedures) Act 1986 under Home Office project licences PA8D4D0E5 and PP6325955 awarded to Jason Rihel. Adult zebrafish were kept according to FELASA guidelines (*Aleström et al., 2020*).

Wild types refer to AB×TL zebrafish, except for the fluvoxamine experiment (*Figure 5—figure supplement 1*) which used TL larvae. Throughout, F0 knockouts refer to wild-type embryos that were injected with Cas9/gRNA RNPs at the single-cell stage.

## Generation of F0 knockout larvae

### crRNA selection

The crRNA was the only component of the Cas9/gRNA RNP specific to the target locus.

In previous work (*Kroll et al., 2021*), we selected predesigned crRNAs from the Integrated DNA Technologies (IDT) database (eu.idtdna.com) based on predicted on-target and off-target scores calculated by a proprietary IDT algorithm. Subsequent benchmarking in zebrafish of various crRNA design algorithms revealed that the IDT on-target scores did not predict mutagenesis rates in vivo (*Uribe-Salazar et al., 2022*). All crRNAs used in F0 knockout experiments were therefore designed using CHOPCHOP (chopchop.cbu.uib.no) (*Labun et al., 2019a*) implementing both the on-target score calculation by CRISPRScan (*Moreno-Mateos et al., 2015*), which benchmarking determined was the best predictor of mutagenesis (*Uribe-Salazar et al., 2022*), and inDelphi (*Shen et al., 2018*), which predicts to some extent the generated mutations in zebrafish embryos (*Naert et al., 2020*).

We prioritised targeting asymmetrical exons and only selected crRNAs without any off-targets with 0, 1, 2, or 3 mismatches, as off-targets with up to 3 mismatches may undergo mutation (*Kroll et al., 2021*; *Höijer et al., 2022*). We then selected the crRNAs based on their on-target score and the prediction of frameshift mutations by inDelphi (crRNAs used had 82.8 ± 5.7% predicted frameshift mutation rate). We also used SNPfisher, an online database of common single-nucleotide polymorphisms (SNP) in zebrafish wild-type strains, to check that putative target sites did not contain any SNP (*Butler et al., 2015*).

A protocol describing how to select crRNAs based on these developments is available at dx.doi.org/10.17504/protocols.io.81wgb6r5qlpk/v1. Sequences of the crRNAs and information about the targeted loci are provided in *Supplementary file 1*.

### Cas9/gRNA preparation

A protocol describing how to generate F0 knockout larvae is available at dx.doi.org/10.17504/protocols.io.5qpvo52wdl4o/v3.

The CRISPR-Cas9 RNP was made of three components bought from IDT: the crRNA (Alt-R CRISPR-Cas9 crRNA) and tracrRNA (Alt-R CRISPR-Cas9 tracrRNA), together forming the gRNA, and the Cas9 (Alt-R S.p. Cas9 Nuclease V3). The crRNA and tracrRNA were received as pellets, which were individually resuspended in Duplex buffer (IDT, received with the tracrRNA) to form 200 μM stocks. Stocks of crRNA and tracrRNA were stored at –70 °C before use. Cas9 was stored at –20 °C before use. Each crRNA was annealed separately with the tracrRNA by mixing 1 μL crRNA 200 μM; 1 μL tracrRNA 200 μM; 1.28 μL Duplex buffer. The mix was heated to 95 °C for 5 min, then cooled on ice, to obtain a 61 μM gRNA solution. The gRNA solutions were then mixed in equal volumes with Cas9 (1 μL gRNA 61 μM; 1 μL Cas9 61 μM directly from the IDT vial), incubated at 37 °C for 5 min then cooled on ice, generating three 30.5 μM RNP solutions. The three RNP solutions were pooled; the final concentration of each RNP in the pool was thus 10.2 μM and the total RNP concentration 30.5 μM.

The RNPs were usually kept overnight in a 4 °C fridge on ice before injections the following day. Some experiments used RNPs stored at –70 °C for a few weeks.

For experiments targeting two genes simultaneously (*appa/appb*, *psen1/psen2*, *apoea/apoeb*), we targeted each gene at two loci to potentially reduce unviability (*Kroll et al., 2021*). While mutating two loci instead of three is predicted to reduce rates of complete biallelic knockout animals, given the high rate of frameshift mutations achieved with the RNPs that were used, ~90% of injected animals were still expected to be complete biallelic knockouts. Across loci that were mutated during double F0 knockout experiments, 97.5 ± 7.0% of the reads were mutated, and 78.9 ± 24.0% of all reads had a frameshift mutation; see francoiskroll.shinyapps.io/frameshiftmodel/ for the theoretical knockout rate prediction. The preparation followed the same steps as above. The four RNP solutions were pooled, so the final concentration of each RNP in the pool was 7.6 μM and the total RNP concentration was 30.5 μM.

### Injections

Approximately 1 nL of the three-RNP pool was injected into the yolk at the single-cell stage before cell inflation. This amounts to ~30.5 fmol of RNP (30.5 fmol [5029 pg] of Cas9 and 30.5 fmol [1070 pg] of total gRNA). Each unique RNP was present in equal amounts in the pool. Therefore, in the case of three RNPs, ~10.2 fmol of each RNP were co-injected.

When targeting two genes simultaneously, approximately 1.3 nL of the four-RNP mix were injected so the amount of RNP per gene would remain equal to when a single gene is targeted at three loci.

The injected eggs were kept at ~20 °C for ~50 min after injection before transfer to a 28.5 °C incubator, as delaying the first cell division by lowering the temperature was tentatively shown to increase mutagenesis and reduce the diversity of alleles (*Terzioglu et al., 2020*).

### Scrambled RNPs

In all F0 knockout experiments, three or four non-targeting, 'scrambled', crRNAs were prepared into RNPs and injected following the same steps as above. These were created by shuffling the spacer sequence of existing crRNAs until the only predicted targets had 4 or more mismatches in protein-coding sequences, or 3 or more in non-coding sequences. Sequences of the scrambled crRNAs are available in *Supplementary file 1*.

## Preparation of samples for Illumina MiSeq

We sequenced every targeted locus in samples of F0 knockout larvae using Illumina MiSeq. For *psen1*, *psen2*, *apoea*, and *apoeb*, the samples were from video-tracked larvae. The *appa*, *appb*, *cd2ap*, *clu*, and *sorl1* (scr1–2 and ko1–6) larvae were generated expressly to be sequenced. The other *sorl1* samples (scr3–4 and ko7–16) were from video-tracked larvae of the fluvoxamine experiment. For each locus, we generally prepared 5–6 F0 knockout samples and 2–3 scrambled-injected samples. Individual larvae were sequenced at the three or four mutated loci.

The larvae were anaesthetised and their genomic DNA extracted by HotSHOT (*Meeker et al., 2007*) as follows. Individual larvae were transferred to a 96-well PCR plate or strips of 12 tubes. Excess liquid was removed from each well before adding 50 µL of 1× base solution (25 mM KOH, 0.2 mM EDTA in water). Plates were sealed and incubated at 95 °C for 30 min then cooled to room temperature before the addition of 50 µL of 1× neutralisation solution (40 mM Tris-HCl in water). Genomic DNA was then stored at –20 °C.

PCR primers were designed for each target locus using Primer-BLAST (NCBI) to amplify a window of 150–200 bp with at least 30 bp between each primer binding site and the predicted double-strand break site, as this is where most deletions are found (*Kroll et al., 2021*). PCR primers were ordered with a Nextera overhang at the 5′-end of each primer to allow indexing (see *Supplementary file 1*).

Each PCR well contained: 7.98 µL PCR mix (2 mM $MgCl_2$, 14 mM pH 8.4 Tris-HCl, 68 mM KCl, 0.14% gelatine in water, autoclaved for 20 min, cooled to room temperature, chilled on ice, then added 1.8% 100 mg/mL BSA and 0.14% 100 mM d[A, C, G, T]TP), 3 µL 5× Phusion HF buffer (New England Biolabs), 2.7 µL $dH_2O$, 0.3 µL forward primer (100 µM), 0.3 µL reverse primer (100 µM), 0.12 µL Phusion High-Fidelity DNA Polymerase (New England Biolabs), 1.0 µL genomic DNA; for a total of 15.4 µL. The PCR plate was sealed and placed into a thermocycler. The PCR program was: 95 °C – 5 min, then 40 cycles of: 95 °C – 30 sec, 60 °C – 30 sec, 72 °C – 30 sec, then 72 °C – 10 min, then cooled to 10 °C until collection. The PCR product's concentration was quantified with Qubit (dsDNA High Sensitivity or Broad Range Assay) and its length was verified on a 2.5% agarose gel with GelRed (Biotium). Excess primers and dNTPs were removed by ExoSAP-IT (Thermo Fisher) following the manufacturer's instructions. The samples were then sent for Illumina MiSeq, which used MiSeq Reagent Nano Kit v2 (300 Cycles).

## Illumina MiSeq data analysis

Illumina MiSeq data was received as two fastq files for each well, one forward and one reverse. The paired-end reads were aligned to the reference amplicon with bwa v0.7.17 and the resulting bam alignment file was sorted and indexed with samtools v1.11 (*Li et al., 2009*). Alignments were then filtered to keep only reads with less than 20% of its length soft-clipped and spanning at least 20 bp on each side of the predicted Cas9 double-strand break site, 4 bp upstream of the 'N' of the NGG protospacer adjacent motif. Whenever necessary, bam alignment files were visualised with IGV v2.16.0. The resulting filtered bam file was converted back to a forward and a reverse fastq file using bedtools v2.30.0 (*Quinlan and Hall, 2010*). The filtered fastq files were used as input to the R package ampliCan v1.20.0 (*Labun et al., 2019b*), together with a csv configuration file containing metadata information about the samples. AmpliCan was run with settings min_freq = 0.005 (any mutation at a frequency below this threshold was considered as a sequencing error), cut_buffer = 12 (any insertion/deletion

starting within ±12 bp of the PAM may be Cas9-generated), and event_filter = FALSE (no filtering of reads), as the reads were already filtered. AmpliCan can normalise mutation counts by ignoring any insertion/deletion found in control samples. On rare occasions, this could artefactually decrease mutation counts in F0 knockout samples because of low-level contamination in control samples (e.g. *Figure 4—figure supplement 4*, locus 1, sample scr1) so we turned this feature off. Instead, we manually checked the scrambled-injected samples in IGV to confirm that there were no insertions or deletions already present in the wild-type background. AmpliCan detected and counted mutations in the reads and wrote results files that were used for subsequent analysis. Any sample with less than 20× paired end (40× single read) coverage were excluded from plots and subsequent analysis, which is why some samples are missing from plots (e.g. locus2, ko5 is missing in *Figure 3c*). Figures like *Figure 3c,d* plot the proportion of mutated reads and the proportion of reads with a frameshift mutation at each locus, as computed by ampliCan. A read was counted as mutated if it contained one or more insertions or deletions (indels), base substitutions were not considered. If a read contained multiple indels, ampliCan summed them to conclude whether the read had a frameshift mutation or not.

## 3D-printed 96-well mesh-bottom plate

The 96-square well mesh-bottom plate (*Figure 2a*) was designed using Fusion 360 and PrusaSlicer. Multiple copies were 3D-printed in clear polylactic acid (PLA) with a Creality Ender-3 3D printer. The 3D model is available at github.com/francoiskroll/FramebyFrame (copy archived at *Kroll, 2025a*).

## Behavioural video-tracking

In F0 knockout experiments, wild-type embryos from separate clutches were injected at the single-cell stage with CRISPR-Cas9 RNPs. Each clutch was from a unique pair of parents allowed to mate for 7–10 min. For each clutch, about half of the embryos were injected with non-targeting 'scrambled' RNPs to generate control siblings. Clutch-to-clutch variability in locomotor activity is substantial in zebrafish larvae, even between wild-type clutches of the same strain (*Joo et al., 2020*). Therefore, keeping clutches from different parents/mating events separate is predicted to increase sensitivity of the assay by reducing variability. More generally, it is crucial never to perform comparisons where treatment larvae (e.g. mutant, transgenic, drug-treated) are from one clutch and the control larvae are from another (i.e. controls are not siblings), as any difference discovered with such an experimental design could be caused by clutch-to-clutch variability.

At 5 dpf, individual larvae were transferred to the wells of 3D-printed mesh-bottom plates (see *3D-printed 96-well mesh-bottom plate*), each sitting in the water bath of a Zebrabox (ViewPoint Behavior Technology). To avoid any potential localisation bias during the tracking, F0 knockout and scrambled-injected larvae were plated in alternating columns of the 96-well plate. From each well, the video-tracking software (ZebraLab, ViewPoint Behavior Technology) recorded the number of pixels that changed intensity between successive frames. To be counted, a pixel must have changed grey value above a sensitivity threshold, which was set at 20. The metric, termed Δ pixel, describes each animal's behaviour over time as a sequence of zeros and positive values, denoting if the larva was still or moving. In ZebraLab, the *freeze* setting was set to 3 and the *burst* setting was set to 200. However, note that these settings determine how the Δ pixel data is summarised into number of seconds active per minute by the software (*middur* parameter, see *One-minute vs. frame-by-frame*); therefore, their values are irrelevant when analysing the data using the FramebyFrame package. Tracking was performed at 25 frames per second on a 14 hr:10 hr light:dark cycle for ~65 hr, generating sequences of roughly 5,850,000 Δ pixel values per animal. During the experiment, a pump on a timer (Kollea Automatic Watering System, Amazon UK) injected fish water from a reservoir to the bath every morning around 9 AM (lights on) to counteract water evaporation. Water could overflow through an exit pipe, whose height was adjusted to set the water level in the bath. The day light level was calibrated at 555 lux with a RS PRO RS-92 light meter (RS Components). Night was in complete darkness with infrared lighting for video recording. Temperature and light transitions were recorded with a HOBO Pendant (Onset Data Loggers) immersed in the water bath. Temperature throughout the experiment was 23–26°C.

At the end of the tracking, we inspected the larvae under a bright-field microscope and excluded from analysis any larva that did not appear healthy or was not responsive to a light touch with a P10

tip. We then randomly selected 2–3 scrambled-injected and 5–6 F0 knockout larvae for sequencing of the targeted loci (see *Preparation of samples for Illumina MiSeq*).

## Behavioural video-tracking with drug treatment

This refers to the experiments treating *sorl1* F0 knockouts with citalopram (*Figure 5c and d*) or fluvox-amine (*Figure 5—figure supplement 1c and d*) and the experiment treating *psen2* F0 knockouts with tinidazole, fenoprofen, and betamethasone (*Figure 6b,c*, *Figure 6—figure supplement 1b*). Broadly, video-tracking was performed as above (see *Behavioural video-tracking*) but larvae were housed in standard 96-square well plates (Whatman) in fish water with drug. Water was topped up every morning shortly after 9 AM (lights on).

Citalopram hydrobromide (Fisher Scientific, #15732987) was stored at room temperature before use. We chose the treatment concentrations (1 µM and 10 µM) based on previous work in zebrafish larvae (*Bachour et al., 2020*) that found that the minimum dose which had a discernible effect on loco-motor activity was 373 µg/L (0.92 µM). A 500× stock (5 mM) for the 10 µM treatment was prepared by diluting 0.0203 g in 10 mL dH$_2$O (molecular weight = 405.3 g/mol). The 5 mM solution was then diluted 1:10 (100 µL citalopram 5 mM + 900 µL dH$_2$O) to obtain a 500× stock (500 µM) for the 1 µM treatment. Using a P1000 pipet with a tip whose end was cut-off, individual 5-dpf larvae were trans-ferred in 650 µL fish water to the wells of clear 96-square well plates (Whatman). 1.3 µL of each 500× stock was added on top of each well, effectively diluting each stock 1:500 (5 mM stock diluted to 10 µM and 500 µM stock diluted to 1 µM). Control wells were topped-up with 1.3 µL dH$_2$O.

Fluvoxamine maleate (MedChemExpress, HY-B0103A) was stored at −20 °C before use. We chose 10 µM as the treatment concentration to match the citalopram experiment. A 500× stock (5 mM) was prepared by diluting the 5 mg received from the vendor in 3.14 mL dH$_2$O (molecular weight = 318.3 g/mol). Aliquots of this stock were kept at −20 °C before the experiment. Treatment was then performed as for citalopram.

Tinidazole (Fisher Scientific, #16594384) was stored at room temperature before use. We chose the treatment concentration (30 µM) based on a preliminary experiment where we video-tracked wild-type larvae treated with 10 µM, 30 µM, 100 µM tinidazole (data not shown). A 50 mM solution was prepared by diluting 0.0247 g in 2 mL DMSO (molecular weight = 247.27 g/mol), we then diluted this solution to 15 mM in DMSO as 500× stock.

Fenoprofen calcium salt hydrate (Sigma-Aldrich, #F1517) was stored at room temperature before use. We chose 15 µM as the treatment concentration based on the results from the *Rihel et al., 2010a* small molecule behavioural screen database. A 200 mM solution was prepared by diluting 0.1114 g in 1065 µL DMSO (molecular weight = 522.6 g/mol), then diluting this solution to 7.5 mM in DMSO as 500× stock.

Betamethasone (Cayman Chemical, #20363) was stored at −20 °C before use. We chose 15 µM as the treatment concentration based on the results from the *Rihel et al., 2010a* small molecule behavioural screen database. A 33 mM solution was prepared by diluting 0.0204 g in 1575 µL DMSO (molecular weight = 393.5 g/mol), then diluting this solution to 7.5 mM in DMSO as 500× stock.

For tinidazole, fenoprofen, and betamethasone, 100 µL of each 500×stock was then mixed in 20 mL fish water in a Falcon, and the solution transferred to one Petri dish per drug. A Petri dish for control larvae was prepared by mixing 100 µL of DMSO in 20 mL fish water. Using a P1000 pipet with a tip whose end was cut-off, individual 5-dpf larvae were transferred in 500 µL fish water to the Petri dishes. Each Petri dish was then topped-up with fish water to 50 mL final volume, which effectively diluted each drug to the final treatment concentration (100 µL 500×stock in 50 mL fish water). The final DMSO concentration was 0.2% (100 µL 100% DMSO in 50 mL fish water). Using a P1000 pipet with a tip whose end was cut-off, individual larvae were then transferred in 650 µL to the 96-square well plates. This process was to avoid adding drug in DMSO directly to the well as the solution sinks in water and may not mix properly.

Video-tracking was then performed with the same settings as above (see *Behavioural video-tracking*). Both mornings shortly after 9 AM (lights on), the wells were manually topped-up with fish water (no drug), which assumes that the drug does not evaporate and was stable in fish water at 23–26°C for a few days.

A protocol for behavioural video-tracking with drug treatment is available at dx.doi.org/10.17504/protocols.io.4r3l27p6pg1y/v1.

## Behavioural data analysis

Behavioural data analysis was performed using the FramebyFrame R package v0.11.0. A tutorial and documentation are available at github.com/francoiskroll/FramebyFrame (copy archived at *Kroll, 2025a*). The important steps of the analysis are summarised below.

The raw file generated by the ZebraLab software was exported into hundreds of xls files each containing 1 million rows of data. Each row of data represented the Δ pixel of one larva at one frame transition. Using the vpSorter(…) function, these data were re-organised in a large csv file where each column was a well, each row was a frame transition, and each cell a Δ pixel value. To visualise activity over time, we smoothed the Δ pixel time course for each larva with a 60-min (~90,000 rows) rolling average then binned the data by summing Δ pixels in 10-min epochs. We generated the activity trace of individual larvae using ggActivityTraceGrid(…) and excluded from subsequent analysis any larva that had an obviously aberrant behaviour, defined as having missed entirely a day-night transition (no jump/drop in the activity trace) or having shown activity during a day lower than the previous night. Each well was assigned to a group using a metadata file generated by genotypeGenerator(…). For each 10-min epoch, we calculated the mean and standard error of the mean (SEM) across larvae in each group to build the activity traces (e.g. *Figure 3g* left).

Sleep was measured on the large csv file containing all the frame-by-frame Δ pixel data. A 60 s (~1500 rows) rolling sum was applied on the Δ pixel time course of each larva, so that any data point that became zero indicated that the previous 60 s of Δ pixels were all zeros, that is, a sleep bout started exactly 60 s ago. The data after the rolling sum was converted into a series of Booleans: *true* if the data point was 0 (i.e. that frame was part of a sleep bout); *false* if the data point was positive (i.e. that frame was not part of a sleep bout, as the larva moved at some point in the preceding 60 s). At this stage, the first 0 of each sleep bout marked the frame at the 60th second of each sleep bout. Therefore, a correction was then applied to extend each sleep bout 60 s in the past by switching the ~1500 frames of *false* before each sleep bout start to *true*. The time spent asleep in 10 min epochs (~15,000 rows) was calculated by counting the number of *true* frames in each epoch and multiplying the counts by 1/ frame rate (typically 1/25 frames per second), which was converted into minutes for the sleep trace. For example, if larva #3 had 5000 out of 15,000 frames marked *true* (i.e. these frames are part of a sleep bout) from minute 20–30 of tracking, that represented $5000 \times \frac{1}{25} \times \frac{1}{60} = 3.33 \; minutes \; asleep$ for that epoch. As for the activity trace, the mean and SEM across larvae in each group were calculated for each 10min epoch to build the sleep traces (e.g. *Figure 3g* right).

15 behavioural parameters were then calculated for each larva and day/night. Two additional parameters, startle response at sunset and sleep latency, were only defined for nights, for a total of 32 unique parameters (15 day parameters and 17 night parameters). Full days/nights during the experiment were: night0, day1, night1, day2, night2. Larvae were 5 dpf during night0, 6 dpf during day1/ night1, 7 dpf during day2/night2, and the experiment was stopped in the morning of 8 dpf. Night0 (5 dpf) was excluded from the analysis as a habituation period; therefore, two day data points and two night data points were calculated for each larva and behavioural parameter. Definitions of the behavioural parameters can be found in the documentation of the FramebyFrame package (github. com/francoiskroll/FramebyFrame, copy archived at *Kroll, 2025a*).

To build the behavioural fingerprints, a z-score was calculated for each day/night, each larva, and each unique parameter. For example, to calculate the z-score for parameter active bout length of larva #5 for day1, we first calculated the mean and standard deviation of all active bout length data points for control larvae during day1. These were then used to calculate larva #5's z-score as:

$$z_{day1} = \frac{x - \mu_{con}}{\sigma_{con}}$$

where $x$ was larva #5's day1 data point, $\mu_{con}$ was the mean of control day1 data points, $\sigma_{con}$ was the standard deviation of control day1 data points. The calculation was repeated for day2. The day1 and day2 z-scores were then averaged to produce a common day z-score for active bout length of larva #5. The process was repeated for each unique parameter, producing 32 z-scores for each larva, which we term each larva's behavioural fingerprint. To draw the fingerprint plots (e.g. *Figure 3h*), the mean and SEM of all z-scores were calculated for each unique parameter and group of larvae. By definition of the z-score, the mean z-score for the control larvae was always 0 and the SEM constant for each

experiment (dependent on the number of control larvae), so the controls' fingerprint was omitted from the plot.

In *Figure 6c*, for each parameter, the mean and SEM of all z-scores from each clutch were calculated separately. The two z-score means and SEMs were then averaged to draw each bar. See *Statistics on behavioural parameters* for the definition of each category.

Each larva's fingerprint can be conceptualised as the coordinates of one data point in a 32-dimension space where each dimension represents one behavioural parameter. In *Figure 5d*, *Figure 5—figure supplement 1d*, the centre of the $H_2O$-treated scrambled-injected larvae or the centre of the $H_2O$-treated *sorl1* F0 knockout larvae was set as the origin of this 32-dimension space (point at coordinates 0, 0, 0, …) by the z-scoring procedure. We then measured the Euclidean distance between each larva's fingerprint and the origin.

## Statistics on behavioural parameters

Each behavioural parameter was statistically compared between F0 knockout and scrambled-injected larvae using linear mixed effects (LME) modelling implemented in the lmer function of the R package lme4 v1.1.31 (*Bates et al., 2015*). The data were parameter values per larva per time window, for example larva #3 slept 4 hr during night2. We always tracked one unique clutch in each ZebraBox run; therefore, the replicate experiment(s) provided both technical and biological replication. The data collected during nights or during days were analysed separately. The fixed effect was the group assignment (genotype or treatment). Random effects were intercepts for clutch assignment (experiment), larva number, and the larva's age (dpf). Random effects of clutch assignment and larva number were modelled as nested, as data points between larvae of the same clutch were expected to be more similar than between larvae of different clutches; and within each clutch, the day1 (or night1) data point of one larva was expected to be similar to its day2 (or night2) data point. The command to create the day or night model was:

$$lmer\left(parameter \sim group + \left(1 \mid experiment/larva\right) + \left(1 \mid dpf\right), data = night\ or\ day\right)$$

The model provided the slope and its standard error reported on top of the parameter plots (e.g. *Figure 3—figure supplement 1b*). A model without the fixed effect group was then created, as:

$$lmer\left(parameter \sim 1 + \left(1 \mid experiment/larva\right) + \left(1 \mid dpf\right), data = night\ or\ day\right)$$

This null model was compared with the full model created above with a likelihood-ratio test, which provided the p-value reported in the figures and legends.

The LME analysis was informed by a published tutorial (*Winter, 2013*). The FramebyFrame R package performs the above LME analysis when generating parameter plots with ggParameterGrid(…) or writing a report of LME statistics using LMEreport(…).

The main likelihood-ratio test (see above) tested the null hypothesis that group assignment had no effect on parameter values. In experiments with more than two groups, each group was also compared to the reference group using estimated marginal means implemented in the R package emmeans, which provided the p-value.

In *Figure 6c*, the LME analysis was performed on data from both clutches, as described above. The colour for each parameter was based on two p-values obtained using estimated marginal means: the p-value when comparing DMSO-treated scrambled-injected controls with DMSO-treated *psen2* F0 knockouts ('knockout p-value'), and the p-value when comparing DMSO-treated scrambled-injected controls with drug-treated *psen2* F0 knockouts ('drug p-value'), such that:

- if knockout p-value <0.05 and drug p-value >0.05, parameter was a 'rescue';
- if knockout p-value <0.05 and drug p-value <0.05, parameter was a 'missed rescue';
- if knockout p-value >0.05 and drug p-value <0.05, parameter was a 'side effect';
- if knockout p-value >0.05 and drug p-value >0.05, parameter was a 'no effect'.

## One-minute vs. frame-by-frame

The 1-min analyses developed previously (*Rihel et al., 2010a*; *Lee et al., 2022*; *Rihel et al., 2010b*) use the *middur* parameter calculated by ZebraLab. The *middur* parameter reports, for each minute, the number of seconds each larva spent above the *freeze* threshold (set by the user, typically 3 Δ pixel)

and below the *burst* threshold (set by the user, typically 200 Δ pixel). In *Figure 2—figure supplement 2b and c*, to better focus the comparisons on the effect of the 1-min binning, we re-calculated the *middur* dataset of the experiment using the FramebyFrame R package (function rawToMiddur(…)) setting the *freeze* threshold to 0. Larvae plotted were the scrambled-injected larvae from *sorl1* F0 knockout clutch 2 (*Figure 4—figure supplement 5*).

## Pilot long-term video-tracking in 96-well plate

For the experiment in *Figure 2b*, we placed 4-dpf scrambled-injected larvae in the wells of a mesh-bottom plate and added to the water bath 50 mL of paramecia culture filtered through a 40 µm cell strainer nylon mesh (Fisherbrand) to remove the large debris which may be detected by the camera of the ZebraBox. Larvae were video-tracked for a total of 208 hr, first on a 14 hr:10 hr light:dark cycle for 63 hr then in constant dim light at 30 lux for 145 hr for the free-running segment. In the afternoon of the first day in constant dim conditions, we removed most of the water from the bath using a 10 mL Pasteur pipette, replaced with fresh fish water, and added another 50 mL of filtered paramecia. As can be seen in *Figure 2b*, this created a sharp rise in activity. We did not intervene for the rest of the experiment. The activity trace was generated as above (see *Behavioural data analysis*).

## Sleep latency in presence of paramecia

We video-tracked wild-type 6-dpf larvae as described above (see *Behavioural video-tracking*) for 24 hr, including a 14 hr night. Larvae were in the wells of a standard 96-square well plate (Whatman), and we added two drops of filtered paramecia in half of the wells. There was no water bath and the light level at night was 0 lux.

## Zebrafish orthologues of Alzheimer's risk genes

We used the GWAS meta-analysis and list of most likely causal genes from *Schwartzentruber et al., 2021*. We downloaded from Ensembl the list of all zebrafish orthologues of human genes (accessed 27/04/2023). More details about GWAS loci, causal genes, and zebrafish orthologues can be found in *Supplementary file 1*.

## Single-cell RNA sequencing data

We downloaded single-cell RNA sequencing (scRNA-seq) data generated by *Raj et al., 2020* from GEO repository GSE158142 (.rds.gz files, Seurat data structure). In *Figure 1b*, each gene was considered expressed if it was detected in at least three cells at this developmental stage. In *Supplementary file 1*, we manually grouped the clusters from *Raj et al., 2020* in broad categories to add colours. The expression data plotted (e.g. *Figure 1c*) were normalised from raw counts by the authors using the 'LogNormalize' method implemented in Seurat (*Stuart et al., 2019*).

## In situ hybridization chain reaction

In situ hybridization chain reaction (HCR) probes were designed as described by *Choi et al., 2018* using a custom Python script. HCR split initiator sequences B1, B3, and B5 were obtained from *Choi et al., 2018*. To generate probe sets, probe pairs were excluded if they fell below melting temperature and %GC thresholds. Probe pairs with strong sequence similarity to off-target transcripts were also excluded. For each target transcript, we generated sets of 15–25 probe pairs. Multiple transcripts were often stained in the same individual larvae. HCR probes were purchased as custom DNA oligos from Thermo Fisher Scientific. HCR amplifiers (B1-Alexa Fluor 488, B3-Alexa Fluor 546, and B5-Alexa Fluor 647) and buffers were purchased from Molecular Instruments. More details about HCR probes can be found in *Supplementary file 1*.

The HCR protocol we used was adapted from MI-Protocol-RNAFISH-Zebrafish (Rev10) (Molecular Instruments). Proteinase K and methanol permeabilisation steps from the protocol were skipped. HCR to label Alzheimer's risk genes was performed in *Tg(elavl3:H2b-GCaMP6s)* larvae homozygous for the *mitfa^{w2}* allele (*nacre*) (*Lister et al., 1999*; *Vladimirov et al., 2014*). 6-dpf larvae were euthanised and fixed in 4% paraformaldehyde (PFA)/Dulbecco's phosphate-buffered saline (DPBS) overnight at 4 °C. Following fixation, larvae were washed three times 5 min in phosphate-buffered saline (PBS). Larvae were then transferred to a SYLGARD-coated Petri dish and the eyes were removed using forceps. Sample preparation was completed by two brief PBST (1×DPBS + 0.1% Tween 20) washes. For the

probe hybridization stage, larvae were first incubated in hybridization buffer at 37 °C for 30 min. Meanwhile, a probe solution was prepared by adding 4 µL of each 1 µM probe set (4 pmol) to 500 µL hybridization buffer. Hybridization buffer was replaced by the probe solution, and larvae were incubated at 37 °C overnight (12–16 hr).

The next day, excess probe solution was removed by washing larvae four times 15 min with 500 µL of probe wash buffer preheated to 37 °C. Larvae were then washed twice 5 min with 5× SSCT at room temperature. Samples were kept at room temperature for subsequent amplification steps. First, larvae were transferred to amplification buffer at room temperature for 30 min. 3 µM stocks of hairpin H1 and hairpin H2 were individually heated to 95 °C for 90 sec then left to cool at room temperature in the dark for 30 min. For HCR on up to 8 larvae together, a hairpin solution was prepared by adding 4 µL (12 pmol) of hairpin H1 and 4 µL of hairpin H2 (12 pmol) to 200 µL amplification buffer. Finally, amplification buffer was removed and the hairpin solution was added to the larvae which were incubated overnight (12–16 hr) in the dark at room temperature. After overnight incubation, excess hairpins were removed by washing in 5× SSCT for twice 5 min, then twice 30 min and finally once 5 min at room temperature. Larvae were transferred to PBS and kept at 4 °C protected from light for up to 3 days.

For a subset of transcripts (*appa*, *appb*, *slc6a4a*, *slc6a4b*, *tph1a*, *tph1b*, *tph2*), HCR was not carried out in *nacre*, *elavl3:H2b-GCaMP6s* larvae. Instead, *gad1b* was labelled as a reference channel. We proceeded as above, with some amendments at the sample preparation stage. Larvae were fixed in 4% PFA with 4% (w/v%) sucrose overnight at 4 °C. Brains were dissected with forceps, removing the eyes and the skin that covers the brain. After dissection, there was a 20 min postfix in 4% PFA with 4% (w/v%) sucrose followed by three washes in PBST to remove the fixative. When performing HCR on *sorl1* F0 knockout and scrambled-injected larvae (*Figure 5e*, *Figure 5—figure supplement 1e*), larvae from both conditions were pooled in a single Eppendorf tube prior to the hybridization steps. Using a scalpel, a portion of the tail was removed from the scrambled-injected larvae to differentiate the genotypes during imaging.

For imaging, larvae were mounted in 1% low melting point agarose (Sigma-Aldrich) in fish water and imaged with a ZEISS LSM 980 equipped with an Airyscan 2 detector (CO-8Y multiplex mode, confocal resolution) and a ZEISS C-Aprochromat 10×/0.45 W M27 objective. The whole brain was imaged without tiling. The image size was 844.29 µm (3188 pixels) × 846.41 µm (3196 pixels). Each pixel was 0.265 × 0.265 × 1.5 µm. The laser wavelength was 639 nm for B5-Alexa Fluor 647; 561 nm for B3-Alexa Fluor 546; 488 nm for GCaMP6s.

Samples in *Figure 1—figure supplement 3a,b*, *Figure 5e,f*, *Figure 5—figure supplement 1e,f* were imaged with a lightsheet ZEISS Z.1 microscope equipped with a W Plan-Aprochromat 10×/0.5 M27 75 mm objective. The image size was 889.56 µm (1920 pixels) × 889.56 µm (1920 pixels). Each pixel was 0.46 × 0.46 × 1.0 µm. The laser wavelength was 638 nm for B5-Alexa Fluor 647; 561 nm for B3-Alexa Fluor 546; 488 nm for GCaMP6s or B1-Alexa Fluor 488 to image *gad1b*.

Brains were registered to the reference brain from the Zebrafish Brain Browser (*Marquart et al., 2015*; *Marquart et al., 2017*) with *elavl3:H2B-GCaMP6s* or *gad1b* as reference channel using ANTs toolbox version 2.1.0 (*Avants et al., 2011*), as described previously (*Antinucci et al., 2019*).

We prepared (*Figure 1d* and *Figure 1—figure supplement 3*, *Figure 1—figure supplement 4*) using Fiji (*Schindelin et al., 2012*). The minimum/maximum and contrast of each stack/channel were adjusted before generating a maximum Z-projection. We created the sagittal views using Image > Stacks > Reslice before generating a maximum Z-projection. Whole-brain dorsal and sagittal outlines were from the Zebrafish Brain Browser.

In the *sorl1* experiments, masks for anatomical regions were from the Zebrafish Brain Browser. To measure total signal intensity in different anatomical regions (*Figure 5f* left and *Figure 5—figure supplement 1f* left), we summed the grey pixel values of every voxel within each mask on the stacks after registration. To count the number of positive voxels in different anatomical regions (*Figure 5f* right and *Figure 5—figure supplement 1f* right), we first rescaled each stack to the maximum grey value of the stack. We then applied a threshold at grey pixel value 15 (8bit image, so grey pixel value 0–255) so that the grey value of any voxel with signal below 15 was turned to 0. The threshold value was decided using the multi-Otsu algorithm with two classes implemented in the scikit-image Python package (*van der Walt et al., 2014*). We then counted for each anatomical region the number of voxels with grey pixel value >0.

To prepare *Figure 5e*, and *Figure 5—figure supplement 1e* we calculated each median stack using MATLAB R2022b. The maximum of all four median stacks were then adjusted to the same value in Fiji to keep intensities comparable. The dorsal and sagittal Z-projections were prepared as above.

## Amyloid beta measurements on *psen1* and *psen2* F0 knockouts

*psen1*, *psen2*, and double *psen1/psen2* F0 knockout larvae were generated as described above (see *Generation of F0 knockout larvae*). Most double *psen1/psen2* F0 knockout larvae died or were severely dysmorphic by 5 dpf. The other larvae were raised until 16 dpf to increase the amount of tissue available for the assay. Larvae were euthanised by an overdose of tricaine then decapitated with a scalpel. Heads were snap-frozen in liquid nitrogen then homogenized using a mechanical homogenizer in 100 µL TBS (50 mM Tris-HCl, pH 8.0) containing 1:200 protease inhibitor cocktail set III (Calbiochem). Homogenates were centrifuged at 16,000× $g$ at 4 °C for 30 min. The supernatant was then collected and stored at –70 °C. We measured total protein concentration of each sample using the Pierce Detergent Compatible Bradford Assay Kit (Thermo Fisher). Aβ38, Aβ40, and Aβ42 concentrations were measured on the Meso Scale Discovery platform (Meso Scale Diagnostics) using the V-PLEX Plus Aβ Peptide Panel 1 (4G8) kit.

Each pool of larvae was measured in four technical replicates: two at the original concentration and two diluted 1:1 in TBS. Meso Scale returned concentrations of Aβ38, Aβ40, Aβ42 in pg/mL. All Aβ38 measurements were below the detection limit. We converted the Aβ concentrations to pg Aβ/µg total protein using the total protein content measured with the Bradford assay. The limit of detection (LOD) for each Aβ species was the concentration of the most diluted standard in pg/mL. We calculated a sample-specific LOD by calculating the Aβ concentration (pg Aβ/µg total protein) that would have given exactly the LOD in pg/mL. For example, the LOD for Aβ40 was 39.79 pg/mL. Therefore, for a sample whose total protein content was 3788 µg/mL, Aβ40 concentration should have been 0.0105 pg Aβ40/g total protein to give exactly the LOD (39.79 pg/mL). Each data point represents the mean of the four technical replicates, except if both diluted replicates were below the sample-specific LOD, in which case the data point represents the mean of the two undiluted technical replicates. If one undiluted sample was below the LOD but not the other, we replaced the below-LOD measurement by the sample-specific LOD itself. If all four technical replicates (undiluted and diluted) were below the sample-specific LOD, the sample was marked with a cross in the plot.

## Annotation of the Rihel et al., 2010 small molecule behavioural database

From the *Rihel et al., 2010a* small molecule zebrafish behavioural database, we extracted the name of each compound tested and the 5756 behavioural fingerprints, which are the mean z-score for each sleep/wake parameter (see *Figure 6a* for the 1-min parameters). The same compound could have alternate database names (e.g. 'Dopamine hydrochloride' and 'Dopamine HCl'), and could vary by its stereochemistry or salts (e.g. atropine sulfate vs. atropine methyl nitrate). Therefore, we first simplified as much as possible each compound name by removing various stereochemistry and salt information. This reduced the number of unique compound names from 5149 to 4731. We then used the webchem R package (*Szöcs et al., 2020*) to automatically search each name in the PubChem database and return the PubChem CID. CID could be retrieved automatically for 4365 (92%) of the names. We manually searched the remaining 366 compound names, eventually leaving only 8 of 4731 compound names without a CID. The final number of unique CIDs was 3677. The Therapeutic Target Database (TTD) uses a custom TTD Drug ID but provided a file for cross-matching PubChem CIDs to TTD Drug IDs (*P1-03-TTD_crossmatching.txt*). We found a TTD Drug ID for 1740 of the 3677 CIDs (47%). Using each TTD Drug ID as query, we then extracted indications and target proteins from databases provided by the TTD using TTD Drug IDs as queries (*P1-05-Drug_disease.txt* and *P1-01-TTD_target_download.txt*). Proteins are given a target ID by the TTD, which we then used as query to extract the KEGG pathways each protein was associated with (*P4-01-Target-KEGGpathway_all.txt*).

As TTD Drug IDs were used to extract annotations but that only 47% of compounds were assigned one, the ZOLTAR app currently uses less than half of the dataset by *Rihel et al., 2010a* for prediction of disrupted pathways. In the future, other sources could be used to enrich the annotations, for example using machine learning to predict target proteins (*Chatterjee et al., 2023*).

## Behavioural pharmacology from *sorl1* F0 knockout's fingerprint

We first simplified the *Rihel et al., 2010a* behavioural database so that each compound (unique PubChem CID) would be present as a single average fingerprint. Without this adjustment, a single compound could in theory drive a significant enrichment because it is present as several replicate fingerprints (e.g. propranolol has 10 fingerprints) all positively or negatively correlating with the query fingerprint. We then measured the cosine similarity (range −1.0–1.0) between the mean fingerprint of the two *sorl1* F0 knockout clutches and each of the 3677 average small molecule fingerprints (one per PubChem CID) from the *Rihel et al., 2010a* database, ranking them from the small molecule fingerprint with the most positive cosine (SU6656: cos = 0.83) to the one with the most negative cosine (nitrocaramiphen HCl: cos = −0.78).

At this stage, the general goal of the analysis is to detect whether compounds with a given annotation (indication, target, or KEGG pathway) are found more towards the top and/or bottom of the ranked list than expected by chance. For example, two cases suggest that inhibition of a pathway is what causes the knockout behavioural phenotype: all compounds *inhibiting* this pathway are found towards the top of the list (positive cosines); or all compounds *activating* this pathway are found towards the bottom of the list (negative cosines). The database may include both agonists and antagonists for a given pathway, so we also want to detect cases where compounds interacting with the pathway are found both towards the top *and* towards the bottom of the ranked list. Therefore, the null hypothesis for a given annotation is specifically that fingerprints with this annotation are found across the ranked list in a pattern that can be explained by chance, or that they are mostly found around the centre of the list (cos ~0 position).

We swapped each compound in the ranked list for its annotations, for example its target(s), keeping for each the original cosine of the compound. If a compound had no annotated target (or indication or KEGG pathway), it was simply deleted. If a compound had multiple annotated targets (or indications or KEGG pathways), all targets received the compound's cosine. For example, aspirin had cos = −0.41 and two annotated targets: HMG-CoA reductase and prostaglandin G/H synthase. Therefore, swapping aspirin for its targets generated two rows with the same cosine −0.41.

To measure enrichment of a given annotation towards the top and/or bottom of the list, we summed the absolute cosines of all its instances. For example, 18 compounds in the database are annotated as binding the serotonin transporter (SLC6A4). In the ranked list of fingerprints for *sorl1*, the sum of those 18 cosines was 6.07. Was this sum of cosines surprising or could it be explained by chance? To test this, we randomly drew 18 positions and measured the sum of cosines 100,000 times to generate a null distribution of sum of cosines (e.g. *Figure 5b*). We counted how many random draws gave a larger sum of cosines than the observed sum of cosines (6.07) to calculate a simulated p-value. In the case of SLC6A4, 1470 random draws gave a larger sum of cosines; therefore, the probability that a sum of cosines as high as 6.07 was obtained by chance was estimated at 1470/100,000 = 0.0147, which we reported as the simulated p-value.

## Behavioural pharmacology from *psen2* F0 knockout's fingerprint

We measured the cosine similarity between the mean fingerprint of the two *psen2* F0 knockout clutches and each of the 5756 small molecule fingerprints from the *Rihel et al., 2010a* database. We then ranked the 5756 small molecule fingerprints from the most positive cosine (3–3-Acetoxypregn-16-en-12,20-dione: cos = 0.82) to the most negative cosine (isogedunin: cos = −0.87). We searched within the ~100 fingerprints with the most negative cosines (range of cosines: −0.72–−0.87) for compounds which *Rihel et al., 2010a* labelled as shortlisted because it affected one behavioural parameter with a large effect size and/or affected the same parameter in the same direction across the two days/nights. From these compounds, we selected fenoprofen and betamethasone because they both had a replicate fingerprint with cosine < −0.50. Tinidazole was selected based on a preliminary experiment in wild-type larvae (data not shown), although it was not shortlisted by *Rihel et al., 2010a*.

To simplify this selection process in the ZOLTAR app (francoiskroll.shinyapps.io/zoltar/), the shortlisted compounds are labelled, and clicking on a candidate compound automatically plots all the fingerprints from this compound (same PubChem CID).

## Protein alignment plots

This refers to the plots as in *Figure 3a and b*. Amino acid sequences were obtained from UniProt. The ClustalOmega algorithm, implemented in R by the msa package, was used to align the sequences. Definitions of 'highly similar' and 'weakly similar' groups of amino acids were taken from the ClustalOmega online documentation. The following groups of amino acids were considered highly similar: STA, NEQK, NHQK, NDEQ, QHRK, MILV, MILF, HY, FYW. The following groups of amino acids were considered weakly similar: CSA, ATV, SAG, STNK, STPA, SGND, SNDEQK, NDEQHK, NEQHRK, FVLIM, HFY. Identity was the number of positions where the zebrafish amino acid matched the human amino acid divided by the total length of the alignment. Similarity was the number of positions where the zebrafish amino acid matched or was highly similar to the human amino acid divided by the total length of the alignment. UniProt IDs of the amino acid sequences used were (human vs. zebrafish): *appa*, P05067 vs. Q6NUZ1; *appb*, P05067 vs. B0V0E5; *psen1*, P49768 vs. Q9W6T7; *psen2*, P49810 vs. Q90ZE4; *apoea*, P02649 vs. Q503V2; *apoeb*, P02649 vs. O42364; *cd2ap*, Q9Y5K6 vs. F1R1N9; *clu*, P10909 vs. Q6PBL3; *sorl1*, Q92673 vs. Q90ZE4.

## Gene schematics

This refers to the plots as in *Figure 3c and d*. Coordinates of 5'-UTRs, 3'-UTRs, and exon boundaries were obtained from Ensembl accessed through R using the biomaRt package v2.54.1 (*Durinck et al., 2009*). For each gene, we drew the transcript that was used as reference when selecting the crRNAs, usually the longest coding transcript annotated in Ensembl. Ensembl IDs of transcripts drawn were: *appa*, ENSDART00000166786; *appb*, ENSDART00000077908; *psen1*, ENSDART00000149864; *psen2*, ENSDART00000006381; *apoea*, ENSDART00000172219; *apoeb*, ENSDART00000058965; *cd2ap*, ENSDART00000102611; *clu*, ENSDART00000127173; *sorl1*, ENSDART00000156995.

## Pictures

For the pictures of the *psen2* F0 knockout larvae and scrambled-injected siblings (*Figure 3f*, *Figure 3— figure supplement 3a*), larvae were anaesthetised and mounted in 1% low melting point agarose (Sigma-Aldrich) in fish water. Pictures were then taken with a Nikon SMZ1500 brightfield microscope with illumination from above the sample.

Pictures of the *psen1/psen2* double F0 knockout larvae (*Figure 3—figure supplement 1a*) were taken with an Olympus MVX10 microscope connected to a computer with the software cellSens (Olympus).

## Statistics

For statistics on behavioural parameters, see *Statistics on behavioural parameters*. For the permutations procedure used in *Figure 5a,b*, *Figure 5—figure supplement 1a,b*, see *Behavioural pharmacology from sorl1 F0 knockout's fingerprint*.

Threshold for statistical significance was $\alpha$=0.05. In figures, ns refers to p>0.05, * to p≤0.05, ** to p≤0.01, and *** to p≤0.001. In text, data distributions are reported as mean ± standard deviation, unless stated otherwise.

In text, estimates of behavioural parameter effect sizes are often reported in % vs. scrambled-injected controls. To calculate those, we first calculated, for each clutch and day or night, the mean of all control data points for this parameter, typically returning four averages (e.g. clutch1 controls, night1 and night2; clutch2 controls, night1 and night2). To each average, the slope from the LME model (see *Statistics on behavioural parameters*) was added/subtracted to estimate the knockout averages. An effect size in % vs. controls (*ef%*) was then calculated for each clutch and day or night as:

$$ef\% \; = 100 \times \left( -1 + \frac{ko}{con} \right)$$

if the slope was positive; or

$$ef\% \; = -100 \times \left( 1 - \frac{ko}{con} \right)$$

if the slope was negative; where *ko* and *con* is the knockout or control average for one clutch and one day or night. The effect sizes were then averaged to return one effect size representing the estimated % increase or decrease vs. scrambled controls.

In *Figure 2—figure supplement 2d*, the data were 47 observations, one per larva. Each observation was the time in hours when the larva had its first sleep bout (status 2). Every larva slept at least once during the experiment, so there were no censored (status 1) observations. We calculated the hazard ratio with a Cox proportional-hazards model and the p-value with a likelihood-ratio test. The functions we used were implemented in the R package survival v3.4.0. The FramebyFrame R package automatically performs this survival analysis when generating sleep latency survival plots with the function ggSleepLatencyGrid(…).

In *Figure 5f*, *Figure 5—figure supplement 1f*, we compared values between groups (*sorl1* F0 knockouts vs. scrambled-injected) in each anatomical region using Welch's t-test. To test the null hypothesis that group assignment had no effect on values across regions, we used LME modelling implemented in the lmer function of the R package lme4 v1.1.31 (*Bates et al., 2015*). Taking total signal intensity as an example, the command to create the LME model was:

$$lmer \left( total\ signal\ intensity \sim group \right) + \left( 1 \mid larva \right) + \left( 1 \mid anatomical\ region \right)$$

The model provided the slope and its standard error reported in the figures. We then created a model without the fixed effect group, as:

$$lmer \left( total\ signal\ intensity \sim 1 \right) + \left( 1 \mid larva \right) + \left( 1 \mid anatomical\ region \right)$$

We compared this null model with the full model created above with a likelihood-ratio test, which provided the p-value reported in the figures.

## Software

Data analysis was performed in R v4.2.2 ran through RStudio 2023.06.0+421. Analysis of HCR on *sorl1* F0 knockouts used Jupyter notebooks written in Python 3 and a MATLAB script ran in MATLAB R2022b. Figures were prepared with Adobe Illustrator 2019. Videos were prepared with Adobe Premiere Pro 2020.

## Code availability

Source code of the FramebyFrame R package is available at github.com/francoiskroll/FramebyFrame (copy archived at *Kroll, 2025a*). The package can be installed directly into R. The GitHub repository includes installation instructions, tutorial, and documentation.

The ZOLTAR online app (francoiskroll.shinyapps.io/zoltar/) was written in R using the Shiny package. Source code is available at github.com/francoiskroll/ZOLTAR (copy archived at *Kroll, 2025b*).

The online app illustrating the simplified model of knockout by frameshift (francoiskroll.shinyapps.io/frameshiftmodel/) was written in R using the Shiny package. Source code is available github.com/francoiskroll/frameshiftShiny (*Kroll, 2022b*).

Other code used for analysis is available at github.com/francoiskroll/ZFAD (copy archived at *Kroll, 2025c*).

## Acknowledgements

We thank the members of the Rihel lab and other zebrafish groups at University College London (UCL) for helpful discussions. We thank Daniel J Stein and Douglas A Lauffenburger (MIT) for inspiration. We thank Alexandra Lubin and Katie-Jo Thorpe for the Illumina MiSeq runs; Chintan Trivedi for code used to design HCR probes and advice on analysis of the HCR images; Tom Ryan for help raising the *psen1*/*psen2* knockout larvae used for the Aβ measurements; Alizée Kastler for the *slc6a4b* HCR probes; Stephen Carter for his assessment of the *psen1*/*psen2* double knockout eye phenotype; Declan Lyons for the measurements of the 3D-printed plate's mesh. We thank all supporting staff at UCL including Fish Facility staff for fish care and husbandry. FK thanks Filippo Del Bene for the time spent in his lab working on the manuscript and Sophie Nunes Figueiredo and Camille Lejeune (PHENO-Zfish platform, Institut du Cerveau, Paris) for kindly giving access to their Zebrabox. Most of the behavioural small molecule dataset was generated by JR and David Prober in Alexander F Schier's

lab at Harvard University. FK was supported and funded by the Leonard Wolfson PhD Programme in Neurodegeneration. The work was also funded by a Wellcome Trust Investigator Award awarded to JR (217150/Z/19/Z).

## Additional information

### Funding

| Funder | Grant reference number | Author |
|---|---|---|
| Wellcome Trust | 10.35802/217150 | Jason Rihel |
| Leonard Wolfson PhD Programme in Neurodegeneration | | François Kroll |

The funders had no role in study design, data collection and interpretation, or the decision to submit the work for publication. For the purpose of Open Access, the authors have applied a CC BY public copyright license to any Author Accepted Manuscript version arising from this submission.

### Author contributions

François Kroll, Conceptualization, Resources, Data curation, Software, Formal analysis, Supervision, Validation, Investigation, Visualization, Methodology, Writing – original draft, Writing – review and editing; Joshua Donnelly, Formal analysis, Visualization, Methodology, Writing – review and editing; Güliz Gürel Özcan, Investigation, Methodology, Writing – review and editing; Eirinn Mackay, Resources, Methodology; Jason Rihel, Conceptualization, Data curation, Formal analysis, Supervision, Funding acquisition, Investigation, Methodology, Writing – original draft, Project administration, Writing – review and editing

### Author ORCIDs

François Kroll ⓘ https://orcid.org/0000-0001-9908-2648
Jason Rihel ⓘ https://orcid.org/0000-0003-4067-2066

### Ethics

Experimental procedures were in accordance with the Animals (Scientific Procedures) Act 1986 under Home Office project licences PA8D4D0E5 and PP6325955 awarded to Jason Rihel. Adult zebrafish were kept according to FELASA guidelines.

Reviewer #1 (Public review): https://doi.org/10.7554/eLife.96839.3.sa1
Reviewer #2 (Public review): https://doi.org/10.7554/eLife.96839.3.sa2
Author response https://doi.org/10.7554/eLife.96839.3.sa3

## Additional files

### Supplementary files

Supplementary file 1. Alzheimer's risk genes, zebrafish orthologues, crRNA sequences, and single-cell RNA-seq clusters. This file contains data related to prediction of Alzheimer's risk genes from genomic studies (Tab 1), zebrafish orthologues (Tab 2), HCR probes (Tab 3), sequences of crRNAs and MiSeq PCR primers (Tab 4), and scRNA-seq clusters (Tab 5).

MDAR checklist

### Data availability

All data generated or analysed during this study are included in the manuscript, supporting files, GitHub and Zenodo links provided therein. Data is available on Zenodo at https://doi.org/10.5281/zenodo.11673115. Source code of the FramebyFrame R package is available at https://github.com/francoiskroll/FramebyFrame (copy archived at *Kroll, 2025a*). Other code used for analysis is available at https://github.com/francoiskroll/ZFAD (copy archived at *Kroll, 2025c*).

The following previously published dataset was used:

| Author(s) | Year | Dataset title | Dataset URL | Database and Identifier |
|---|---|---|---|---|
| Raj B, Farrell JA, Liu J, El Kholtei J, Carte AN, Acedo JN, Du LY, McKenna A, Relić D, Leslie JM, Schier AF | 2020 | Emergence of neuronal diversity during vertebrate brain development | https://www.ncbi.nlm.nih.gov/geo/query/acc.cgi?acc=GSE158142 | NCBI Gene Expression Omnibus, GSE158142 |

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
