## [Editor Report · eLife Assessment]

This **important** manuscript sets out to identify sleep/arousal phenotypes in larval zebrafish carrying mutations in Alzheimer's disease (AD)-associated genes. The authors provide detailed phenotypic data for F0 knockouts of each of 7 AD-associated genes and then compare the resulting behavioral fingerprints to those obtained from a large-scale chemical screen to generate new hypotheses about underlying molecular mechanisms. The data presented are **solid**, although extensive interpretation of pharmacological screen data does not necessarily reflect the limited mechanistic data. Nonetheless, the authors address most reviewer concerns in their revised version, providing invaluable new analyses. Phenotypic characterization presented is comprehensive, and the authors develop a well-designed behavioral analysis pipeline that will provide considerable value for zebrafish neuroscientists.

---

## [Referee Report · Reviewer #1 (Public review)]

Summary:

In this study, Kroll et al. conduct an in-depth behavioral analysis of F0 knockouts of 4 genes associated with late-onset Alzheimer's Disease (AD), together with 3 genes associated with early-onset AD. Kroll and colleagues developed a web application (ZOLTAR) to compare sleep-associated traits between genetic mutants with those obtained from a panel of small molecules to promote identification of affected pathways and potential therapeutic interventions. The authors make a set of potentially important findings vis-à-vis the relationship between AD-associated genes and sleep. First, they find that loss-of-function in late-onset AD genes universally result in nighttime sleep loss, consistent with the well-supported hypothesis that sleep disruption contributes to Alzheimer's-related pathologies. psen-1, an early-onset associated AD gene, which the authors find is principally responsible for the generation of AB40 and AB42 in zebrafish, also shows a slight increase in activity at night and slight decreases in nighttime sleep. Conversely, psen-2 mutations increase daytime sleep, while appa/appb mutations have no impact on sleep. Finally, using ZOLTAR, the authors identify serotonin receptor activity as potentially disrupted in sorl1 mutants, while betamethasone is identified as a potential therapeutic to promote reversal of psen2 knockout-associated phenotypes.

This is a highly innovative and thorough study, yet a handful of key questions remain. First, are the nighttime sleep loss phenotypes observed in all knockouts for late-onset AD genes in the larval zebrafish a valid proxy for AD risk? Can 5-HT reuptake inhibitors reverse other AD-related pathologies in zebrafish? Can compounds be identified which have a common behavioral fingerprint across all or multiple AD risk genes? Do these modify sleep phenotypes? Finally, the authors propose but do not test the hypothesis that sorl1 might regulate localization/surface expression of 5-HT2 receptors. This could provide exciting / more convincing mechanistic support for the assertion that serotonin signaling is disrupted upon loss of AD-associated genes. Despite these important considerations, this study provides a valuable platform for high-throughput analysis of sleep phenotypes and correlation with small-molecule induced sleep phenotypes. The platform could also be expanded to facilitate comparison of other behavioral phenotypes, including stimulus-evoked behaviors. Moreover, the new analyses looking for pathways that might be co-regulated by AD risk genes and discussion of cholinergic signaling as a potentially meaningful target downstream of 5/7 knockouts are valuable.

Strengths:

- Provides a useful platform for comparison of sleep phenotypes across genotypes/drug manipulations.

- Presents convincing evidence that nighttime sleep is disrupted in mutants for multiple late-onset AD-related genes.

- Provides potential mechanistic insights for how AD-related genes might impact sleep and identifies a few drugs that modify their identified phenotypes.

Weaknesses:

- Exploration of potential mechanisms for serotonin disruption in sorl1 mutants is limited

- The pipeline developed is only used to examine sleep-related / spontaneous movement phenotypes. Stimulus-evoked behaviors are not examined.

---

## [Referee Report · Reviewer #2 (Public review)]

Summary:

This work delineates the larval zebrafish behavioral phenotypes caused by F0 knockout of several important genes that increase risk for Alzheimer's disease. Using behavioral pharmacology, comparing the behavioral fingerprint of previously assayed molecules to the newly generated knockout data, compounds were discovered that impacted larval movement in ways that suggest interaction with or recovery of disrupted mechanisms.

Strengths:

This is a well-written manuscript that uses newly developed analysis methods to present the findings in a clear, high-quality way. The addition of an extensive behavioral analysis pipeline is of value to the field of zebrafish neuroscience and will be particularly helpful for researchers who prefer the R programming language. Even the behavioral profiling of these AD risk genes, regardless of the pharmacology aspect, is an important contribution. The recovery of most behavioral parameters in the psen2 knockout with betamethasone, predicted by comparing fingerprints, is an exciting demonstration of the approach. The hypotheses generated by this work are important stepping stones to future studies uncovering the molecular basis of the proposed gene-drug interactions and discovering novel therapeutics to treat AD or co-occurring conditions such as sleep disturbance. Most concerns are sufficiently addressed in the revised manuscript or response to reviewers.

Weaknesses:

- The overarching concept of the work is that comparing behavioral fingerprints can align genes and molecules with similarly disrupted molecular pathways. While the recovery of the psen2 phenotypes by one molecule with the opposite phenotype is interesting, as are previous studies that show similar behaviorally-based recoveries, the underlying assumption that normalizing the larval movement normalizes the mechanism still lacks substantial support. While I agree with the authors detailed response that rescuing most behavioral parameters is a good indication that the underlying mechanism is normalized, I disagree that high-throughput larval behavior kinematics is a sufficient enough representation of most behavioral parameters to be indicative of molecular mechanism normalization. There are many instances of mutants with completely normal kinetics at baseline, but a behavioral difference that emerges during stimulation or in a new paradigm such as hunting. Without testing far more behavioral paradigms than are possible in the multi-well plate format, as well as possibly multiple life stages, I remain unconvinced that this approach will yield valuable therapeutic insights. I do agree that it can yield insight for future investigation, such as in the case of cntnap2a/cntnap2b and GABA receptor agonists, but even in that instance is it not clear that such an agonist would rescue abnormalities in a meaningful way. In the case of a disorder such as autism, the early locomotor phenotypes may be disconnected from the molecular mechanisms underlying later social deficits, and it is far more challenging to screen on juvenile behaviors that would be a more appropriate target for a behavior-first approach. The added experiment of testing fluvoxamine, a second SSRI, yielded very different behavioral responses to the SSRI citalopram, supporting my assertion that this approach and the disrupted underlying mechanisms are more complicated than suggested by the authors. I disagree that the connection between sorl1 and serotonin is strengthened by this experiment. The authors suggest that since the knockout larvae react differently than control siblings to both SSRIs, it indicates that serotonin is disrupted. There is no negative control included, where a pathway that is clearly not indicated to be important is pharmacologically manipulated. It is possible that the mutants would also behave differently compared to siblings when other pathways are perturbed. The authors acknowledge in the reviewers that they may not have identified the underlying molecular disruption in this mutant, but they did not substantially alter the Discussion section on this point. I agree with the authors that using a different wild-type strain in a different lab could lead to discrepancies, but these issues could have been experimentally mitigated or more clearly highlighted in the manuscript itself.

---

## [Author Response]

The following is the authors’ response to the original reviews.

**Public Reviews:**

**Reviewer #1 (Public Review):**
Summary:In this study, Kroll et al. conduct an in-depth behavioral analysis of F0 knockouts of 4 genes associated with late-onset Alzheimer's Disease (AD), together with 3 genes associated with early-onset AD. Kroll and colleagues developed a web application (ZOLTAR) to compare sleep-associated traits between genetic mutants with those obtained from a panel of small molecules to promote the identification of affected pathways and potential therapeutic interventions. The authors make a set of potentially important findings vis-à-vis the relationship between AD-associated genes and sleep. First, they find that loss-of-function in late-onset AD genes universally results in night-time sleep loss, consistent with the well supported hypothesis that sleep disruption contributes to Alzheimer's-related pathologies. psen-1, an early-onset associated AD gene, which the authors find is principally responsible for the generation of AB40 and AB42 in zebrafish, also shows a slight increase in activity at night and slight decreases in night-time sleep. Conversely, psen-2 mutations increase daytime sleep, while appa/appb mutations have no impact on sleep. Finally, using ZOLTAR, the authors identify serotonin receptor activity as potentially disrupted in sorl1 mutants, while betamethasone is identified as a potential therapeutic to promote reversal of psen2 knockout-associated phenotypes.This is a highly innovative and thorough study, yet a handful of key questions remain. First, are night-time sleep loss phenotypes observed in all knockouts for late-onset AD genes in the larval zebrafish a valid proxy for AD risk?

We cannot say, but it is an interesting question. We selected the four late-onset Alzheimer’s risk genes (*APOE*, *CD2AP*, *CLU*, *SORL1*) based on human genetics data and brain expression in zebrafish larvae, not based on their likelihood to modify sleep behaviour, which we could have tried by searching for overlaps with GWAS of sleep phenotypes, for example. Consequently, we find it remarkable that all four of these genes caused a night-time sleep phenotype when mutated. We also find it reassuring that knockout of *appa/appb* and *psen2* did not cause a night-time sleep phenotype, which largely excludes the possibility that the phenotype is a technical artefact (e.g. caused by the F0 knockout method) or a property of every gene expressed in the larval brain.

Having said that, it could still be a coincidence, rather than a special property of genes associated with late-onset AD. In addition to testing additional late-onset Alzheimer’s risk genes, the ideal way to answer this question would be to test in parallel a random set of genes expressed in the brain at this stage of development. From this random set, one could estimate the proportion of genes that cause a night-time sleep phenotype when mutated. One could then use that information to test whether late-onset Alzheimer’s risk genes are indeed enriched for genes that cause a night-time sleep phenotype when mutated.

For those mutants that cause night-time sleep disturbances, do these phenotypes share a common underlying pathway? e.g. Do 5-HT reuptake inhibitors promote sleep across all 4 late-onset genes in addition to psen1? Can 5-HT reuptake inhibitors reverse other AD-related pathologies in zebrafish? Can compounds be identified that have a common behavioral fingerprint across all or multiple AD risk genes? Do these modify sleep phenotypes?

To attempt to answer these questions, we used ZOLTAR to generate predictions for all the knockout behavioural fingerprints presented in the study, in the same way as for *sorl1* in Fig. 5 and Fig. 5–supplement 1. Here are the indications, targets, and KEGG pathways which are shared by the largest number of knockouts (Author response image 1):

– One **indication** is shared by 4/7 knockouts: “opioid dependence” (significant for *appa/appb*, *psen1*, *apoea/apoeb*, *cd2ap*).

– Four **targets** are shared by 4/7 knockouts: “strychnine-binding glycine receptor” (*psen1*, *apoea/apoeb*, *clu*, *sorl1*); “neuronal acetylcholine receptor beta-2” (*psen1*, *apoea/apoeb*, *cd2ap*, *clu*); thyroid peroxidase (*psen1*, *apoea/apoeb*, *cd2ap*, *clu*); carbonic anhydrase IV (*appa/appb*, *psen1*, *psen2*, *cd2ap*).

– Three **KEGG pathways** are shared by 5/7 knockouts: “cholinergic synapse” (*psen1*, *apoea/apoeb*, *cd2ap*, *clu*, *sorl1*); tyrosine metabolism (*psen2*, *apoea/apoeb*, *cd2ap*, *clu*, *sorl1*); and “nitrogen metabolism” (*appa/appb*, *psen1*, *psen2*, *apoea/apoeb*, *cd2ap*).

As reminder, we hypothesised that loss of Sorl1 affected serotonin signalling based on the following annotations being significant: indication “depression”, target “serotonin transporter”, and KEGG pathway “serotonergic synapse”. Indication “depression” is only significant for *sorl1* knockouts; target “serotonin transporter” is also significant for *appa/appb* and *psen2* knockouts; and KEGG pathway “serotonergic synapse” is also significant for *psen2* knockouts. ZOLTAR therefore does not predict serotonin signalling to be a major theme common to all mutants with a night-time sleep loss phenotype.

Particularly interesting is cholinergic signalling appearing in the most common targets and KEGG pathways. Acetylcholine signalling is a major theme in research on AD. For example, the first four drugs ever approved by the FDA to treat AD were acetylcholinesterase inhibitors, which increase acetylcholine signalling by preventing its breakdown by acetylcholinesterase. These drugs are generally considered only to treat symptoms and not modify disease course, but this view has been called into question (Munoz-Torrero, 2008; Relkin, 2007). If, as ZOLTAR suggests, mutations in several Alzheimer’s risk genes affect cholinergic signalling early in development, this would point to a potential causal role of cholinergic disruption in AD.

**Author response image 1. sa3fig1:** Common predictions from ZOLTAR for the seven Alzheimer’s risk genes tested. Predictions from ZOLTAR which are shared by multiple knockout behavioural fingerprints presented in the study. Only indications, targets, and KEGG pathways which are significant for at least three of the seven knockouts tested are shown, ranked from the annotations which are significant for the largest number of knockouts.

Finally, the web-based platform presented could be expanded to facilitate comparison of other behavioral phenotypes, including stimulus-evoked behaviors.

Yes, absolutely. The behavioural dataset we used (Rihel et al., 2010) did not measure other stimuli than day/night light transitions, but the “SauronX” platform and dataset (Myers-Turnbull et al., 2022) seems particularly well suited for this. To provide some context, we and collaborators have occasionally used the dataset by Rihel et al. (2010) to generate hypotheses or find candidate drugs that reverse a behavioural phenotype measured in the sleep/wake assay (Ashlin et al., 2018; Hoffman et al., 2016). The present work was the occasion to enable a wider and more intuitive use of this dataset through the ZOLTAR app, which has already proven successful. Future versions of ZOLTAR may seek to incorporate larger drug datasets using more types of measurements.

Finally, the authors propose but do not test the hypothesis that sorl1 might regulate localization/surface expression of 5-HT2 receptors. This could provide exciting / more convincing mechanistic support for the assertion that serotonin signaling is disrupted upon loss of AD-associated genes.

While working on the Author Response, we made some changes to the analysis ran by ZOLTAR to calculate enrichments (see Methods and github.com/francoiskroll/ZOLTAR, notes on v2). With the new version, 5-HT receptor type 2 is not a significantly enriched target for the *sorl1* knockout fingerprint but type 4 is. 5-HT receptor type 4 was also shown to interact with sorting nexin 27, a subunit of retromer, so is a promising candidate (Joubert et al., 2004). Antibodies against human 5-HT receptor type 2 and 4a exist; whether they would work in zebrafish remains to be tested. In our experience, the availability of antibodies suitable for immunohistochemistry in the zebrafish is a serious experimental roadblock.

Note, all the results presented in the “Version of Records” are from ZOLTAR v2.

Despite these important considerations, this study provides a valuable platform for high-throughput analysis of sleep phenotypes and correlation with small-molecule-induced sleep phenotypes.Strengths:- Provides a useful platform for comparison of sleep phenotypes across genotypes/drug manipulations.- Presents convincing evidence that night-time sleep is disrupted in mutants for multiple late-onset AD-related genes.- Provides potential mechanistic insights for how AD-related genes might impact sleep and identifies a few drugs that modify their identified phenotypesWeaknesses:- Exploration of potential mechanisms for serotonin disruption in sorl1 mutants is limited.- The pipeline developed can only be used to examine sleep-related / spontaneous movement phenotypes and stimulus-evoked behaviors are not examined.- Comparisons between mutants/exploration of commonly affected pathways are limited.

Thank you for these excellent suggestions, please see our answers above.

**Reviewer #2 (Public Review):**
Summary:This work delineates the larval zebrafish behavioral phenotypes caused by the F0 knockout of several important genes that increase the risk for Alzheimer's disease. Using behavioral pharmacology, comparing the behavioral fingerprint of previously assayed molecules to the newly generated knockout data, compounds were discovered that impacted larval movement in ways that suggest interaction with or recovery of disrupted mechanisms.Strengths:This is a well-written manuscript that uses newly developed analysis methods to present the findings in a clear, high-quality way. The addition of an extensive behavioral analysis pipeline is of value to the field of zebrafish neuroscience and will be particularly helpful for researchers who prefer the R programming language. Even the behavioral profiling of these AD risk genes, regardless of the pharmacology aspect, is an important contribution. The recovery of most behavioral parameters in the psen2 knockout with betamethasone, predicted by comparing fingerprints, is an exciting demonstration of the approach. The hypotheses generated by this work are important stepping stones to future studies uncovering the molecular basis of the proposed gene-drug interactions and discovering novel therapeutics to treat AD or co-occurring conditions such as sleep disturbance.Weaknesses:- The overarching concept of the work is that comparing behavioral fingerprints can align genes and molecules with similarly disrupted molecular pathways. While the recovery of the psen2 phenotypes by one molecule with the opposite phenotype is interesting, as are previous studies that show similar behaviorally-based recoveries, the underlying assumption that normalizing the larval movement normalizes the mechanism still lacks substantial support. There are many ways that a reduction in movement bouts could be returned to baseline that are unrelated to the root cause of the genetically driven phenotype. An ideal experiment would be to thoroughly characterize a mutant, such as by identifying a missing population of neurons, and use this approach to find a small molecule that rescues both behavior and the cellular phenotype. If the connection to serotonin in the sorl1 was more complete, for example, the overarching idea would be more compelling.

Thank you for this cogent criticism.

On the first point, we were careful not to claim that betamethasone normalises the molecular/cellular mechanism that causes the *psen2* behavioural phenotype. Having said that, yes, to a certain extent that would be the hope of the approach. As you say, every compound which normalises the behavioural fingerprint will not normalise the underlying mechanism, but the opposite seems true: every compound that normalises the underlying mechanism should also normalise the behavioural fingerprint. We think this logic makes the “behaviour-first” approach innovative and interesting. The logic is to discover compounds that normalise the behavioural phenotype first, only subsequently test whether they also normalise the molecular mechanism, akin to testing first whether a drug resolves the symptoms before testing whether it actually modifies disease course. While in practice testing thousands of drugs in sufficient sample sizes and replicates on a mutant line is challenging, the dataset queried through ZOLTAR provides a potential shortcut by shortlisting *in silico* compounds that have the opposite effect on behaviour.

You mention a “reduction in movement bouts” but note here that the number of behavioural parameters tested is key to our argument. To take the two extremes, say the only behavioural parameter we measured in *psen2* knockout larvae was time active during the day, then, yes, any stimulant used at the right concentration could probably normalise the phenotype. In this situation, claiming that the stimulant is likely to also normalise the underlying mechanism, or even that it is a genuine “phenotypic rescue”, would not be convincing. Conversely, say we were measuring thousands of behavioural parameters under various stimuli, such as swimming speed, position in the well, bout usage, tail movements, and eye angles, it seems almost impossible for a compound to rescue most parameters without also normalising the underlying mechanism. The present approach is somewhere inbetween: ZOLTAR uses six behavioural parameters for prediction (e.g. Fig 6a), but all 17 parameters calculated by FramebyFrame can be used to assess rescue during a subsequent experiment (Fig. 6c). For both, splitting each parameter in day and night increases the resolution of the approach, which partly answers your criticism. For example, betamethasone rescued the day-time hypoactivity without causing night-time hyperactivity, so we are not making the “straw man argument” explained above of using any broad stimulant to rescue the hypoactivity phenotype.

Furthermore, for diseases where the behavioural defect is the primary concern, such as autism or bipolar disorder, perhaps this behaviour-first approach is all that is needed, and whether or not the compound precisely rescues the underlying mechanism is somewhat secondary. The use of lithium to prevent manic episodes in bipolar disorder is a good example. It was initially tested because mania was thought to be caused by excess uric acid and lithium can dissolve uric acid (Mitchell and Hadzi-Pavlovic, 2000). The theory is now discredited, but lithium continues to be used without a precise understanding of its mode of action. In this example, behavioural rescue alone, assuming the secondary effects are tolerable, is sufficient to be beneficial to patients, and whether it modulates the correct causal pathway is secondary.

On the second point, we agree that testing first ZOLTAR on a mutant for which we have a fairly good understanding of the mechanism causing the behavioural phenotype could have been a productive approach. Note, however, that examples already exist in the literature (Ashlin et al., 2018; Hoffman et al., 2016). The example from Hoffman et al. (2016) is especially convincing. Drugs generating behavioural fingerprints that positively correlate with the *cntnap2a/cntnap2b* double knockout fingerprint were enriched with NMDA and GABA receptor antagonists. In experiments analogous to our citalopram and fluvoxamine treatments (Fig. 5c,d and Fig. 5–supplement 1c,d), *cntnap2a/cntnap2b* knockout larvae were overly sensitive to the NMDA receptor antagonist MK-801 and the GABAA receptor antagonist pentylenetetrazol (PTZ). Among other drugs tested, zolpidem, a GABAA receptor agonist, caused opposite effects on wild-type and *cntnap2a/cntnap2b* knockout larvae. Knockout larvae were found to have fewer GABAergic neurons in the forebrain. While these studies did not use precisely the same analysis that ZOLTAR runs, they used the same rationale and behavioural dataset to make these predictions (Rihel et al., 2010), which shows that approaches like ZOLTAR can point to causal processes.

On your last point, we hope our experiment testing fluvoxamine, another selective serotonin reuptake inhibitor (SSRI), makes the connection between Sorl1 and serotonin signalling more convincing.

- The behavioral difference between the sorl1 KO and scrambled at the higher dose of the citalopram is based on a small number of animals. The KO Euclidean distance measure is also more spread out than for the other datasets, and it looks like only five or so fish are driving the group difference. It also appears as though the numbers were also from two injection series. While there is nothing obviously wrong with the data, I would feel more comfortable if such a strong statement of a result from a relatively subtle phenotype were backed up by a higher N or a stable line. It is not impossible that the observed difference is an experimental fluke. If something obvious had emerged through the HCR, that would have also supported the conclusions. As it stands, if no more experiments are done to bolster the claim, the confidence in the strength of the link to serotonin should be reduced (possibly putting the entire section in the supplement and modifying the discussion). The discussion section about serotonin and AD is interesting, but I think that it is excessive without additional evidence.

We mostly agree with this criticism. One could interpret the larger spread of the data for *sorl1* KO larvae treated with 10 µM citalopram as evidence that the knockout larvae do indeed react differently to the drug at this dose, regardless of being driven by a subset of the animals. The result indeed does not survive removing the top 5 (p = 0.87) or top 3 (p = 0.18) *sorl1* KO + 10 µM larvae, but this amounts to excluding 20 (3/14) or 35 (5/14) % of the datapoints as potential outliers, which is unreasonable. In fact, excluding the top 5 *sorl1* KO + 10 µM is equivalent to calling any datapoint with z-score > 0.2 an outlier (z-scores of the top 5 datapoints are 0.2–1.8). Applying consistently the same criterion to the scrambled + 10 µM group would remove the top 6 datapoints (z-scores = 0.5–3.9). Comparing the resulting two distributions again gives the *sorl1* KO + 10 µM distribution as significantly higher (p = 0.0015). We would also mention that Euclidean distance, as a summary metric for distance between behavioural fingerprints, has limitations. For example, the measure will be more sensitive to changes in some parameters but not others, depending on how much room there is for a given parameter to change. We included this metric to lend support to the observation one can draw from the fingerprint plot (Fig. 5c) that *sorl1* mutants respond in an exaggerated way to citalopram across many parameters, while being agnostic to which parameter might matter most.

Given that the HCR did not reveal anything striking, we agree with you that too much of our argument relied on this result being robust. As you and Reviewer #3 suggested, we repeated this experiment with a different SSRI, fluvoxamine (Fig. 5–supplement 1). We cannot readily explain why the result was opposite to what we found with citalopram, but in both cases *sorl1* knockout larvae reacted differently than their control siblings, which adds an argument to our claim that ZOLTAR correctly predicted serotonin signalling as a disrupted pathway from the behavioural fingerprint. Accordingly, we mostly kept the Discussion on Sorl1 the same, although we concede that we may not have identified the molecular mechanism.

- The authors suggest two hypotheses for the behavioral difference between the sorl1 KO and scrambled at the higher dose of the citalopram. While the first is tested, and found to not be supported, the second is not tested at all ("Ruling out the first hypothesis, sorl1 knockouts may react excessively to a given spike in serotonin." and "Second, sorl1 knockouts may be overly sensitive to serotonin itself because post-synaptic neurons have higher levels of serotonin receptors."). Assuming that the finding is robust, there are probably other reasons why the mutants could have a different sensitivity to this molecule. However, if this particular one is going to be mentioned, it is surprising that it was not tested alongside the first hypothesis. This work could proceed without a complete explanation, but additional discussion of the possibilities would be helpful or why the second hypothesis was not tested.

There are no strong scientific reasons why this hypothesis was not tested. The lead author (F Kroll) moved to a different lab and country so the project was finalised at that time. We do not plan on testing this hypothesis at this stage. However, we adapted the wording to make it clear this is one possible alternative hypothesis which could be tested in the future. The small differences found by HCR are actually more in line with the new results from the fluvoxamine experiment, so it may also be that both hypotheses (pre-synaptic neurons releasing less serotonin when reuptake is blocked; or post-synaptic neurons being less sensitive) contribute. The fluvoxamine experiment was performed in a different lab (ICM, Paris; all other experiments were done in UCL, London) in a different wild-type strain (TL in ICM, AB x Tup LF in UCL), which complicates how one interprets this discrepancy.

- The authors claim that "all four genes produced a fairly consistent phenotype at night". While it is interesting that this result arose in the different lines, the second clutch for some genes did not replicate as well as others. I think the findings are compelling, regardless, but the sometimes missing replicability should be discussed. I wonder if the F0 strategy adds noise to the results and if clean null lines would yield stronger phenotypes. Please discuss this possibility, or others, in regard to the variability in some phenotypes.

For the first part of this point, please see below our answer to Reviewer #3, point (2) c.

Regarding the F0 strategy potentially adding variability, it is an interesting question which we tested in a larger dataset of behavioural recordings from F0 and stable knockouts for the same genes (unpublished). In summary, the F0 knockout method does not increase clutch-to-clutch or larva-to-larva variability in the assay. F0 knockout experiments found many more significant parameters and larger effect sizes than stable knockout experiments, but this difference could largely be explained by the larger sample sizes of F0 knockout experiments. In fact, larger sample sizes within individual clutches appears to be a major advantage of the F0 knockout approach over in-cross of heterozygous knockout animals as it increases sensitivity of the assay without causing substantial variability. We plan to report in more detail on this analysis in a separate paper as we think it would dilute the focus of the present work.

- In this work, the knockout of appa/appb is included. While APP is a well-known risk gene, there is no clear justification for making a knockout model. It is well known that the upregulation of app is the driver of Alzheimer's, not downregulation. The authors even indicate an expectation that it could be similar to the other knockouts ("Moreover, the behavioural phenotypes of appa/appb and psen1 knockout larvae had little overlap while they presumably both resulted in the loss of Aβ." and "Comparing with early-onset genes, psen1 knockouts had similar night-time phenotypes, but loss of psen2 or appa/appb had no effect on night-time sleep."). There is no reason to expect similarity between appa/appb and psen1/2. I understand that the app knockouts could unveil interesting early neurodevelopmental roles, but the manuscript needs to be clarified that any findings could be the opposite of expectation in AD.

On “there is no reason to expect similarity […]”, we disagree. Knockout of *appa/appb* and knockout of *psen1* will both result in loss of Aβ (*appa/appb* encode Aβ and *psen1* cleaves Appa/Appb to release Aβ, cf. Fig. 3e). Consequently, a phenotype caused by the loss of Aβ, or possibly other Appa/Appb cleavage products, should logically be found in both *appa/appb* and *psen1* knockouts.

On “it is well known that the upregulation of *APP* is the driver of Alzheimer’s, not downregulation”; we of course agree. Among others, the examples of Down syndrome, *APP* duplication (Sleegers et al., 2006), or mouse models overexpressing human *APP* show definitely that overexpression of *APP* is sufficient to cause AD. Having said that, we would not be so quick in dismissing *APP* knockout as potentially relevant to understanding of AD.

Loss of soluble Aβ due to aggregation could contribute to pathology (Espay et al., 2023). Without getting too much into this intricate debate, links between levels of Aβ and risk of disease are often counter-intuitive too. For example, out of 138 PSEN1 mutations screened in vitro, 104 reduced total Aβ production and 11 even seemingly abolished the production of both Aβ40 and Aβ42 (Sun et al., 2017). In short, loss of soluble Aβ occurs in both AD and in our *appa/appb* knockout larvae.

We added a sentence in Results (section *psen2 knockouts […]*) to briefly justify our *appa/appb* knockout approach. To be clear, we do not want to imply, for example, that the absence of a night-time sleep phenotype for *appa/appb* is contradictory to the body of literature showing links between Aβ and sleep, including in zebrafish (Özcan et al., 2020). As you say, our experiment tested loss of App, including Aβ, while the literature typically reports on overexpression of *APP*, as in *APP/PSEN1*-overexpressing mice (Jagirdar et al., 2021).

**Reviewer #3 (Public Review):**
In this manuscript by Kroll and colleagues, the authors describe combining behavioral pharmacology with sleep profiling to predict disease and potential treatment pathways at play in AD. AD is used here as a case study, but the approaches detailed can be used for other genetic screens related to normal or pathological states for which sleep/arousal is relevant. The data are for the most part convincing, although generally the phenotypes are relatively small and there are no major new mechanistic insights. Nonetheless, the approaches are certainly of broad interest and the data are comprehensive and detailed. A notable weakness is the introduction, which overly generalizes numerous concepts and fails to provide the necessary background to set the stage for the data.Major points(1) The authors should spend more time explaining what they see as the meaning of the large number of behavioral parameters assayed and specifically what they tell readers about the biology of the animal. Many are hard to understand--e.g. a "slope" parameter.

We agree that some parameters do not tell something intuitive about the biology of the animal. It would be easy to speculate. For example, the “activity slope” parameter may indicate how quickly the animal becomes tired over the course of the day. On the other hand, fractal dimension describes the “roughness/smoothness” of the larva’s activity trace (Fig. 2–supplement 1a); but it is not obvious how to translate this into information about the physiology of the animal. We do not see this as an issue though. While some parameters do provide intuitive information about the animal’s behaviour (e.g. sleep duration or sunset startle as a measure of startle response), the benefit of having a large number of behavioural parameters is to compare behavioural fingerprints and assess rescue of the behavioural phenotype by small molecules (Fig. 6c). For this purpose, the more parameters the better. The “MoSeq” approach from Wiltschko et al., 2020 is a good example from literature that inspired our own Fig. 6c. While some of the “behavioural syllables” may be intuitive (e.g. running or grooming), it is probably pointless to try to explain the ‘meaning’ of the “small left turn in place with head motion” syllable (Wiltschko et al., 2020). Nonetheless, this syllable was useful to assess whether a drug specifically treats the behavioural phenotype under study without causing too many side effects. Unfortunately, ZOLTAR has to reduce the FramebyFrame fingerprint (17 parameters) to just six parameters to compare it to the behavioural dataset from Rihel et al., 2010, but here, more parameters would almost certainly translate into better predictions too, regardless of their intuitiveness.

It is true however that we did not give much information on how some of the less intuitive parameters, such as activity slope or fractal dimension, are calculated or what they describe about the dataset (e.g. roughness/smoothness for fractal dimension). We added a few sentences in the legend of Fig. 2–supplement 1.

(2) Because in the end the authors did not screen that many lines, it would increase confidence in the phenotypes to provide more validation of KO specificity. Some suggestions include:a. The authors cite a *psen1* and *psen2* germline mutant lines. Can these be tested in the FramebyFrame R analysis? Do they phenocopy F0 KO larvae?

We unfortunately do not have those lines. We investigated the availability of importing a *psen2* knockout line from abroad, but the process of shipping live animals is becoming more and more cost and time prohibitive. However, we observed the same pigmentation phenotype for *psen2* knockouts as reported by Jiang et al., 2018, which is at least a partial confirmation of phenocopying a loss of function stable mutant.

b. *psen2* KO is one of the larger centerpieces of the paper. The authors should present more compelling evidence that animals are truly functionally null. Without this, how do we interpret their phenotypes?

We disagree that there should be significant doubt about these mutants being truly functionally null, given the high mutation rate and presence of the expected pigmentation phenotype (Jiang et al., 2018, Fig. 3f and Fig. 3–supplement 3a). The *psen2* F0 knockouts were virtually 100% mutated at three exons across the gene (mutation rates were locus 1: 100 ± 0%; locus 2: 99.99 ± 0.06%; locus 3: 99.85 ± 0.24%). Additionally, two of the three mutated exons had particularly high rates of frameshift mutations (locus 1: 97 ± 5%; locus 2: 88 ± 17% frameshift mutation rate). It is virtually impossible that a functional protein is translated given this burden of frameshift mutations. Phenotypically, in addition to the pigmentation defect, double *psen1/psen2* F0 knockout larvae had curved tails, the same phenotype as caused by a high dose of the γ-secretase inhibitor DAPT (Yang et al., 2008). These double F0 knockouts were lethal, while knockout of *psen1* or *psen2* alone did not cause obvious morphological defects. Evidently, most larvae must have been *psen2* null mutants in this experiment, otherwise functional Psen2 would have prevented early lethality.

Translation of zebrafish *psen2* can start at downstream start codons if the first exon has a frameshift mutation, generating a seemingly functional Psen2 missing the N-terminus (Jiang et al., 2020). Zebrafish homozygous for this early frameshift mutation had normal pigmentation, showing it is a reliable marker of Psen2 function even when it is mutated. This mechanism is not a concern here as the alternative start codons are still upstream of two of the three mutated exons (the alternative start codons discovered by Jiang et al., 2020 are in exon 2 and 3, but we targeted exon 3, exon 4, and exon 6).

We understand that the zebrafish community may be cautious about F0 phenotyping compared to stably generated mutants. As mentioned to Reviewer #2, we are planning to assemble a paper that expressly compares behavioural phenotypes measured in F0 vs. stable mutants to allay some of these concerns. Our current manuscript, which combines CRISPR-Cas9 rapid F0 screening with *in silico* pharmacological predictions, inevitability represents a first step in characterizing the functions of these genes.

c. Related to the above, for *cd2ap* and *sorl1* KO, some of the effect sizes seem to be driven by one clutch and not the other. In other words, great clutch-to-clutch variability. Should the authors increase the number of clutches assayed?

Correct, there is substantial clutch-to-clutch variability in this behavioural assay. This is not specific to our experiments. Even within the same strain, wild-type larvae from different clutches (i.e. non-siblings) behave differently (Joo et al., 2021). This is why it is essential to compare behavioural phenotypes within individual clutches (i.e. from a single pair of parents, one male and one female), as we explain in Methods (section *Behavioural video-tracking*) and in the documentation of the FramebyFrame package. We often see two different experimental designs in literature: comparing non-sibling wild-type and mutant larvae, or pooling different clutches which include all genotypes (e.g. pooling multiple clutches from heterozygous in-crosses or pooling wild-type clutches before injecting them). The first experimental design causes false positive findings (Joo et al., 2021), as the clutch-to-clutch variability we and others observe gets interpreted as a behavioural phenotype. The second experimental design should not cause false positives but likely decreases the sensitivity of the assay by increasing the spread within genotypes. In both cases, the clutch-to-clutch variability is hidden, either by interpreting it as a phenotype (first case) or by adding it to animal-to-animal variability (second case). Our experimental design is technically more challenging as it requires obtaining large clutches from unique pairs of parents. However, this approach is better as it clearly separates the different sources of variability (clutch-to-clutch or animal-to-animal). As for every experiment, yes, a larger number of replicates would be better, but we do not plan to assay additional clutches at this time. Our work heavily focuses on the *sorl1* and *psen2* knockout behavioural phenotypes. The key aspects of these phenotypes were effectively tested in four experiments (five to six clutches) as *sorl1* knockout larvae were also tracked in the citalopram and fluvoxamine experiments (Fig. 5 and Fig. 5–supplement 1), and *psen2* knockout larvae were also tracked in the small molecule rescue experiment (Fig. 6 and Fig. 6–supplement 1).

The *psen2* behavioural phenotype replicated well across the six clutches tested (pairwise cosine similarities: 0.62 ± 0.15; Author response image 2a). 5/6 clutches were less active and initiating more sleep bouts during the day, as we claimed in Fig. 3.

In the citalopram experiment, the H_2_O-treated *sorl1* knockout fingerprint replicated fairly well the baseline recordings in Fig. 4, despite the smaller sample size (cos = 0.30 and 0.78; Author response image 2b, see “KO Fig. 5”). 5/6 of the significant parameters presented in Fig. 4–supplement 4 moved in the same direction, and knockout larvae were also hypoactive during the day but hyperactive at night. Note that two clutches were tracked on the same 96-well plate in this experiment. We calculated each larva’s z-score using the average of its control siblings, then we averaged all the z-scores to generate the fingerprint. The H_2_O treated *sorl1* knockout clutch from the fluvoxamine experiment did not replicate well the baseline recordings (cos = 0.08 and 0.11; Author response image 2b, see “KO Fig. 5–suppl. 1”). Knockout larvae were hypoactive during the day as expected, but behaviour at night was not as robustly affected. As mentioned above, knockouts were made in a different genetic background (TL, instead of AB x Tup LF used for all other experiments), which could explain the discrepancy.

We also took the opportunity to check whether our SSRI treatments replicated well the data from Rihel et al., 2010. For both citalopram (n = 3 fingerprints in the database) and fluvoxamine (n = 4 fingerprints in the database), replication was excellent (cos ≥ 0.67 for all comparisons of a fingerprint from this study vs. a fingerprint from Rihel et al. 2010; Author response image 2c,d). Note that the scrambled + 10 µM citalopram and + 10 µM fluvoxamine fingerprints correlate extremely well (cos = 0.92; can be seen in Author response image 2c,d), which was predicted by the small molecule screen dataset.

**Author response image 2. sa3fig2:** Replication of *psen2* and *sorl1* F0 knockout fingerprints and SSRI treatments from Rihel et al., 2010a. (**a**), (left) Every *psen2* F0 knockout behavioural fingerprint generated in this study. Each dot represents the mean deviation from the same-clutch scrambled-injected mean for that parameter (z-score, mean ± SEM). From the experiments in Fig. 6, presented is the *psen2* F0 knockout + H_2_O fingerprints. The fingerprints in grey (“not shown”) are from a preliminary drug treatment experiment we did not include in the final study. These fingerprints are from *psen2* F0 knockout larvae treated with 0.2% DMSO, normalised to scrambled-injected siblings also treated with 0.2% DMSO. (right) Pairwise cosine similarities (−1.0–1.0) for the fingerprints presented. (**b**) Every *sorl1* F0 knockout behavioural fingerprint, as in (a). (**c**) The scrambled-injected + citalopram (10 µM) fingerprints (grey) in comparison to the citalopram (10–15 µM) fingerprints from the Rihel et al., 2010 database (green). (**d**) The scrambled-injected + fluvoxamine (10 µM) fingerprint (grey) in comparison to the fluvoxamine fingerprints from the Rihel et al., 2010 database (pink). In (c) and (d), the scrambled-injected fingerprints are from the experiments in Fig. 5 and Fig. 5–suppl. 1, but were converted here into the behavioural parameters used by Rihel et al., 2010 for comparison. Parameters: 1, average activity (sec active/min); 2, average waking activity (sec active/min, excluding inactive minutes); 3, total sleep (hr); 4, number of sleep bouts; 5, sleep bout length (min); 6, sleep latency (min until first sleep bout).

(3) The authors make the point that most of the AD risk genes are expressed in fish during development. Is there public data to comment on whether the genes of interest are expressed in mature/old fish as well? Just because the genes are expressed early does not at all mean that early-life dysfunction is related to future AD (though this could be the case, of course). Genes with exclusive developmental expression would be strong candidates for such an early-life role, however. I presume the case is made because sleep studies are mainly done in juvenile fish, but I think it is really a pretty minor point and such a strong claim does not even need to be made.

This is a fair criticism but we do not make this claim (“early-life dysfunction is related to future AD”) from expression alone. The reviewer is probably referring to the following quote:

“[…] most of these were expressed in the brain of 5–6-dpf zebrafish larvae, suggesting they play a role in early brain development or function,” which does not mention future risk of AD. We do *suggest* that these genes have a function in development. After all, every gene that plays a role in brain development must be expressed during development, so this wording seemed reasonable. Nevertheless, we adapted the wording to address this point and Reviewer #2’s complaint below. As noted, the primary goal was to check that the genes we selected were indeed expressed in zebrafish larvae before performing knockout experiments. Our discussion does raise the hypothesis that mutations in Alzheimer’s risk genes impact brain development and sleep early in life, but this argument primarily relies on our observation that knockout of late-onset Alzheimer’s risk genes causes sleep phenotypes in 7-day old zebrafish larvae and from previous work showing brain structural differences in children at high genetic risk of AD (Dean et al., 2014; Quiroz et al., 2015), not solely on gene expression early in life.

Please also see our answer to a similar point raised by Reviewer #2 below (cf. Author response image 7).

(4) A common quandary with defining sleep behaviorally is how to rectify sleep and activity changes that influence one another. With *psen2* KOs, the authors describe reduced activity and increased sleep during the day. But how do we know if the reduced activity drives increased behavioral quiescence that is incorrectly defined as sleep? In instances where sleep is increased but activity during periods during wake are normal or elevated, this is not an issue. But here, the animals might very well be unhealthy, and less active, so naturally they stop moving more for prolonged periods, but the main conclusion is not sleep per se. This is an area where more experiments should be added if the authors do not wish to change/temper the conclusions they draw. Are *psen2* KOs responsive to startling stimuli like controls when awake? Do they respond normally when quiescent? Great care must be taken in all models using inactivity as a proxy for sleep, and it can harm the field when there is no acknowledgment that overall health/activity changes could be a confound. Particularly worrisome is the betamethasone data in Figure 6, where activity and sleep are once again coordinately modified by the drug.

This is a fair criticism. We agree it is a concern, especially in the case of *psen2* as we claim that *day-time* sleep is increased while zebrafish are diurnal. We do not rely heavily on the day-time inactivity being sleep (the ZOLTAR predictions or the small molecule rescue do not change whether the parameter is called sleep or inactivity), but our choice of labelling can fairly be challenged.

To address “are *psen2* KO responsive to startling stimuli like controls when awake/when quiescent”, we looked at the larvae’s behaviour immediately after lights abruptly switched on in the mornings. Almost every larva, regardless of genotype, responded strongly to every lights-*off* transition during the experiment. Instead, we chose the lights-*on* transition for this analysis because it is a weaker startling stimulus for the larvae than the lights-off transition (Fig. 3–supplement 3), potentially exposing differences between genotypes or behavioural states (quiescent or awake). We defined a larva as having reacted to the lights switching on if it made a swimming bout during the second (25 frames) after the lights-on transition. Across two clutches and two lights-on transitions, an average of 65% (range 52–73%) of all larvae reacted to the stimulus. *psen2* knockout larvae were similarly likely, if not *more* likely, to respond (in average 69% responded, range 60–76%) than controls (60% average, range 44– 75%). When the lights switched on, about half of the larvae (39–51%) would have been classified as asleep according to the one-minute inactivity definition (i.e. the larva did not move in the minute preceding the lights transition). This allowed us to also compare behavioural states, as suggested by the reviewer. For three of the four light transitions, larvae which were awake when lights switched on were more likely to react than asleep larvae, but this difference was not striking (overall, awake larvae were only 1.1× more likely to react; Author response image 3). Awake *psen2* knockout larvae were 1.1× (range 1.04–1.11×) more likely to react than awake control larvae, so, yes, *psen2* knockout larvae respond normally when awake. Asleep *psen2* knockout larvae were 1.4× (range 0.63–2.19×) more likely to react than asleep control larvae, so *psen2* knockouts are also more or equally likely to react than control larvae when asleep. In summary, the overall health of *psen2* knockouts did not seem to be a significant confound in the experiment. As the reviewer suggested, if *psen2* knockout larvae were seriously unhealthy, they would not be as responsive as control larvae to a startling stimulus.

**Author response image 3. sa3fig3:** *psen2* F0 knockouts react normally to lights switching on, indicating they are largely healthy. At each lights-on transition (9 AM), each larva was categorised as awake if it had moved in the preceding one minute or asleep if it had been inactive for at least one minute. Darker tiles represent larvae which performed a swimming bout during the second following lights-on; lighter tiles represent larvae which did not move during that second. The total count of each waffle plot was normalised to 25 so plots can be compared to each other. The real count is indicated in the corner of each plot. Data is from the baseline *psen2* knockout trackings presented in Fig. 3 and Fig. 3–suppl. 2.

Next, we compared inactive period durations during the day between *psen2* and control larvae. If *psen2* knockout larvae indeed sleep more during the day compared to controls, we may predict inactive periods longer than one minute to increase disproportionately compared to the increase in shorter inactive periods. This broadly appeared to be the case, especially for one of the two clutches (Author response image 4). In clutch 1, inactive periods lasting 1–60 sec were equally frequent in both *psen2* and control larvae (fold change 1.0× during both days), while inactive periods lasting 1–2 min were 1.5× (day 1) and 2.5× (day 2) more frequent in *psen2* larvae compared to control larvae. In clutch 2, 1–60 sec inactive periods were also equally frequent in both *psen2* and control larvae, while inactive periods lasting 1–2 min were 3.4× (day 1) and 1.5× (day 2) more frequent in *psen2* larvae compared to control larvae. Therefore, *psen2* knockouts disproportionately increased the frequency of inactive periods longer than one minute, suggesting they genuinely slept more during the day.

**Author response image 4. sa3fig4:** *psen2* F0 knockouts increased preferentially the frequency of longer inactive bouts. For each day and clutch, we calculated the mean distribution of inactive bout lengths across larvae of same genotype (*psen2* F0 knockout or scrambled-injected), then compared the frequency of inactive bouts of different lengths between the two genotypes. For example, in clutch 1 during day 2, 0.01% of the average scrambled-injected larva’s inactive bouts lasted 111–120 seconds (X axis 120 sec) while 0.05% of the average *psen2* F0 knockout larva lasted this long, so the fold change was 5×. Inactive bouts lasting <1 sec were excluded from the analysis. In clutch 2, day 1 plot, two datapoints fall outside the Y axis limit: 140 sec, Y = 32×; 170 sec, Y = 16×. Data is from the baseline *psen2* knockout trackings presented in Fig. 3 and Fig. 3–suppl. 2.

Ultimately, this criticism seems challenging to definitely address experimentally. A possible approach could be to use a closed-loop system which, after one minute of inactivity, triggers a stimulus that is sufficient to startle an awake larva but not an asleep larva. If *psen2* knockout larvae indeed sleep more during the day, the stimulus should usually not be sufficient to startle them. Nevertheless, we believe the two analyses presented here are consistent with *psen2* knockout larvae genuinely sleeping more during the day, so we decided to keep this label. We agree with the reviewer that the one-minute inactivity definition has limitations, especially for day-time inactivity.

(5) The conclusions for the serotonin section are overstated. Behavioural pharmacology purports to predict a signaling pathway disrupted with *sorl1* KO. But is it not just possible that the drug acts in parallel to the true disrupted pathway in these fish? There is no direct evidence for serotonin dysfunction - that conclusion is based on response to the drug. Moreover, it is just one drug - is the same phenotype present with another SSRI? Likewise, language should be toned down in the discussion, as this hypothesis is not "confirmed" by the results (consider "supported"). The lack of measured serotonin differences further raises concern that this is not the true pathway. This is another major point that deserves further experimental evidence, because without it, the entire approach (behavioral pharm screen) seems more shaky as a way to identify mechanisms. There are any number of testable hypotheses to pursue such as (a) Using transient transgenesis to visualize 5HT neuron morphology (is development perturbed: cell number, neurite morphology, synapse formation); (b) Using transgenic Ca reporters to assay 5HT neuron activity.

Regarding the comment, “is it not just possible that the drug acts in parallel to the true disrupted pathway”, we think no, assuming we understand correctly the question. Key to our argument is the fact that *sorl1* knockout larvae react differently to the drug(s) than control larvae. As an example, take night-time sleep bout length, which was not affected by knockout of *sorl1* (Fig. 4–supplement 4). For the sake of the argument, say only dopamine signalling (the “true disrupted pathway”) was affected in *sorl1* knockouts and that serotonin signalling was intact. Assuming that citalopram specifically alters serotonin signalling, then treatment should cause the same increase in sleep bout length in both knockouts and controls as serotonin signalling is intact in both. This is not what we see, however. Citalopram caused a *greater* increase in sleep bout length in *sorl1* knockouts than in scrambled-injected larvae. In other words, the effect is non-additive, in the sense that citalopram did not add the same number of z-scores to *sorl1* knockouts or controls. We think this shows that serotonin signalling is somehow different in *sorl1* knockouts. Nonetheless, we concede that the experiment does not necessarily say much about the importance of the serotonin disruption caused by loss of Sorl1. It could be, for example, that the most salient consequence of loss of Sorl1 is cholinergic disruption (see reply to Reviewer #1 above) and that serotonin signalling is a minor theme.

Furthermore, we agree with the reviewer and Reviewer #2 that the conclusions were overly confident. As suggested, we decided to repeat this experiment with another SSRI, fluvoxamine. Please find the results of this experiment in Fig. 5–supplement 1. The suggestions to further test the serotonin system in the *sorl1* knockouts are excellent as well, however we do not plan to pursue them at this stage.

**Recommendations for the authors:**

**Reviewer #1 (Recommendations For The Authors):**
Major Comments:- Data are presented in a variety of different ways, occasionally making comparisons across figures difficult. Perhaps at a minimum, behavioral fingerprints as in Figure 3 - Supplementary Figure 1 should be presented for all mutants in the main figures.

We like this suggestion! Thank you. We brought the behavioural fingerprints figure (previously Fig. 4–supplement 5) as main Fig. 4, and put the figure focused on the *sorl1* knockout behavioural phenotype in supplementary, with the other gene-by-gene figures.

- It is not clear why some data were selected for supplemental rather than main figures. In many cases, detailed phenotypic data is provided for one example mutant in the main figures, and then additional mutants are described in detail in the supplement. Again, to facilitate comparisons between mutants, fingerprints could be provided for all mutants in a main figure, with detailed analyses moved to the supplements.

The logic was to dedicate one main figure to *psen2* (Fig. 3) as an example of an early-onset Alzheimer’s risk gene, and one to *sorl1* (previously Fig. 4) as an example of a late-onset Alzheimer’s risk gene. We focused on them in main figures as they are both tested again later (Fig. 5 and Fig. 6). Having said that, we agree that the fingerprints may be a better use of main figure space than the parameters plots. In addition to the above (fingerprints of late-onset Alzheimer’s risk genes in main figure), we rearranged the figures in the early-onset AD section to have the *psen2* F0 knockout fingerprint in main.

- The explication of the utility of behavioral fingerprinting on page 35 is somewhat confusing. The authors describe drugs used to treat depression as enriched among small molecules anti-correlating with the sorl1 fingerprint. However, in Figure 5 - Supplementary Figure 1, drugs used to treat depression are biased toward positive cosines, which are indicated as having a more similar fingerprint to sorl1. These drugs should be described as more present among compounds positively correlating with the *sorl1* fingerprint.

Sorry, the confusion is about “(anti-)correlating”. Precisely, we meant “correlating and/or anti-correlating”, not just anti-correlating. We changed to that wording. In short, the analysis is by design agnostic to whether compounds with a given annotation are found more on the positive cosines side (left side in Fig. 5–supplement 1a) or the negative cosines side (right side). This is because the dataset often includes both agonists and antagonists to a given pathway but these are difficult to annotate. For example, say 10 compounds in the dataset target the dopamine D4 receptor, but these are an unknown mix of agonists and antagonists. In this case, we want ZOLTAR to generate a low p-value when all 10 compounds are found at extreme ends of the list, regardless of which end(s) that is (e.g. top 8 and bottom 2 should give an extremely low p-value). Initially, we were splitting the list, for each annotation, into positive-cosine fingerprints and negative-cosine fingerprints and testing enrichment on both separately, but we think the current approach is better as it reflects better the cases we want to detect and considers all available examples for a given annotation in one test. In sum, yes, in this case drugs used to treat depression were mostly in the positive-cosine side, but the other drugs on the negative-cosine side also contributed to what the p-value is, so it reflects better the analysis to say “correlating and/or anticorrelating”. You can read more about our logic for the analysis in Methods (section *Behavioural pharmacology from* sorl1 *F0 knockout’s fingerprint*).

- The authors conclude the above-described section by stating: "*sorl1* knockout larvae behaved similarly to larvae treated with small molecules targeting serotonin signaling, suggesting that the loss of Sorl1 disrupted serotonin signaling." Directionality here may be important. Are all of the drugs targeting the serotonin transporter SSRIs or similar? If so, then a correct statement would be that loss of Sorl1 causes similar phenotypes to drugs enhancing serotonin signaling. Finally, based on the correlation between serotonin transporter inhibitor trazodone and the *sorl1* crispant phenotype, it is potentially surprising that the SSRI citalopram caused the opposite phenotype from *sorl1*, that is, increased sleep during the day and night. It is potentially interesting that this result was enhanced in mutants, and suggests dysfunction of serotonin signaling, but the statement that "our behavioral pharmacology approach correctly predicted from behaviour alone that serotonin signaling was disrupted" is too strong a conclusion.

We understand “disrupt” as potentially going either way, but this may not be the common usage. We changed to “altered”.

The point regarding directionality is excellent, however. We tested the proportion of serotonin transporter agonists and antagonists (SSRIs) on each side of the ranked list of small molecule fingerprints. We used the STITCH database for this analysis as it has more drug–target interactions, but likely less curated, than the Therapeutic Target Database (Szklarczyk et al., 2016). As with the Therapeutic Target Database, most fingerprints of compounds interacting with the serotonin transporter SLC6A4 were found on the side of positive cosines (p ~ 0.005 using the custom permutation test), which replicates Fig. 5a with a different source for the drug–target annotations (Author response image 5). On the side of positive cosines (small molecules which generate behavioural fingerprints correlating with the *sorl1* fingerprint), there were 2 agonists and 26 antagonists. On the side of negative cosines (small molecules which generate behavioural fingerprints anti-correlating with the *sorl1* fingerprint), there were 3 agonists and 2 antagonists. Using a Chi-squared test, this suggests a significant (p = 0.002) over-representation of antagonists (SSRIs) on the positive side (expected count = 24, vs. 26 observed) and agonists on the negative side (expected count = 1, vs. 3 observed). If SLC6A4 antagonists, i.e. SSRIs, indeed tend to cause a similar behavioural phenotype than knockout of *sorl1*, this would point in the direction of our original interpretation of the citalopram experiment; which was that excessive serotonin signalling is what causes the *sorl1* behavioural phenotype.

**Author response image 5. sa3fig5:** Using the STITCH database as source of annotations also predicts SLC6A4 as an enriched target for the *sorl1* behavioural fingerprint. Same figures as Fig. 5a,b but using the STITCH database (Szklarczyk et al., 2016) as source for the drug targets. (**a**) Compounds annotated by STITCH as interacting with the serotonin transporter SLC6A4 tend to generate behavioural phenotypes similar to the *sorl1* F0 knockout fingerprint. 40,522 compound–target protein pairs (vertical bars; 1592 unique compounds) are ranked from the fingerprint with the most positive cosine to the fingerprint with the most negative cosine in comparison with the mean *sorl1* F0 knockout fingerprint. Fingerprints of drugs that interact with SLC6A4 are coloured in yellow. Simulated p-value = 0.005 for enrichment of drugs interacting with SLC6A4 at the top (positive cosine) and/or bottom (negative cosine) of the ranked list by a custom permutation test. (**b**) Result of the permutation test for top and/or bottom enrichment of drugs interacting with SLC6A4 in the ranked list. The absolute cosines of the fingerprints of drugs interacting with SLC6A4 (n = 52, one fingerprint per compound) were summed, giving sum of cosines = 15.9. To simulate a null distribution, 52 fingerprints were randomly drawn 100,000 times, generating a distribution of 100,000 random sum of cosines. Here, only 499 random draws gave a larger sum of cosines, so the simulated p-value was p = 499/100,000 = 0.005 **.

If this were true, we would expect, as the reviewer suggested, SSRI treatment (citalopram or fluvoxamine) on control larvae to give a similar behavioural phenotype as knockout of *sorl1*. However, this generally did not appear to be the case (*sorl1* knockout fingerprint vs. SSRI-treated control fingerprint, cosine = 0.08 ± 0.35; Author response image 6).

**Author response image 6. sa3fig6:** *sorl1* F0 knockouts in comparison to controls treated with SSRIs. (**a**) *sorl1* F0 knockout fingerprints (baseline recordings and *sorl1* + H_2_O fingerprint from the citalopram experiment) in comparison with the scrambled-injected + citalopram (1 or 10 µM) fingerprints. Each dot represents the mean deviation from the same-clutch scrambled-injected H_2_O-treated mean for that parameter (z-score, mean ± SEM). (**b**) As in (a), *sorl1* F0 knockout fingerprints (baseline recordings and *sorl1* + H_2_O fingerprint from the fluvoxamine experiment) in comparison with the scrambled-injected + fluvoxamine (10 µM) fingerprint.

The comparison with trazodone is an interesting observation, but it is only a weak serotonin reuptake inhibitor (Ki for SLC6A4 = 690 nM, vs. 8.9 nM for citalopram; Owens et al., 1997) and it has many other targets, both as agonist or antagonist, including serotonin, adrenergic, and histamine receptors (Mijur, 2011). In any case, the average trazodone fingerprint does not correlate particularly well to the *sorl1* knockout fingerprint (cos = 0.3). Finally, the *sorl1* knockout behavioural phenotype could be primarily caused by altered serotonin signalling in the hypothalamus, where we found both the biggest difference in *tph1a/1b/2* HCR signal intensity (Fig. 5f) and the highest expression of *sorl1* across scRNA-seq clusters (Fig. 1– supplement 2). In this case, it would be correct to expect *sorl1* knockouts to react differently to SSRIs than controls, but it would be incorrect to expect SSRI treatment to cause the same behavioural phenotype, as it concurrently affects every other serotonergic neuron in the brain.

Finally, we agree the quoted conclusion was too strong given the current evidence. We since tested another SSRI, fluvoxamine, on *sorl1* knockouts.

- Also in reference to Figure 5: in panel c, data are presented as deviation from vehicle treated. Because of this data presentation choice, it's no longer possible to determine whether, in this experiment, sorl1 crispants sleep less at night relative to their siblings. Does citalopram rescue / reverse sleep deficits in sorl1 mutants?

On your first point, please see our response to Reviewer #3 (2)c and Author Response 2b above.

On “does citalopram rescue/reverse sleep deficits in *sorl1* mutants”: citalopram (and fluvoxamine) tends to *reverse* the key aspects of the *sorl1* knockout behavioural phenotype by reducing night-time activity (% time active and total Δ pixels), increasing night-time sleep, and shortening sleep latency (Author response image 7). Extrapolating from the hypothesis presented in Discussion, this may be interpreted as a hint that *sorl1* knockouts have reduced levels of 5-HT receptors, as increasing serotonin signalling using an SSRI tends to rescue the phenotype. However, we do not think that focusing on the significant behavioural parameters necessarily make sense here. Rather, one should take all parameters into account to conclude whether knockouts react differently to the drug than wild types (also see answer to Reviewer #3, (7) on this). For example, citalopram increased *more* the night-time sleep bout length of *sorl1* knockouts than the one of controls (Fig. 5), but this parameter was not modified by knockout of *sorl1* (Fig. 4). To explain the rationale more informally, citalopram is only used as a tool here to probe serotonin signalling in *sorl1* knockouts, whether it worsens or rescues the behavioural phenotype is somewhat secondary, the key question is whether knockouts react differently than controls.

**Author response image 7. sa3fig7:** Comparing untreated *sorl1* F0 knockouts vs treated with SSRIs. (**a**) *sorl1* F0 knockout fingerprints (baseline recordings and *sorl1* + H_2_O fingerprint from the citalopram experiment) in comparison with the *sorl1* knockout + citalopram (1 or 10 µM) fingerprints. Each dot represents the mean deviation from the same-clutch scrambled-injected H_2_O-treated mean for that parameter (z-score, mean ± SEM). (**b**) As in (a), *sorl1* F0 knockout fingerprints (baseline recordings and *sorl1* + H_2_O fingerprint from the fluvoxamine experiment) in comparison with the *sorl1* + fluvoxamine (10 µM) fingerprint.

Possible molecular pathways targeted by tinidazole, fenoprofen, and betamethasone are not described.

Tinidazole is an antibiotic, fenoprofen is a non-steroidal anti-inflammatory drug (NSAIDs), betamethasone is a steroidal anti-inflammatory drug. Interestingly, long-term use of NSAIDs reduces the risk of AD (in ’t Veld Bas A. et al., 2001). Several mechanisms are possible (Weggen et al., 2007), including reduction of Aβ42 production by interacting with γ-secretase (Eriksen et al., 2003). However, we did not explore the mechanism of action of these drugs on *psen2* knockouts so do not feel comfortable speculating. We do not know, for example, whether these findings apply to betamethasone.

Minor Comments:- On page 25, panel "g" should be labeled as "f".

Thank you!

- On page 35, a reference should be provided for the statement "From genomic studies of AD, we know that mutations in genes such as SORL1 modify risk by disrupting some biological processes.".

Thank you, this is now corrected. There were the same studies as mentioned in Introduction.

- On page 43, the word "and" should be added - "in wild-type rats and mice, overexpressing mutated human APP and PSEN1, AND restricting sleep for 21 days...".

Right, this sentence could be misread, we edited it. “overexpressing […]” only applied to the mice, not the rats (as they are wild-type); and both are sleep-deprived.

- On page 45, a reference should be provided for the statement "SSRIs can generally be used continuously with no adverse effects" and this statement should potentially be softened.

The reference is at the end of that sentence (Cirrito et al., 2011). You are correct though; we reformulated this statement to: “SSRIs can generally be used safely for many years”. SSRIs indeed have side effects.

- On page 54, a 60-minute rolling average is described as 45k rows, but this seems to be a 30-minute rolling average.

Thank you! We corrected. It should have been 90k rows, as in: 25 frames-per-second × 60 seconds × 60 minutes.

**Reviewer #2 (Recommendations For The Authors)**:"As we observed in the scRNA-seq data, most genes tested (appa, appb, psen1, psen2, apoea, cd2ap, sorl1) were broadly expressed throughout the 6-dpf brain (Fig. 1d and Fig. 1 supplement 3 and 4)."- *apoea* and *appb* are actually not expressed highly in the scRNA-seq data, and the *apoea* in situ looks odd, as if it has no expression. The *appb* gene mysteriously does not look as though it has high expression in the Raj data, but it is clearly expressed based on the in situ. I had previously noticed the same discrepancy, and I attribute it to the transcriptome used to map the Raj data, as the new DanioCell data uses a new transcriptome and indicates high *appb* expression in the brain. Please point out the discrepancy and possible explanation, perhaps in the figure legend.

All excellent points, thank you. We included them directly in Results text.

"most of these were expressed in the brain of 5-6-dpf zebrafish larvae, suggesting they play a role in early brain development or function."- Evidence of expression does not suggest function, particularly not a function in brain development. As one example, almost half of the genome is expressed prior to the maternal-zygotic transition but does not have a function in those earliest stages of development. There are numerous other instances where expression does not equal function. Please change the sentence even as simply as "it is possible that they".

We mostly agree and edited to “[…], so they could play a role […]”.

Out of curiosity, we plotted, for each zebrafish developmental stage, the proportion of Alzheimer’s risk gene orthologues expressed in comparison to the proportion of all genes expressed (Author response image 8). We defined “all genes” as every gene that is expressed in at least one of the developmental stages (n = 24,856), not the complete transcriptome, to avoid including genes that are never expressed in the brain or whose expression is always below detection limit. We counted a gene as “expressed” if at least three cells had detectable transcripts. Using these definitions, 82 ± 7% of genes are expressed during development. For every developmental stage except 5 dpf (so 11/12), a larger proportion of Alzheimer’s risk genes than all genes are expressed (+5 ± 4%).

**Author response image 8. sa3fig8:** Proportion of Alzheimer’s risk genes orthologues expressed throughout zebrafish development. Proportion of Alzheimer’s risk genes orthologues (n = 42) and all genes (n = 24,856) expressed in the zebrafish brain at each developmental stage, from 12 hours post-fertilisation (hpf) to 15 days post-fertilisation (dpf). “All genes” corresponds to every gene expressed in the brain at any of the developmental stages, not the complete transcriptome. A gene is considered “expressed” (green) if at least three cells had detectable transcripts. Single-cell RNA-seq dataset from Raj et al., 2020.

"This frame-by-frame analysis has several advantages over previous methods that analysed activity data at the one-minute resolution."- Which methods are these? There are no citations. There are certainly existing methods in the zebrafish field that can produce similar data to the method developed for this project. This new package is useful, as most existing software is not written in R, so it would help scientists who prefer this programming language. However, I would be careful not to oversell its novelty, since many methods do exist that produce similar results.

We added the references. There were referenced above after “we combined previous sleep/wake analysis methods”, but should have been referenced again here.

We are not convinced by this criticism. We would obviously not claim that the FramebyFrame package is as sophisticated and versatile as video-tracking tools like SLEAP or DeepLabCut, but we do think it answers a genuine need that was not addressed by other methods. Specifically, we know of many labs recording pixel count data across multiple days using the Zebrabox or DanioVision (we added support for DanioVision data after submission), but there were no packages to extract behavioural parameters from these data. Other methods involved standalone scripts with no documentation or version tracking. We would concede the FramebyFrame package is mostly targeted at these labs, but we already know of six labs routinely using it and were recently contacted by a researcher tracking Daphnia in the Zebrabox.

"F0 knockouts of both cutches" - "clutches"

Thank you!

**Reviewer #3 (Recommendations For The Authors):**
I would suggest totally revamping the Introduction section, and being sure to provide readers with the context and background they need for the data that comes thereafter. Key areas to touch on, in no particular order, include:• Far more detail on the behavioral pharm screen upon which this paper builds, as a brief overview of that approach and the data generated are needed.

Thank you for the suggestion, we added a sentence hinting at this work in the last Introduction paragraph.

• Limitations of current zebrafish sleep/arousal assays that motivated the authors to develop a new, temporally high-resolution system.

We think this is better explained in Results, as is currently. For example, we need to point to Fig. 2–supplement 2a,b,c to explain that one-minute methods were missing sleep bouts and how FramebyFrame resolves this issue.

• A paragraph about sleep and AD, that does a better job of citing work in humans, mammalian, and invertebrate models that motivate the interest in the connection pursued here.

Sorry, we think this would place too much focus on sleep and AD. We want the main topic of the paper to be the behavioural pharmacology approach, not AD or sleep *per se*. As the Introduction states, we see Alzheimer’s risk genes as a case study for the behavioural pharmacology approach, rather than the reason why the approach was developed. Additionally, presenting sleep and AD in Introduction risks sounding like ZOLTAR is specifically designed for this context, while we conceived of it as much more generalisable and explicitly encourage its use to study genes associated to other diseases. Note that the paragraph you suggest is, we think, mostly present in Discussion (section Disrupted sleep and serotonin signalling […]).

• I modestly suggest eliminating making such a strong case for a gene-first approach being the best way to understand disease. It is not a zero-sum game, and there is plenty to learn from proteomics, metabolomics, etc. I suspect nobody will argue with the authors saying they leveraged the strength of their system and focused on key AD genes of interest.

From your point below, we understand the following quote is the source of the issue: “For finding causal processes, studying the genome, rather than the transcriptome or epigenome, is advantageous because the chronology from genomic variant to disease is unambiguous […]”. We did not want to suggest it is a zero-sum game, but we now understand how it can be read this way. We adapted slightly the wording. What we want to do is highlight the causality argument as the advantage of the genomics approach. We feel we do not read this argument often enough, while it remains a ‘magic power’ of genomics. One essentially does not have to worry about causality when studying a pathogenic germline variant, while it is a constant concern when studying the transcriptome or epigenome (i.e. did the change in this transcript’s level cause disease, or vice-versa?). To take an example in the context of AD, arguments based on genomics (e.g. Down syndrome or *APP* duplication) are often the definite arbiters when debating the amyloid hypothesis, exactly because their causality cannot be doubted.

Minor comments(1) The opening of the introduction is perhaps overly broad, spending an entire paragraph on genome vs transcriptome, etc and making the claim that a gene-first approach is the best path. It isn't zero-sum, and the authors could just get right into AD and study genes of interest. Similar issues occur throughout the manuscript, with sentences/paragraphs that are not necessarily needed.

Please see our answer to your previous point. On the introduction being overly broad, we perfectly agree it is broad, but related to your point about presenting sleep and AD in the Introduction, we wish to talk about finding causal processes from genomics findings using behavioural pharmacology. We purposefully present research on AD as one instance of this broader goal, not the primary topic of the paper.

Another example are these sentences, which could be totally removed as the following paragraph starts off making the same point much more succinctly. "From genomic studies of AD, we know that mutations in genes such as SORL1 modify risk by disrupting some biological processes. Presumably, the same processes are disrupted in zebrafish sorl1 knockouts, and some caused the behavioural alterations we observed. Can we now follow the thread backwards and predict some of the biological processes in which Sorl1 is involved based on the behavioural profile of *sorl1* knockouts?"

Thanks for the suggestion, but we think these sentences are useful to place back this Results section in the context of the Introduction. Think of the paper as mainly about the behavioural pharmacology approach, not on Alzheimer’s risk genes. The function of the paragraph here is not simply to explain the method by which we decided to study *sorl1*; it is to reiterate the rationale behind the behavioural pharmacology approach so that the reader understands where this Results section fits in the overall structure.

(2) Related to the above, the authors use lecanemab as an example to support their approach, but there has been a great deal of controversy regarding this drug. I don't think such extensive justification is needed. This study uses AD risk genes as a case study in a newly developed behavioral pharm pipeline. A great deal of the rest of the intro seems to just fill space and could be more focused on the study at hand. Interestingly, after gene selection, the next step in their pipeline is sleep/wake analysis yet nothing is covered about AD and sleep in the intro. Some justification of that approach (why focus on sleep/wake as a starting point for behavioral pharm rather than learning and memory?) would be a better use of intro space.

There has indeed been controversy about lecanemab, but even the harshest critiques of the amyloid hypothesis concede that it slows down cognitive decline (Espay et al., 2023). That is all that is needed to support our argument, which is that research on AD started primarily from genomics and thereby yielded a disease-modifying drug. The controversy seems mostly focused on whether this effect size is clinically significant, and we think we correctly represent this uncertainty (e.g. “antibodies against Aβ such as lecanemab show promise in slowing down disease progression” and “the beneficial effects from targeting Aβ aggregaRon currently remain modest”).

Your next point is entirely fair. We mostly answered it above. To explain further, the primary reason why we measured sleep/wake behaviour is to match the behavioural dataset from Rihel et al., 2010 so we can use it to make predictions, not to study sleep in the context of AD *per se*. Sure, perhaps learning and memory would have been interesting, but we do not know of any study testing thousands of small molecules on zebrafish larvae during a memory task. We understand it can be slightly confusing though, as we then spend a paragraph of Discussion on sleep as a causal process in AD, but we obviously need to discuss this topic given the findings. However, to reiterate, we purposefully designed FramebyFrame and ZOLTAR to be useful beyond studying sleep/wake behaviour. For example, FramebyFrame would not calculate 17 behavioural parameters if the only goal was to measure sleep. We now mention the Rihel et al., 2010 study in the Introduction as you suggested above (“Far more detail on the behavioral pharm screen […]”), as that is the real reason why sleep/wake behaviour was measured in the first place.

(3) Also related to the above, another more relevant point that could be talked about in the intro is the need for more refined approaches to analyze sleep in zebrafish, given the effort that went into the new analysis system described here. Again, I think the context for why the authors developed this system would be more meaningful than the current content.

Thank you, we think we answered this point above (especially below Limitations of current zebrafish sleep/arousal assays […]).

(4) GWAS can stand for Genome-wide associate studies (plural) so I do not think the extra "s" is needed (GWASs) .

Indeed, that seems to be the common usage. Thank you.

(5) AD candidate risk genes were determined from loci using "mainly statistic colocalization". Can the authors add a few more details about what was done and what the "mainly" caveat refers to?

“Mainly” simply refers to the fact that other methods were used by Schwartzentruber et al. (2021) to annotate the GWAS loci with likely causal genes, but that most calls were ultimately made from statistic colocalisation. Readers can refer to this work to learn more about the methods used.

(6) The authors write "The loss of psen1 only had mild effects on behaviour" but I think they mean "sleep behaviors" as there could be many other behaviors that are disrupted but were not assessed. The same issue a few sentences later with "Behaviour during the day was not affected" and at the end of the following paragraph.

Yes, that would be more precise, thank you.

(7) For the Sorl1 pharmacology data, it is very hard to understand what is being measured behaviorally. Are the authors measuring sleep +/- citalopram, or something else, and why the change to Euclidean distance rather than all the measures we were just introduced to earlier in the manuscript?

We understand these plots (Fig. 5c,d) are less intuitive, but it is important that we show the difference in behaviour compared to H_2_O-treated larvae of same genotype. The claim is that citalopram has a larger effect on knockouts than on controls, so the reader needs to focus on the effect of the drug on each genotype, not on the effect of *sorl1* knockout. We added the standard fingerprints (i.e. setting controls to z-score = 0) here in Author response figures.

Euclidean distance takes as input all the measures we introduced. The point is precisely not to select a single measure. For example, say we were only plotting active bout number during the day, we would conclude that 10 µM citalopram has the same effect on knockouts and controls. Conversely, if we had taken sleep bout length at night, we would conclude 10 µM has a stronger effect on knockouts. What is the correct parameter to select? Using Euclidean distance resolves this by taking all parameters into account, rather than arbitrarily choosing one.

And what exactly is a "given spike in serotonin"? and how is this hypothesis the conclusion based on the lack of evidence for the second hypothesis? As the authors say, there could be other ways *sorl1* knockouts are more sensitive to citalopram, so the absence of evidence for one hypothesis certainly does not support the other hypothesis.

We mean a given release of serotonin in the synaptic cleft. We have fixed this wording.

We tend to disagree on the second point. We can think of two ways that *sorl1* knockouts are more sensitive to citalopram: (1) they produce more serotonin, so blocking reuptake causes a larger spike in knockouts; or (2) blocking reuptake causes the same increase in both knockouts and wild-types but knockouts react more strongly to serotonin. We cannot in fact think of another way to explain the citalopram results. Not finding overwhelming evidence for (1) surely supports (2) somewhat, even if we do not have direct evidence for it. As an analogy, if two diagnoses are possible for a patient, testing negative for the first one supports the other one, even before it is directly tested.

(8) Again some language is used without enough care. Fish are referred to as "drowsier" under some drug conditions. How do the authors know the animal is drowsy? The phenotype is more specific - more sleep, less activity.

Thank you, we switched to “Furthermore, fenoprofen worsened the day-time hypoactivity of *psen2* knockout larvae […]”.

(9) This sentence is misleading as it gives the impression that results in this manuscript suggest the conclusion: "Our observation that disruption of genes associated with AD diagnosis after 65 years reduces sleep in 7-day zebrafish larvae suggest that disrupted sleep may be a common mechanism through which these genes exert an effect on risk." That idea is widely held in the field, and numerous other previous manuscripts/reviews should be cited for clarity of where this hypothesis came from.

This idea is not widely held in the field. You likely read this point as “disrupted sleep is a risk factor for AD”, which, yes, is widely discussed in the field, but is not precisely what we are saying. We hypothesise that mutations in some of the Alzheimer’s risk genes *cause* disrupted sleep, possibly from a very early age, which then causes AD decades later. Studies and reviews on sleep and AD rarely make this hypothesis, at least not explicitly. The closest we know of are a few recent human genetics studies, typically using Mendelian Randomisation, finding that higher genetic risk of AD correlates with some sleep phenotypes, such as sleep duration (Chen et al., 2022; Leng et al., 2021). The work of Muto et al. (2021) is particularly interesting as it found correlations between higher genetic risk of AD and some sleep phenotypes in men in their early twenties, which seems unlikely to be a consequence of early pathology (Muto et al., 2021). Note, however, that even these studies do not mention sleep possibly being disrupted early in development, which is what our findings in zebrafish larvae support. As we mention, we think a team should test whether sleep is different in infants at higher genetic risk of AD, essentially performing an analogous, but obviously much more difficult, experiment as we did in zebrafish larvae. We do not know of any study testing this or even raising this idea, so evidently it is not widely held. Having said that, the studies we mention here were not referenced in the Discussion paragraph. We have now corrected this.

Ashlin TG, Blunsom NJ, Ghosh M, Cockcroft S, Rihel J. 2018. Pitpnc1a Regulates Zebrafish Sleep and Wake Behavior through Modulation of Insulin like Growth Factor Signaling. *Cell Rep* 24:1389–1396. doi:10.1016/j.celrep.2018.07.012

Chen D, Wang X, Huang T, Jia J. 2022. Sleep and Late-Onset Alzheimer’s Disease: Shared Genetic Risk Factors, Drug Targets, Molecular Mechanisms, and Causal Effects. *Front Genet* 13. doi:10.3389/fgene.2022.794202

Cirrito JR, Disabato BM, Restivo JL, Verges DK, Goebel WD, Sathyan A, Hayreh D, D’Angelo G, Benzinger T, Yoon H, Kim J, Morris JC, Mintun MA, Sheline YI. 2011. Serotonin signaling is associated with lower amyloid-β levels and plaques in transgenic mice and humans. *Proc Natl Acad Sci U S A* 108:14968–14973. doi:10.1073/pnas.1107411108

Dean DC, Jerskey BA, Chen K, Protas H, Thiyyagura P, Roontiva A, O’Muircheartaigh J, Dirks H, Waskiewicz N, Lehman K, Siniard AL, Turk MN, Hua X, Madsen SK, Thompson PM, Fleisher AS, Huentelman MJ, Deoni SCL, Reiman EM. 2014. Brain Differences in Infants at Differential Genetic Risk for Late-Onset Alzheimer Disease A Cross-sectional Imaging Study. *JAMA Neurol* 71:11–22. doi:10.1001/jamaneurol.2013.4544

Eriksen JL, Sagi SA, Smith TE, Weggen S, Das P, McLendon DC, Ozols VV, Jessing KW, Zavitz KH, Koo EH, Golde TE. 2003. NSAIDs and enantiomers of flurbiprofen target γ-secretase and lower Aβ42 in vivo. *J Clin Invest* 112:440–449. doi:10.1172/JCI18162

Espay AJ, Herrup K, Kepp KP, Daly T. 2023. The proteinopenia hypothesis: Loss of Aβ42 and the onset of Alzheimer’s Disease. *Ageing Res Rev* 92:102112. doi:10.1016/j.arr.2023.102112

Hoffman EJ, Turner KJ, Fernandez JM, Cifuentes D, Ghosh M, Ijaz S, Jain RA, Kubo F, Bill BR, Baier H, Granato M, Barresi MJF, Wilson SW, Rihel J, State MW, Giraldez AJ. 2016. Estrogens Suppress a Behavioral Phenotype in Zebrafish Mutants of the Autism Risk Gene, CNTNAP2. *Neuron* 89:725–733. doi:10.1016/j.neuron.2015.12.039

in ’t Veld Bas A, Ruitenberg A, Hofman A, Launer LJ, van Duijn CM, Stijnen T, Breteler MMB, Stricker BHC. 2001. Nonsteroidal Anti inflammatory Drugs and the Risk of Alzheimer’s Disease. *N Engl J Med* 345:1515–1521. doi:10.1056/NEJMoa010178

Jagirdar R, Fu C-H, Park J, Corbett BF, Seibt FM, Beierlein M, Chin J. 2021. Restoring activity in the thalamic reticular nucleus improves sleep architecture and reduces Aβ accumulation in mice. *Sci Transl Med* 13:eabh4284. doi:10.1126/scitranslmed.abh4284

Jiang H, Newman M, Lardelli M. 2018. The zebrafish orthologue of familial Alzheimer’s disease gene PRESENILIN 2 is required for normal adult melanotic skin pigmentation. *PLOS ONE* 13:e0206155. doi:10.1371/journal.pone.0206155

Jiang H, Pederson SM, Newman M, Dong Y, Barthelson K, Lardelli M. 2020. Transcriptome analysis indicates dominant effects on ribosome and mitochondrial function of a premature termination codon mutation in the zebrafish gene psen2. *PloS One* 15:e0232559. doi:10.1371/journal.pone.0232559

Joo W, Vivian MD, Graham BJ, Soucy ER, Thyme SB. 2021. A Customizable Low-Cost System for Massively Parallel Zebrafish Behavioral Phenotyping. *Front Behav Neurosci* 14.

Joubert L, Hanson B, Barthet G, Sebben M, Claeysen S, Hong W, Marin P, Dumuis A, Bockaert J. 2004. New sorting nexin (SNX27) and NHERF specifically interact with the 5-HT4a receptor splice variant: roles in receptor targeting. *J Cell Sci* 117:5367–5379. doi:10.1242/jcs.01379

Leng Y, Ackley SF, Glymour MM, Yaffe K, Brenowitz WD. 2021. Genetic Risk of Alzheimer’s Disease and Sleep Duration in Non-Demented Elders. *Ann Neurol* 89:177–181. doi:10.1002/ana.25910

Mitchell PB, Hadzi-Pavlovic D. 2000. Lithium treatment for bipolar disorder. *Bull World Health Organ* 78:515–517.

Mikur A. 2011. Trazodone: properties and utility in multiple disorders. *Expert Rev Clin Pharmacol* 4:181–196. doi:10.1586/ecp.10.138

Munoz-Torrero D. 2008. Acetylcholinesterase Inhibitors as Disease-Modifying Therapies for Alzheimer’s Disease. *Curr Med Chem* 15:2433–2455. doi:10.2174/092986708785909067

Muto V, Koshmanova E, Ghaemmaghami P, Jaspar M, Meyer C, Elansary M, Van Egroo M, Chylinski D, Berthomier C, Brandewinder M, Mouraux C, Schmidt C, Hammad G, Coppieters W, Ahariz N, Degueldre C, Luxen A, Salmon E, Phillips C, Archer SN, Yengo L, Byrne E, Collette F, Georges M, Dijk D-J, Maquet P, Visscher PM, Vandewalle G. 2021. Alzheimer’s disease genetic risk and sleep phenotypes in healthy young men: association with more slow waves and daytime sleepiness. *Sleep* 44. doi:10.1093/sleep/zsaa137

Myers-Turnbull D, Taylor JC, Helsell C, McCarroll MN, Ki CS, Tummino TA, Ravikumar S, Kinser R, Gendelev L, Alexander R, Keiser MJ, Kokel D. 2022. Simultaneous analysis of neuroactive compounds in zebrafish. doi:10.1101/2020.01.01.891432

Owens MJ, Morgan WN, Plok SJ, Nemeroff CB. 1997. Neurotransmiker receptor and transporter binding profile of antidepressants and their metabolites. *J Pharmacol Exp Ther* 283:1305– 1322.

Özcan GG, Lim S, Leighton PL, Allison WT, Rihel J. 2020. Sleep is bi-directionally modified by amyloid beta oligomers. *eLife* 9:e53995. doi:10.7554/eLife.53995

Quiroz YT, Schultz AP, Chen K, Protas HD, Brickhouse M, Fleisher AS, Langbaum JB, Thiyyagura P, Fagan AM, Shah AR, Muniz M, Arboleda-Velasquez JF, Munoz C, Garcia G, Acosta-Baena N, Giraldo M, Tirado V, Ramírez DL, Tariot PN, Dickerson BC, Sperling RA, Lopera F, Reiman EM. 2015. Brain Imaging and Blood Biomarker Abnormalities in Children With Autosomal Dominant Alzheimer Disease: A Cross-Sectional Study. *JAMA Neurol* 72:912–919. doi:10.1001/jamaneurol.2015.1099

Relkin NR. 2007. Beyond symptomatic therapy: a reexaminatoon of acetylcholinesterase inhibitors in Alzheimer’s disease. *Expert Rev Neurother* 7:735–748. doi:10.1586/14737175.7.6.735

Rihel J, Prober DA, Arvanites A, Lam K, Zimmerman S, Jang S, Haggarty SJ, Kokel D, Rubin LL, Peterson RT, Schier AF. 2010. Zebrafish Behavioral Profiling Links Drugs to Biological Targets and Rest/Wake Regulation. *Science* 327:348–351. doi:10.1126/science.1183090

Sleegers K, Brouwers N, Gijselinck I, Theuns J, Goossens D, Wauters J, Del-Favero J, Cruts M, van Duijn CM, Van Broeckhoven C. 2006. APP duplication is sufficient to cause early onset Alzheimer’s dementia with cerebral amyloid angiopathy. *Brain J Neurol* 129:2977–2983. doi:10.1093/brain/awl203

Sun L, Zhou R, Yang G, Shi Y. 2017. Analysis of 138 pathogenic mutations in presenilin-1 on the in vitro production of Aβ42 and Aβ40 peptides by γ-secretase. *Proc Natl Acad Sci* 114:E476– E485. doi:10.1073/pnas.1618657114

Szklarczyk D, Santos A, von Mering C, Jensen LJ, Bork P, Kuhn M. 2016. STITCH 5: augmenting protein–chemical interaction networks with tissue and affinity data. *Nucleic Acids Res* 44:D380–D384. doi:10.1093/nar/gkv1277

Weggen S, Rogers M, Eriksen J. 2007. NSAIDs: small molecules for prevention of Alzheimer’s disease or precursors for future drug development? *Trends Pharmacol Sci* 28:536–543. doi:10.1016/j.Jps.2007.09.004

Wiltschko AB, Tsukahara T, Zeine A, Anyoha R, Gillis WF, Markowitz JE, Peterson RE, Katon J, Johnson MJ, Daka SR. 2020. Revealing the structure of pharmacobehavioral space through motion sequencing. *Nat Neurosci* 23:1433–1443. doi:10.1038/s41593-020-00706-3

Yang T, Arslanova D, Gu Y, Augelli-Szafran C, Xia W. 2008. Quantification of gamma-secretase modulation differentiates inhibitor compound selectivity between two substrates Notch and amyloid precursor protein. *Mol Brain* 1:15. doi:10.1186/1756-6606-1-15